# *Streptococcus pyogenes* EVs induce the alternative inflammasome via caspase-4/-5 in human monocytes

Kathrin Krause [ID] [1,9 ✉], Sandra Franch Arroyo[1,2,9], Matteo Ugolini[1], Tonya Kueck[1], Timothy J Sullivan [ID] [1], Eric J C Gálvez [ID] [1,3], Matthias Muenzner[1], Christian Goosmann[4], Volker Brinkmann [ID] [4], Christian K Frese[1], Kathirvel Alagesan [ID] [1], Tim Vierbuchen [ID] [5,6], Holger Heine [ID] [5], Ulrike Resch [ID] [7 ✉], Leif E Sander[8] & Emmanuelle Charpentier [ID] [1,2 ✉]

## Abstract

The sensing of Gram-negative Extracellular Vesicles (EVs) by the innate immune system has been extensively studied in the past decade. In contrast, recognition of Gram-positive EVs by innate immune cells remains poorly understood. Comparative genome-wide transcriptional analysis in human monocytes uncovered that *S. pyogenes* EVs induce proinflammatory signatures that are markedly distinct from those of their parental cells. Among the 209 genes exclusively upregulated by EVs, caspase-5 prompted us to study inflammasome signaling pathways in depth. We show that lipoteichoic acid (LTA), a structural component of Gram-positive bacterial membranes present on EVs from *S. pyogenes* and other Gram-positive species, is sensed by TLR2 which triggers the alternative inflammasome composed of NLRP3 and the inflammatory caspases-4/-5 to mount an IL-1β response without inducing cell death. For *S. pyogenes*, we identify TLR8 as a sensor to mediate caspase-4/-5-dependent IL-1β secretion. Notably, inflammasome activation by intact bacteria is independent of the global virulence regulator CovS in monocytes. Overall, our study highlights a new role for TLR2 and caspase-4/-5 in the recognition of Gram-positive EVs in human monocytes.

**Keywords** *Streptococcus pyogenes*; Monocytes; Extracellular Vesicles; Caspases; Inflammasome
**Subject Categories** Immunology; Membranes & Trafficking; Microbiology, Virology & Host Pathogen Interaction

## Introduction

*Streptococcus pyogenes* (*S. pyogenes*) is a strict human pathogen that causes superficial infections (e.g., impetigo) as well as life-threatening invasive disease (e.g., Streptococcal toxic shock syndrome), with a global burden of at least 500,000 deaths annually (Carapetis et al, 2005). *S. pyogenes* expresses a wide array of virulence components that facilitate its survival within the host by hijacking the activity of immune cells (Graham et al, 2002; Tsatsaronis et al, 2014; Hynes and Sloan, 2016). Factors associated with bacterial virulence can be classified according to their location: membrane-bound, membrane-anchored, or cytosolic. Interestingly, several virulence mediators present in *S. pyogenes* extracellular vesicles (*Spy* EVs) do not possess a secretion signal peptide and are therefore not secreted through the canonical bacterial secretory pathway (Sec) (Lei et al, 2000; Resch et al, 2016). Since no alternative secretion mechanism has been identified in *S. pyogenes*, it is likely that EVs contribute to the release of components that cannot be secreted by the Sec pathway (Resch et al, 2016).

EVs are nanoparticles consisting of a lipid bilayer shell encapsulating proteins and nucleic acids derived from their parental cells. The release of EVs is conserved across all domains of life to coordinate intra- and interspecies communication via cellular components (Deatheragea et al, 2012; Yoon et al, 2014; Woith et al, 2019). In Gram-negative bacteria, the composition, biogenesis, and functions of outer membrane vesicles (OMVs) in bacterial physiology have been extensively characterized (Schwechheimer and Kuehn, 2015; Orench-Rivera and Kuehn, 2016). Numerous reports have addressed the intricate communication process that occurs between the host and OMVs of pathogens or microbiota species (Kaparakis-Liaskos and Ferrero, 2015; Ñahui Palomino et al, 2021). Similarly, over the last few decades, numerous studies have demonstrated that Gram-positive model

[1]Max Planck Unit for the Science of Pathogens, Berlin D-10117, Germany. [2]Institute for Biology, Humboldt University, Berlin D-10115, Germany. [3]Roche Pharma Research and Early Development, Roche Innovation Center Basel, Basel CH-4051, Switzerland. [4]Max Planck Institute for Infection Biology, Microscopy Core Facility, Berlin D-10117, Germany. [5]Division of Innate Immunity, Research Center Borstel - Leibniz Lung Center, Airway Research Center North (ARCN), German Center for Lung Research (DZL), Borstel D-23845, Germany. [6]Program in Innate Immunity, University of Massachusetts Medical School, Worcester, MA 01605, USA. [7]Medical University of Vienna, Center of Physiology and Pharmacology, Department of Vascular Biology and Thrombosis Research, Vienna A-1090, Austria. [8]Department of Infectious Diseases and Respiratory Medicine, Charité–Universitätsmedizin, Berlin D-13353, Germany. [9]These authors contributed equally: Kathrin Krause, Sandra Franch Arroyo. ✉E-mail: krause@mpusp.mpg.de; ulrike.resch@meduniwien.ac.at; research@emmanuelle-charpentier.org

organisms and pathogens also release EVs (Lee et al, 2009; Rivera et al, 2010; Brown et al, 2015). However, only a few of these studies have addressed the impact of Gram-positive EVs on the host (Gurung et al, 2011; Olaya-Abril et al, 2014; Wang et al, 2020).

Previously, our laboratory and others have shown that *Spy* EVs encapsulate virulence factors, highlighting their potential role in modulating the host-pathogen interface (Biagini et al, 2015; Resch et al, 2016; Uhlmann et al, 2016; Murase et al, 2021). Several groups have reported that enrichment and exclusion of bacterial components in EVs, a process termed cargo selectivity, commonly occurs (Haurat et al, 2011; Elhenawy et al, 2014; Veith et al, 2014). For example, *Spy* EVs are enriched in the Streptococcal inhibitor of the complement (Sic) and Streptolysin O (SLO) compared to their parental cells (Resch et al, 2016), which are components that have been associated with pathogenicity and virulence (Harder et al, 2009; Pence et al, 2010; Nasser et al, 2014). Due to their small size and resistance to degradation, bacterial EVs are able to diffuse and travel long distances in the human body, reaching locations that their parental cells might not access (Tulkens et al, 2020). In addition, microbial EVs can be internalized and/or recognized through pathogen recognition receptors (PRRs), which may differ from the PRRs participating in the recognition of their parental cells (Vanaja et al, 2016).

The delivery of Gram-negative bacterial components including lipopolysaccharide (LPS) to the cytosol of host cells via OMVs has been shown to promote canonical as well as non-canonical inflammasome assembly, leading to caspase-1-dependent cleavage and subsequent release of the proinflammatory cytokines IL-1α, IL-1β, and IL-18 as well as the induction of pyroptotic cell death (Cecil et al, 2017; Finethy et al, 2017; Yang et al, 2020). Activation of this large multiprotein complex requires existing or de-novo synthetized inflammasome protein components and their signal-induced oligomerization: PRR-engagement on the cell surface stimulates NFκB-dependent gene expression of inactive precursor molecules for the cytokines (priming). Subsequently, the presence of bacterial products within the cytosol is sensed by various NOD-like receptors (NLRs) triggering oligomerization and recruitment of caspase-1 (activation). However, unlike macrophages, LPS has been shown to trigger an alternative one-step inflammasome activation pathway in human monocytes, which is dependent on TLR4 signaling and does not require LPS internalization or priming (Netea et al, 2009; Gaidt et al, 2016; Gritsenko et al, 2020). In addition to TLR4, a more recent study implicated multiple TLRs to activate the alternative inflammasome in human monocytes leading to IL-1β secretion, indicating that pathogen-associated molecular patterns (PAMPs) other than LPS can be recognized by this pathway (Unterberger et al, 2023). However, to date, no publication reports activation of the alternative inflammasome in response to live, Gram-positive bacteria or their secreted EVs.

In the present study, we have comparatively examined the response of primary human monocytes to *Spy* EVs and their parental cells. Interestingly, although both stimuli induce a significant overlap in the monocyte's response, we identified specific signatures triggered by either stimulus. We provide evidence, that *Spy* EVs are sensed by TLR2 on monocytes which triggers the alternative inflammasome comprised of NLRP3 and the inflammatory caspases-4 and -5, leading to the release of IL-1β without inducing cell death. We identify lipoteichoic acid (LTA), a

structural component of Gram-positive bacterial membranes, present on EVs but also in EV-depleted culture supernatant, as the ligand for TLR2. Accordingly, EVs from other Gram-positive species stimulate caspase-4/-5-dependent IL-1β secretion, underlining the conservation of this mechanism. In contrast, inflammasome activation in response to infection with *S. pyogenes* requires TLR8 and is characterized by ASC speck formation and GSDMD-dependent pyroptotic cell death, indicating that distinct innate immune pathways are involved in IL-1β production in response to *S. pyogenes* or its EVs.

# Results

## Transcriptome analysis reveals a unique set of genes that are upregulated upon encounter with *Spy* EVs

Our laboratory has previously characterized the content of *Spy* EVs derived from the hypervirulent clinical isolate ISS3348 by proteomic, RNA sequencing, and lipidomic analysis (Resch et al, 2016). *Spy* EVs and their parental cells show an asymmetrical distribution in their composition (e.g., of virulence factors) (Resch et al, 2016), supporting the idea that EVs might trigger distinct responses than *S. pyogenes* cells. To characterize and compare the response of innate immune cells when encountering *Spy* EVs or intact bacteria, we performed RNA sequencing on human primary monocytes (Dataset EV1) and B-cell Leukemia C/EBPα Estrogen Receptor clone 1 (BLaER1)-derived monocytes (Dataset EV2) after incubation with *Spy* EVs or infection with *S. pyogenes*. *Spy* EV purification and physiochemical characterization is summarized in Fig. EV1A,B. EV-quantification on the basis of size and concentration was initially performed using Nanoparticle tracking analysis (NTA) (Fig. EV1C) and confirmed by Tunable Resistive Pulse Sensing (TRPS), both methodologies proven to be more sensitive as compared to protein-content based quantification. To rule out any potential contamination of the *Spy* EVs with LPS, vesicle preparations were tested using the Pro-Q™ Emerald 300 lipopolysaccharide gel stain (Fig. EV1D) and mass spectrometry (Appendix Fig. S1). We did not detect any LPS in our EV preparations. Transmission electron microscopy (TEM) was used to visualize the EVs (Fig. EV1E) as previously described (Resch et al, 2016). While there are no published reports showing quantitative data on the number of EVs released by a bacterium during an in vivo infection, we found that the ratio between *S. pyogenes* and its EVs in THB culture is ~20 (17.7) EVs per bacterial cell. To assess a suitable dose for *Spy* EVs, we challenged human monocytes with increasing particle-concentrations of EVs and measured IL-1β and IL-6 in cell culture supernatants at 18 h post-stimulation (Fig. EV1F). We observed an EV-particle-dose-dependent release of both cytokines, however, secretion of IL-1β was considerably lower than that of IL-6, which explains why a dose of $7 \times 10^8$ EVs (corresponding to 7000 EVs/cell) was defined for future experiments.

To obtain an unbiased view on the early (4 h) transcriptional response of monocytes to *Spy* EVs or infections with *S. pyogenes*, we performed RNA sequencing as outlined in Fig. 1A. Principle component analysis (PCA) showed that samples were clearly separated and clustered by treatment, indicating that donor variability had minor impact in our experimental setting (Fig. 1B).

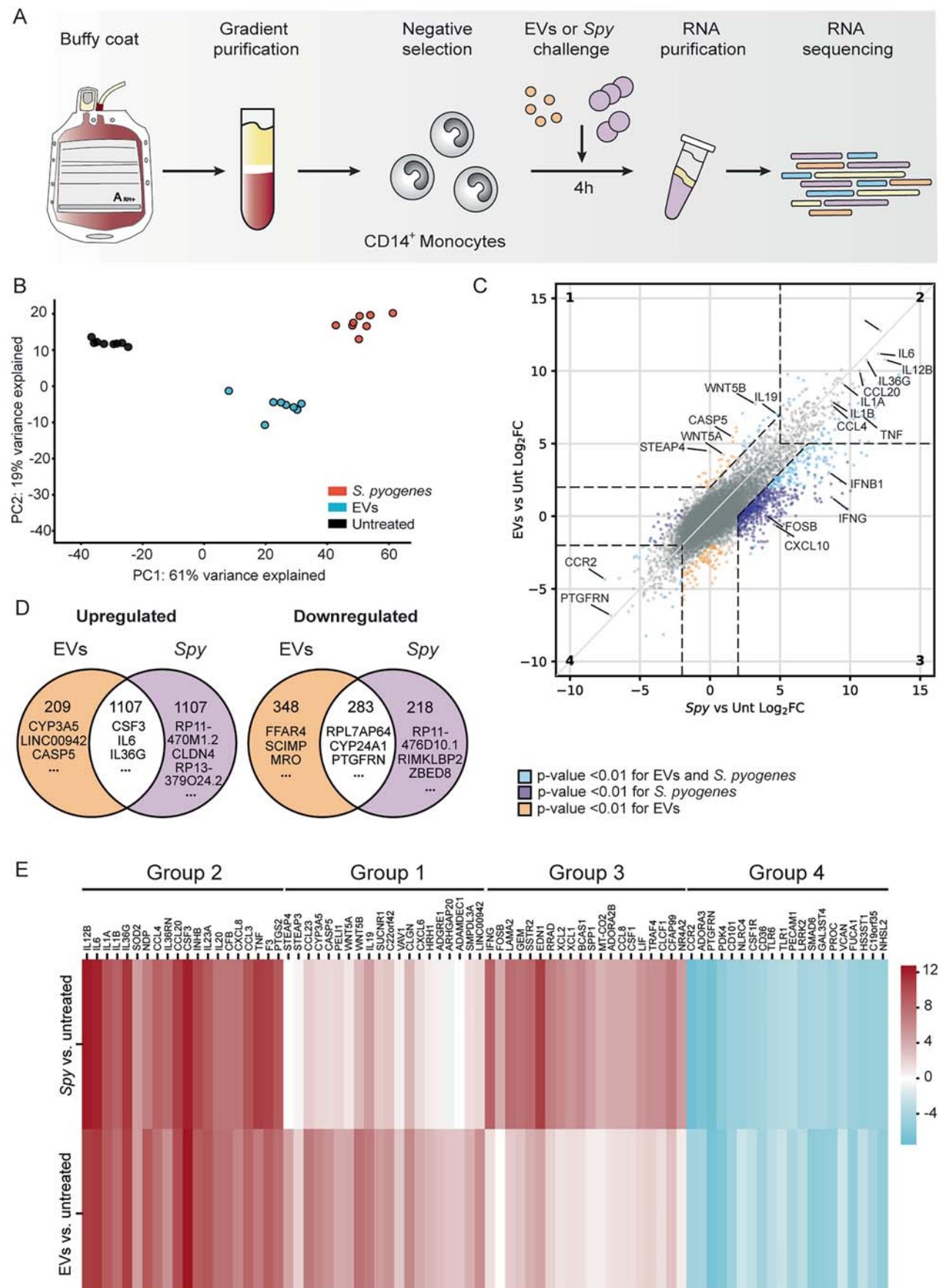

**Figure 1. *S. pyogenes* EVs induce a distinct expression profile in monocytes.**

(A) Schematic summary of the study design. Primary monocytes were purified through density gradient followed by magnetic negative selection. Four hours after *S. pyogenes* infection or challenge with its EVs, RNA from monocytes was purified and processed for RNA sequencing. (B) Principal component analysis of gene transcription colored by treatment. (C) Scatterplot comparison of Log$_2$ fold change (FC) (treatment vs untreated control, Unt) in transcript abundance for *S. pyogenes*-infected (*x* axis) or EV-treated samples (*y* axis). Dashed lines mark cut-off criteria of a log$_2$ FC > |2|. Grey dots represent transcripts commonly regulated but with an FC < |2|. Blue dots denote commonly regulated transcripts with a FC ≥ |2|. Orange and purple dots indicate differentially expressed genes for either EVs or *S. pyogenes*, respectively. Dashed areas indicate four distinct gene groups which meet our cut-off criteria that share (2 and 4) or display unique (1 and 3) transcription patterns for both or either stimulus, respectively. Representative genes in each group are highlighted. Wald test was applied for statistical analyses. (D) Venn diagrams summarizing unique and overlapping up- or downregulated genes with FC ≥ |2|. The top three genes for each category are displayed. (E) Heatmap of twenty representative genes from each gene group in (D). Data information: (B–E) Data are representative of eight biological replicates. (C–E) Graphs include gene transcription after 4 h upon EV or *S. pyogenes* stimulation compared to untreated monocytes. Genes were defined as differentially regulated when FC ≥ |2| as compared to untreated cells and the adjusted *P* value was <0.01 after Benjamini-Hochberg correction for multiple comparisons.

Our transcriptomic datasets showed that 1107 and 445 genes are upregulated in response to both *Spy* EVs and bacterial cells in monocytes and BLaER1 cells, respectively (Figs. 1C,D and EV2A,B). Since primary monocytes respond more strongly than BLaER cells, we focused on our monocyte dataset for further analyses. Accordingly, pathway analysis of primary monocytes revealed that KEGG pathways and GO terms enriched for each treatment show a high similarity (Fig. EV2C,D). Among the commonly upregulated genes (Fig. 1C–E, group 2), we found cytokines (e.g., *IL1A*, *IL1B* and *IL6*), chemokines (e.g., *CCL3*, *CCL4* and *CXCL8*) and growth factors (e.g., *CSF3*). In contrast, several immune receptors (e.g., *TLR1*, *TLR6* and *CCR2*) and adhesion factors (e.g., *PECAM1* and *VCAN*) were downregulated for both stimuli (Fig. 1C–E, group 4). We then analyzed the differentially expressed genes (DEGs) belonging specifically to one of the treatments. DEGs upon challenge with *Spy* EVs included the metalloreductase *STEAP4*, components of the Wnt signaling pathway (*WNT5A*), and, strikingly, the immune sensor *CASP5* (Fig. 1C–E, group 1). Infection with *S. pyogenes* specifically upregulated an additional 1107 genes (Fig. 1C–E, group 3), including interferon and interferon-related genes (IRGs, e.g., *IFNG*, *IFNB1* and *CXCL10*), growth factors (e.g., *CSF1*), and chemokines (e.g., *CCL8*), indicating that the sensing of whole bacteria induces responses remarkably distinct from those of EVs. Real-time quantitative PCR (RTqPCR) was used to validate representative candidates from each gene group (Fig. EV2E–H).

## *Spy* EVs induce a proinflammatory response

During *S. pyogenes* infection, inflammation facilitates bacterial clearance by innate immune cells but may also promote a potentially detrimental immune response. In mouse models, this balance is orchestrated by IL-1β and type I interferon signaling, which promote and repress inflammation, respectively (Castiglia et al, 2016). Accordingly, IL-1β inhibitors used to treat patients with autoimmune diseases increase the risk of developing invasive *S. pyogenes* infections by 330-fold, suggesting a central role for this cytokine in controlling *S. pyogenes* pathogenesis in humans (LaRock et al, 2016). Interestingly, while IL-1β and other members of the IL-1 cytokine family, such as IL-1α and IL-36γ, are upregulated by both *S. pyogenes* and its EVs, we observed expression of IFNβ, IFNγ, and IRGs (e.g., CXCL9, CXCL10, CXCL11) predominantly during bacterial infection (Dataset EV1; Figs. 1C,E and EV4G,H). Although RNA sequencing detected both IFNβ and -γ expression also in response to *Spy* EVs, they were only

classified among the lowest 5% or 10% of the DEGs. In contrast, infection with *S. pyogenes* triggered a significantly higher expression of both cytokines, with IFNβ ranked among the lowest 50% and IFNγ among the highest 50% of the DEGs, indicating that only bacterial cells induce a robust IFN response in human monocytes.

## Caspase-4/-5 are required for *Spy* EV-induced IL-1β release

Inflammasomes in innate immune and epithelial cells play an important role in bacterial sensing (Martinon et al, 2002; Hayward et al, 2018). In monocytes, IL-1β is secreted upon activation of the canonical, non-canonical and/or alternative inflammasome (Martinon et al, 2002; Kayagaki et al, 2011; Shi et al, 2014; Gaidt et al, 2016). Notably, we observed that *Spy* EVs treatment induced expression of the non-canonical inflammasome component, caspase-5, compared with untreated cells (log$_2$FC: 5.45, Dataset EV1; Figs. 1C,D and EV4E). Caspase-4 and -5 and their murine homolog, caspase-11, have been shown to recognize intracellular LPS during Gram-negative infection, and are also activated in response to the Gram-positive species *Staphylococcus aureus* (*S. aureus*) and *Listeria monocytogenes* (*L. monocytogenes*) (Shi et al, 2014; Casson et al, 2015; Vanaja et al, 2016; Hara et al, 2018; Krause et al, 2019). Upregulation of caspase-5 upon EV treatment in the absence of a robust interferon signature (Figs. 1 and EV4) prompted us to investigate whether IL-1β might be induced differently by *Spy* EVs than by their parental cells.

To determine the role of caspase-4 and -5 in *Spy* EV-induced IL-1β release, we used the chemical inhibitor Ac-LEVD-CHO, which blocks the action of these two caspases. Incubation of monocytes with the caspase-4/-5 inhibitor prior to the addition of *Spy* EVs or *S. pyogenes* decreased IL-1β secretion, albeit with a greater reduction in IL-1β in response to EVs (Fig. 2A,B). In contrast, the levels of caspase-independent cytokines such as IL-6 and IL-1α remained unchanged. Activation of the non-canonical inflammasome can lead to pyroptotic cell death (Shi et al, 2015). However, cell death measured by secretion of lactate dehydrogenase (LDH) did not vary upon inhibition of caspase-4/-5 (Fig. 2A,B). To validate that Ac-LEVD-CHO only impairs the function of caspase-4/-5 and does not affect the expression of its genes, we measured the mRNA levels of caspase-4 and -5 as well as *IL1B* and *IL6* using RTqPCR. No difference in the transcript levels for each target gene between untreated vs. Ac-LEVD-CHO-treated cells could be detected (Fig. EV3A), indicating that the observed reduction in IL-1β secretion upon inhibition of caspase-4/-5 does not originate

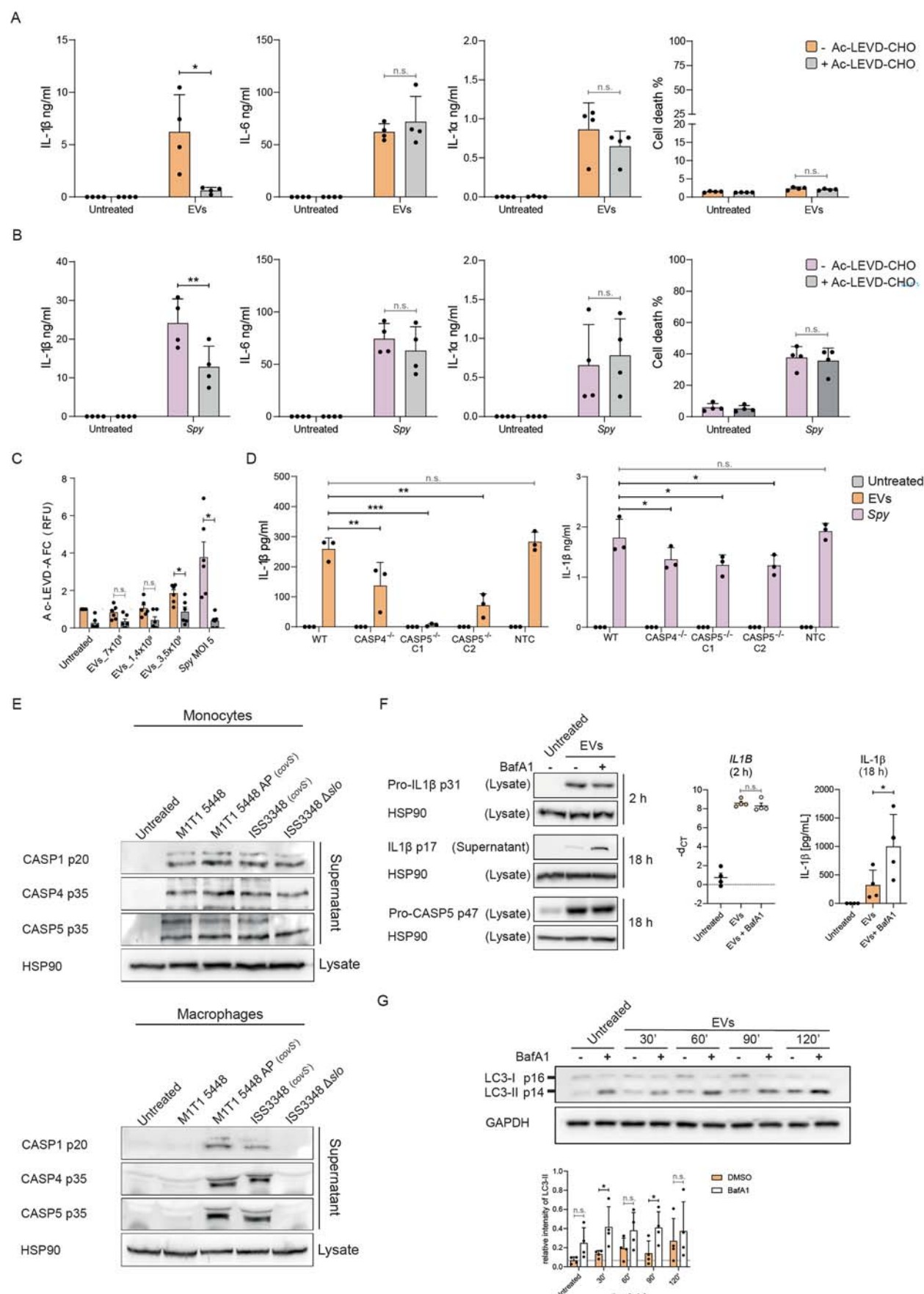

Figure 2. *S. pyogenes* EVs are recognized by caspase-4 and caspase-5.

(A, B) IL-1β, IL-6, IL-1α, and LDH released by monocytes after stimulation with either *Spy* EVs (A) or intact *S. pyogenes* (B) for 18 h. Monocytes were either preincubated with the caspase-4/-5 inhibitor Ac-LEVD-CHO or left untreated. The percentage of LDH released from the positive control is shown as a measure of cell death. Bars represent the mean ± standard deviation (SD) of four biological replicates. (A) $P = 0.0385$, (B) $P = 0.0036$, (C) Caspase-4/5 substrate cleavage in supernatants from monocytes after stimulation with *Spy* EVs or intact *S. pyogenes*. Bars represent the mean ± SD of six biological replicates. $P = 0.0139$ (EVs_3.5 × 10$^9$), $P = 0.0024$ (*Spy* MOI 5). (D) IL-1β released by BLaER1 cells after stimulation with either *Spy* EVs (left panel) or *S. pyogenes* (right panel) for 18 h. BLaER1 cell lines: wild-type (WT), Caspase-4 knock-out (KO, CASP4$^{-/-}$), Caspase-5 KO (CASP5$^{-/-}$, clones 1 and 2) and non-targeting control (NTC). Bars represent the mean ± SD of three biological replicates. (Left panel) $P = 0.0034$, $P < 0.0001$, $P = 0.003$. (Right panel) $P = 0.0239$, $P = 0.0131$, $P = 0.0131$. (E) Immunoblot analysis of cleavage products for caspase-1/4/5 in cell culture supernatants from human monocytes & macrophages. Cells were either left untreated or infected with *S. pyogenes* strains M1T1 5448, M1T1 5448 AP, ISS3348, or ISS3348 Δ*slo* at MOI 5 (*covS⁻*: *covS* inactivation). Representative of four biological replicates. (F) Immunoblot, RTqPCR & ELISA analysis of IL-1β released by monocytes after stimulation with *Spy* EVs in the presence or absence of BafA1 (100 nM) for 2 h or 18 h. A dotted line indicates the mean expression of the housekeeping genes (*GAPDH/TUBB*). Representative of four biological replicates. $P = 0.0254$. (G) Immunoblot analysis of LC3 levels in human monocytes after stimulation with *Spy* EVs in the presence or absence of BafA1 (100 nM). Bars represent the mean ± SD of four biological replicates. $P = 0.0132$ (30'), $P = 0.0151$ (90'). Data information: (ABDG) Two-way ANOVA with Holm–Šídák correction for multiple comparisons was applied for statistical analyses. (C) Multiple *t* tests with Holm–Šídák correction for multiple comparisons. (F) One-way ANOVA with Holm–Šídák correction for multiple comparisons was applied for statistical analyses. *$P ≤ 0.05$, **$P ≤ 0.01$, ***$P ≤ 0.001$, n.s. not significant. Source data are available online for this figure.

from lower caspase-5 or pro-IL1β gene expression. In addition, we analyzed the amounts of pro-IL1β by immunoblotting. While the induction of pro-IL1β in response to *Spy* EVs was much lower than for bacterial cells, no difference was found between untreated and Ac-LEVD-CHO-treated cells (Fig. EV3B). Consistent with the fact that LPS is a major activator of caspase-4/-5, Ac-LEVD-CHO also significantly reduced the release of IL-1beta in response to LPS (Fig. EV3C). Finally, we detected significant cleavage of the caspase-4/-5 substrate Ac-LEVD-AFC in cell culture supernatants from monocytes infected with *S. pyogenes* and, to a lesser extent, in response to *Spy* EVs as well (Fig. 2C). Collectively, our data suggest that IL-1β release in response to *Spy* EVs and *S. pyogenes* requires caspase-4/-5 without inducing cell death.

Human monocytes are short-lived, non-replicating cells (Patel et al, 2017), which complicates loss-of-function studies using engineering tools (e.g., CRISPR/Cas9). Furthermore, commonly used monocytic cell lines, such as THP-1, have been shown to only partially mimic the functions of primary cells (Gaidt et al, 2016, 2018). To overcome these limitations, we targeted genes in BLaER1 cells, which have been described to closely recapitulate human monocytic functions such as TLR and inflammasome signaling (Rapino et al, 2013; Gaidt et al, 2016). BLaER1 is an immortalized B cell line which, upon induction of the transcription factor C/EBPα, irreversibly differentiates into monocyte-like cells (Rapino et al, 2013). We have shown that BLaER1 cells respond similarly, albeit less pronounced, to *Spy* EVs and bacterial cells (Fig. EV2AB). To dissect the individual contributions of caspase-4 and -5 in *Spy* EV-induced IL-1β release, we generated gene knock-out (KO) BLaER1 cell clones using CRISPR-Cas9 (Fig. EV3D). BLaER1 CASP4$^{-/-}$ cells secreted less IL-1β than their wild-type counterparts upon *Spy* EVs challenge, while IL-1β secretion in CASP5$^{-/-}$ cells was almost abolished (Fig. 2D, left panel). A slight reduction in IL-1β production was also observed upon stimulation with live *S. pyogenes* (Fig. 2D, right panel). Consistent with our experiments in primary monocytes, IL-6 and LDH levels were unaltered in wild-type and CASP4$^{-/-}$ and CASP5$^{-/-}$ cells (Fig. EV3E–H). Non-targeted control (NTC, electroporated with a control guide RNA) cells showed similar secretion of IL-1β, IL-6 and LDH when compared to wild-type cells (Figs. 2D and EV3E–H). Immunoblot analysis of cell culture supernatants from human monocytes infected with different M1 serotypes of *S. pyogenes* revealed no difference in caspase-1/-4/-5 cleavage between wild-

type and *covS*-inactivated (*covS⁻*) strains (Fig. 2E). In contrast, when monocytes were differentiated to macrophages, cleavage products for all three caspases were only observed when cells were infected with hypervirulent *S. pyogenes* harboring *covS* inactivating mutations leading to increased production of streptolysin O (Slo) (Sumby et al, 2006). Altogether, our experiments suggest that caspase-4 and caspase-5 contribute to IL-1β release in response to *Spy* EVs and intact *S. pyogenes*, which is independent of the virulence regulator CovS in monocytes but dependent on CovS in differentiated macrophages.

During inflammasome activation, mature IL-1β is typically secreted through Gasdermin D (GSDMD) pores. However, *Spy* EV stimulation of human monocytes did not result in cell death. Alternative secretion pathways for IL-1β through secretory lysosomes (Andrei et al, 1999; Semino et al, 2018), microvesicle shedding (MacKenzie et al, 2001), or autophagy (Iula et al, 2018) have been described in LPS-activated monocytes or neutrophils. Consistent with these previous findings, we found that Bafilomycin A (BafA1), a V-ATPase inhibitor shown to stimulate lysosomal exocytosis (Tapper and Sundler, 1995), promotes an increase in EV-induced IL-1β release from human monocytes (Fig. 2F) without affecting pro-IL-1β or caspase-5 expression. Finally, we also observed an increase in LC3-II levels and autophagic flux in human monocytes upon EV treatment (Fig. 2G), suggesting an increase in the formation of autophagosomes.

## EV signaling through TLR2-MyD88 are required for cytokine production

Caspase-4/-5 are cytosolic receptors that do not bind directly to any component of Gram-positive lysates (Shi et al, 2014a). Consequently, signaling through other proteins acting upstream of caspase-4 and -5 may be the trigger for IL-1β production. TLRs are the most thoroughly characterized family of PRRs involved in *S. pyogenes* recognition (Gratz et al, 2008; Loof et al, 2008; Eigenbrod et al, 2015; Fieber et al, 2015). In particular, TLR2 signaling via the adaptor protein MyD88 is essential for generating an inflammatory response to *S. pyogenes* and other Gram-positive pathogens by promoting NFκB-dependent gene expression (Underhill et al, 1999; Loof et al, 2008). Yet, conflicting results have been published on the role of TLR2 in *S. pyogenes* recognition (Gratz et al, 2008; Fieber et al, 2015). Since LPS-induced activation of the alternative

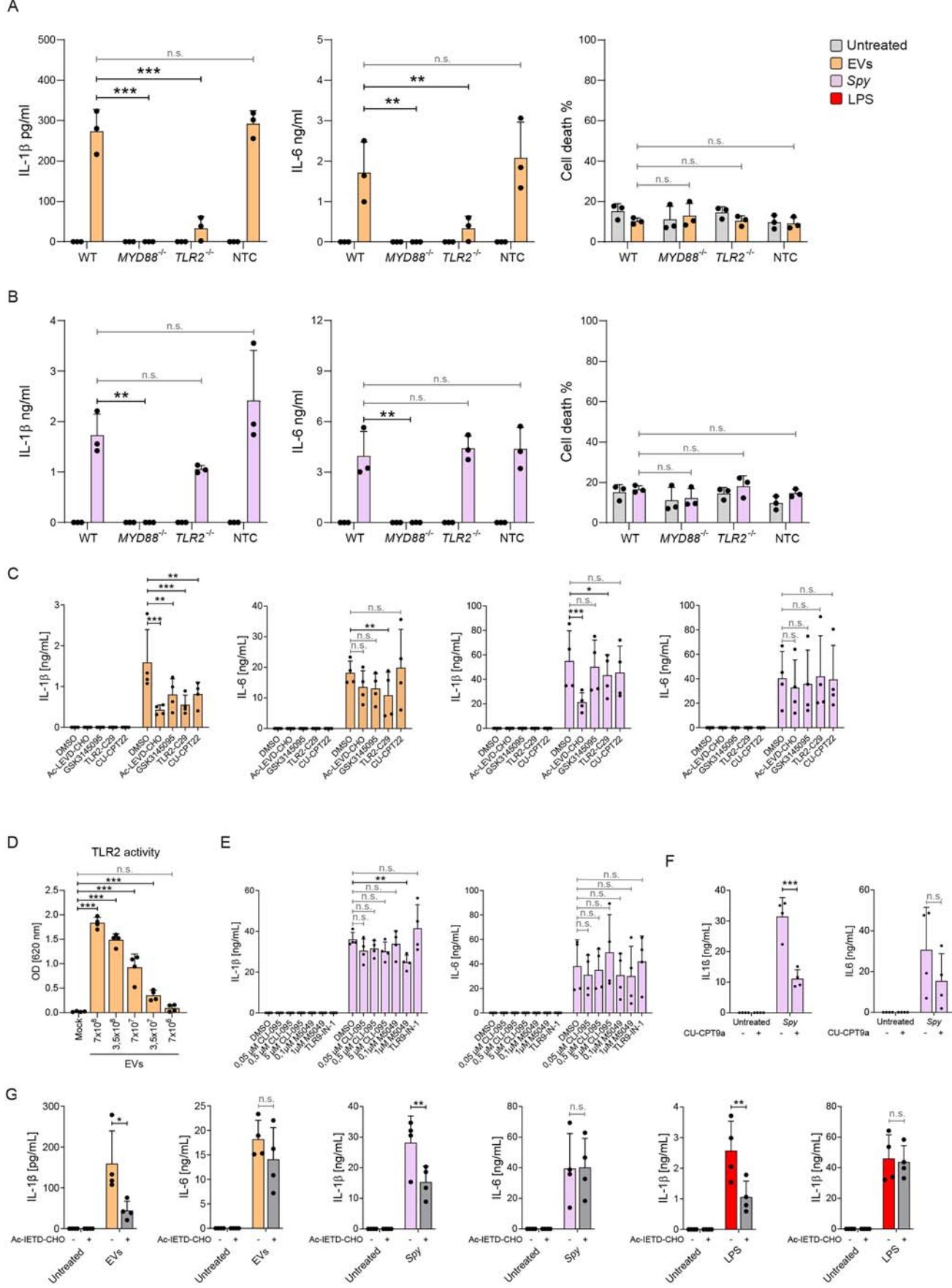

Figure 3. **EV-dependent cytokine release requires Toll-like receptor 2-mediated signaling.**

(A) IL-1β, IL-6, and LDH in supernatants of BLaER1 WT, *MYD88* KO (*MYD88*$^{-/-}$), Toll-like receptor 2 KO (*TLR2*$^{-/-}$) and NTC cells stimulated with *Spy* EVs for 18 h. Bars represent the mean ± SD of three biological replicates. (left panel, IL-1β) $P < 0.0001$ (*MYD88*$^{-/-}$), $P < 0.0001$ (*TLR2*$^{-/-}$). (middle panel, IL-6) $P = 0.0019$ (*MYD88*$^{-/-}$), $P = 0.0040$ (*TLR2*$^{-/-}$). (B) IL-1β, IL-6, and LDH released by BLaER1 WT, *MYD88*$^{-/-}$, *TLR2*$^{-/-}$, and NTC cells after 18 h stimulation with *S. pyogenes*. Bars represent the mean ± SD of three biological replicates. (left panel, IL-1β) $P = 0.0017$ (*MYD88*$^{-/-}$). (Middle panel, IL-6) $P = 0.0016$ (*MYD88*$^{-/-}$). (A, B) The percentage of LDH released from the positive control is shown as a measure of cell death. (C) IL-1β and IL-6 released by human monocytes that were either left untreated or preincubated with Ac-LEVD-CHO (caspase-4/5), GSK3145095 (RIPK1), TLR2-C29 (TLR2/1, TLR2/6), or CU-CPT22 (TLR2/6). Cells were then left unstimulated or challenged with *Spy* EVs or *S. pyogenes* for 18 h. Bars represent the mean ± SD of four biological replicates. (EVs, IL-1β) $P = 0.0001$, $P = 0.0018$, $P = 0.0003$, $P = 0.0018$. (EVs, IL-6) $P = 0.0082$. (*Spy*, IL-1β) $P < 0.0001$, $P = 0.0437$. (D) TLR2 activity in HEK-Blue TLR2 cells after stimulation with *Spy* EVs for 18 h. Baseline represents untreated cells. Bars represent the mean ± SD of four biological replicates. ($7 \times 10^8$) $P < 0.0001$, ($3.5 \times 10^8$) $P < 0.0001$, ($7 \times 10^7$) $P < 0.0001$, ($3.5 \times 10^7$) $P = 0.0081$. (E) IL-1β and IL-6 released by human monocytes that were either left untreated or preincubated with TLR inhibitors CLI-095 (TLR4), M5049 (TLR7/8, 0,1 μM = TLR7, 1 μM = TLR8), or TLR9-IN-1 (TLR9). Cells were then left unstimulated or infected with *S. pyogenes* for 18 h. Bars represent the mean ± SD of four biological replicates. $P = 0.0048$. (F) IL-1β and IL-6 released by human monocytes that were either left untreated or preincubated with the TLR8 inhibitor CU-CPT9a. Cells were then left unstimulated or infected with *S. pyogenes* for 18 h. Bars represent the mean ± SD of four biological replicates. $P < 0.0001$. (G) IL-1β and IL-6 released by human monocytes that were either left untreated or preincubated with the caspase-8 inhibitor Ac-IETD-CHO. Cells were then left unstimulated or challenged with *Spy* EVs, *S. pyogenes*, or LPS for 18 h. Bars represent the mean ± SD of four biological replicates. (EVs) $P = 0.0043$, (*Spy*) $P = 0.0075$, (LPS) $P = 0.0043$. Data information: (A–C, E–G) Two-way ANOVA was applied with Holm–Šídák correction for multiple comparisons. (D) One-way ANOVA was applied with Holm–Šídák correction for multiple comparisons. *$P \le 0.05$, **$P \le 0.01$, ***$P \le 0.001$, n.s. not significant. Source data are available online for this figure.

inflammasome in human monocytes has been linked to lipid A-induced TLR4 signaling, we investigated whether the lipid and lipoteichoic acid (LTA) sensing TLR2 and its adaptor protein MyD88 might coordinate the sensing of *Spy* EVs. Indeed, TLR2- or MyD88-deficiency in BLaER1 cells severely impaired both IL-1β and IL-6 secretion in response to *Spy* EVs challenge (Fig. 3A), implying a critical role for TLR2-MyD88 signaling in *Spy* EVs recognition. In contrast, sensing of *S. pyogenes* required MyD88, but not TLR2, to induce IL-6 and IL-1β (Fig. 3B), indicating a redundant role for other MyD88-coupled TLRs such as TLR1 or TLR4-9. To address this redundancy, we used a variety of small molecule inhibitors. Treatment of monocytes with TLR2/1 (TLR-IN-C29) or TLR2/6 (Cu-CPT22) inhibitors significantly reduced the release of *Spy* EV-induced IL-1β, but only minor effects were observed following infection with *S. pyogenes* (Fig. 3C). Notably, inhibition of the Receptor-Interacting serine/threonine-Protein Kinase 1 (RIPK1), the key kinase mediating the alternative inflammasome pathway upon LPS treatment, resulted in a reduction in IL-1β secretion only in *Spy* EV-treated cells, but not in those treated with *S. pyogenes*. In support of TLR2 appearing as the major sensor of *Spy* EVs, a TLR2 reporter cell-line (HEK-Blue) showed a dose-dependent activity in response to *Spy* EVs (Fig. 3D). We further examined the potential involvement of TLR4, TLR7, TLR8, and TLR9 in the sensing of intact *S. pyogenes* cells by means of small molecule inhibitor treatment. We found a role for intracellular TLR8 in mediating IL-1β secretion upon live *S. pyogenes* infection (Fig. 3E,F). Lastly, and in accordance with previous observations for LPS-stimulated monocytes (Gaidt et al, 2016), both *S. pyogenes* and its EVs required caspase-8 for IL-1β release (Fig. 3G). Together, we conclude that the activation of the inflammasome in response to *Spy* EVs is most likely initiated at the cell surface via TLR2, whereas *S. pyogenes* triggers IL-1β secretion predominantly through TLR8, thereby indicating that internalization of the bacteria is required to activate the inflammasome.

## The NLRP3 inflammasome coordinates EV-mediated secretion of IL-1β

The NLRP3 inflammasome is composed of three main proteins: Nucleotide-binding domain Leucine-rich Repeat family containing a Pyrin domain 3 (NLRP3), Apoptosis-associated Speck-like containing a Caspase recruitment domain (ASC), and caspase-1 (Martinon et al, 2002; Muñoz-Planillo et al, 2013). Previous studies have shown that the NLRP3 inflammasome coordinates IL-1β secretion during *S. pyogenes* infection (Harder et al, 2009; Lin et al, 2015; Valderrama et al, 2017) and a recent study established that *S. aureus* EVs are also recognized by this system (Wang et al, 2020). We therefore investigated whether, in addition to caspase-4/-5, the NLRP3 inflammasome contributed to IL-1β release in response to *Spy* EVs. To this end, we left monocytes untreated or treated them with the NLRP3 inhibitor MCC950, the caspase-1 inhibitor Ac-YVAD-cmk, the caspase-4/-5 inhibitor Ac-LEVD-CHO, or the caspase-1/-4/-5 inhibitor VX-765 prior to stimulation with *S. pyogenes* or *Spy* EVs. All inhibitors significantly reduced IL-1β for both *S. pyogenes* and its EVs (Fig. 4A,B). Release of IL-6 and LDH were comparable for all treatments, with the exception of increased IL-6 and decreased cell death upon inhibition of NLRP3 following infection with *S. pyogenes* (Figs. 4A,B and EV4A).

To further validate these results, we used BLaER1 monocytes lacking ASC (Vierbuchen et al, 2017), caspase-1, or NLRP3. Consistent with the pharmacological inhibition, the genetic deficiency of *NLRP3*, *ASC*, or *CASP1* abolished IL-1β secretion in response to *S. pyogenes* or *Spy* EVs treatment (Fig. 4C–F), while IL-6 levels remained unchanged for all cell lines and conditions tested (Fig. EV4B,C). In addition, whereas the release of LPS-induced IL-1β and LDH was strongly dependent on K$^+$ efflux, no effect of extracellular potassium could be detected for *Spy* EVs (Fig. 4G). In contrast, *S. pyogenes*-infected monocytes showed significantly reduced IL-1β secretion in the presence of 50 mM KCl, indicating that the reduction of intracellular K$^+$ contributes to activation of the inflammasome triggered by *S. pyogenes*, but not by its derived EVs. Using immunofluorescence microscopy, we further analyzed ASC speck formation in human monocytes either treated with *Spy* EVs or infected with *S. pyogenes*. In accordance with the previously reported alternative inflammasome pathway, which lacks the typical features of classical NLRP3 inflammasome activation (Gaidt et al, 2016), we observed ASC specks and cleavage of GSDMD only in *S. pyogenes*-infected monocytes, but not in EV-treated cells (Figs. 4H and EV4E).

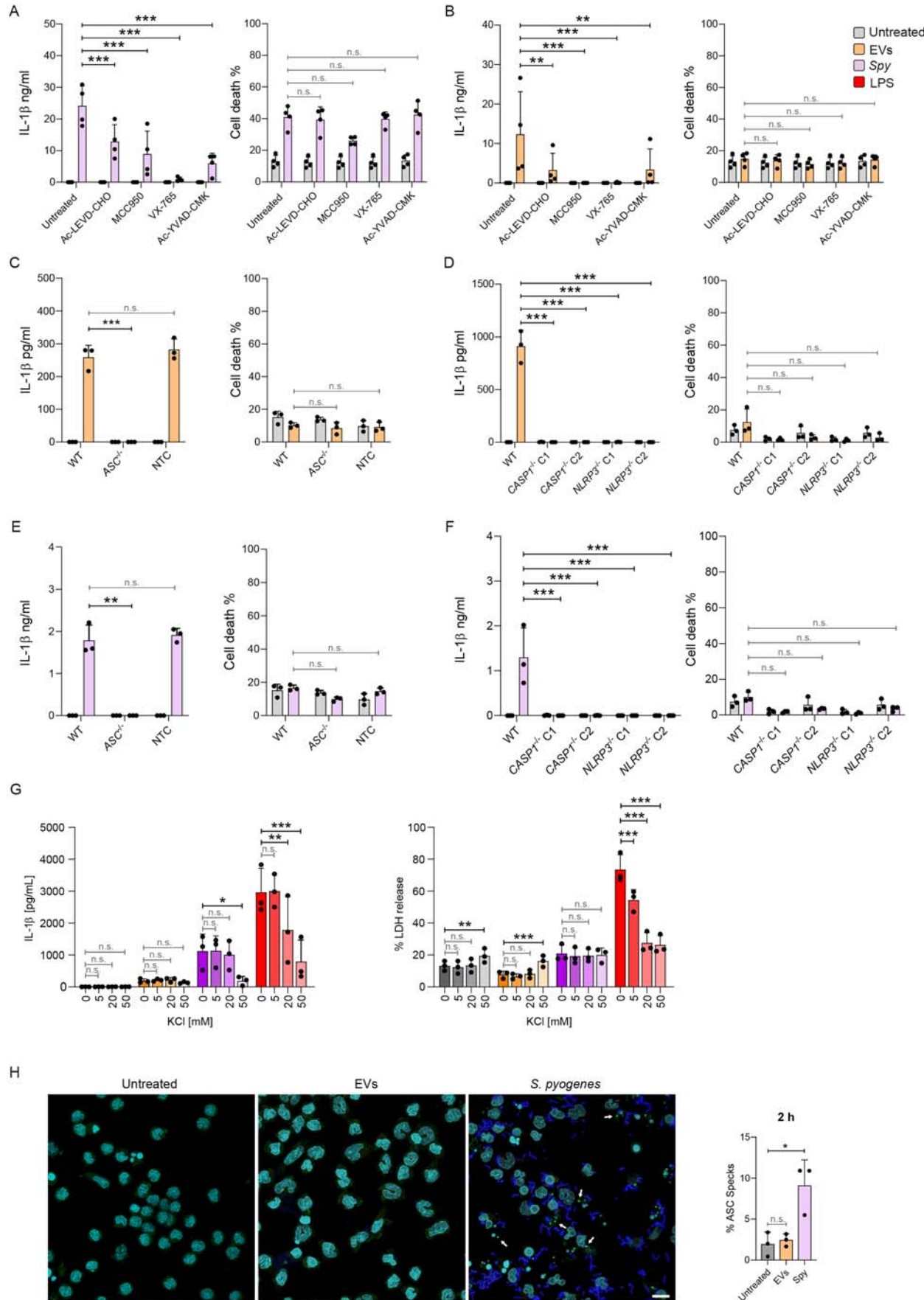

**Figure 4. The NLRP3 inflammasome contributes to IL-1β and LDH release upon recognition of *S. pyogenes* EVs.**

(A, B) IL-1β and IL-6 released by human monocytes that were either left untreated or preincubated with Ac-LEVD-CHO, MCC950, VX-765, or Ac-YVAD-CMK. Cells were then left unstimulated or stimulated with either *S. pyogenes* (A) or *Spy* EVs (B) for 18 h. Bars represent the mean ± SD of four biological replicates. (A) $P < 0.0001$ (for all comparisons). (B) $P = 0.0068$, $P = 0.0006$, $P = 0.0006$, $P = 0.0076$. (C, D) IL-1β and LDH were measured in supernatants of BLaER1 WT, PYCARD KO (ASC$^{-/-}$), Caspase-1 KO (*CASP1$^{-/-}$* clone 1 and 2), NLRP3 KO (*NLRP3$^{-/-}$* clone 1 and 2), and NTC cells after 18 h either left untreated or stimulated with *Spy* EVs. Bars represent the mean ± SD of three biological replicates. (C) $P = 0.0010$. (D) $P < 0.0001$ (for all comparisons). (E, F) IL-1β and LDH were measured in supernatants of BLaER1 WT, ASC$^{-/-}$, CASP1$^{-/-}$ clone 1 and 2, *NLRP3$^{-/-}$* clone 1 and 2, and NTC cells after 18 h either left untreated or stimulated with *S. pyogenes*. Bars represent the mean ± SD of three biological replicates. (E) $P = 0.0016$. (F) $P = 0.0003$ (for all comparisons). (G) IL-1β and LDH in supernatants from BLaER1 WT cells after 18 h, either left untreated or stimulated with *Spy* EVs, *S. pyogenes*, or LPS (200 ng/mL). Bars represent the mean ± SD of three biological replicates. (IL-1β) $P = 0.0177$ (*Spy*, 50 mM), $P = 0.0017$ (LPS, 20 mM), $P < 0.0001$ (LPS, 50 mM). (LDH) $P = 0.0084$ (Untreated, 50 mM), $P = 0.0007$ (EVs, 50 mM), $P < 0.0001$ (LPS, all comparisons). (H) Immunofluorescence staining and corresponding quantification results for ASC speck formation in human monocytes either left untreated or stimulated with *Spy* EVs or *S. pyogenes* for 2 h (scale bar: 10 μm). White arrows indicate ASC specks. Bars represent the mean ± SD of three biological replicates. $P = 0.0103$. Data information: (A–G) The percentage of LDH released from the positive control is shown as a measure of cell death. Two-way ANOVA was applied with Holm–Šídák correction for multiple comparisons. (H) One-way ANOVA was applied with Holm–Šídák correction for multiple comparisons. *$P ≤ 0.05$, **$P ≤ 0.01$, ***$P ≤ 0.001$, n.s. not significant. Source data are available online for this figure.

Since NLRP6 and NLRP7 have been reported to recognize LTA and acetylated lipoproteins (Khare et al, 2012; Hara et al, 2018), we also attempted to generate corresponding knockout cells for these proteins to study their involvement in inflammasome activation upon *Spy* EVs or *S. pyogenes* challenge. However, we were unable to detect expression of either sensor in BLaER1 cells or in primary monocytes (Fig. EV4D). Collectively, these results indicate that *Spy* EVs or *S. pyogenes* engage the NLRP3 inflammasome as well as caspase-4 and -5, but employ distinct upstream sensors and signaling cascades for the induction and release of IL-1β in monocytes.

## Caspase-4/-5-Mediated Secretion of IL-1β is Triggered by Lipoteichoic Acid

LTA is a major component of the cell membrane of Gram-positive bacteria that is sensed by TLR2 (Schwandner et al, 1999; Takeuchi et al, 1999) and has been shown to induce caspase-11-mediated secretion of IL-1β in mice (Hara et al, 2018). Since we found that *Spy* EVs induced secretion of IL-1β depends on TLR2, we hypothesized that LTA might be the bacterial ligand triggering caspase-4/-5 signaling. The clinical isolate used in this study is highly encapsulated due to its inactive two-component system CovRS (Resch et al, 2016). Therefore, we first examined whether LTA was available on the surface of this bacterial strain and its EVs. Using immunogold-labelling of LTA and electron microscopy, we observed that this structural acid can indeed be detected on the surface of *S. pyogenes* and its EVs (Fig. 5A,B, white arrows). In addition, we used immunoblotting to confirm the presence of LTA in our EV preparations (Fig. 5C).

LTA mutants of Gram-positive bacteria have been reported to display severe growth defects (Schneewind and Missiakas, 2017), which is why we used purified LTA from *S. pyogenes* to test its role in monocyte activation. Consistent with our hypothesis, stimulation of monocytes or BLaER1 cells with *S. pyogenes* LTA induced IL-1β secretion, which was significantly reduced upon pretreatment with the caspase-4/-5 inhibitor peptide Ac-LEVD-CHO (Fig. 5D,E), while IL6 secretion and LDH release were not affected (Fig. EV5A,B). Moreover, the lipofectamine-mediated encapsulation and intracellular delivery of LTA into monocytes did not further increase IL-1β production (Fig. 5F) or affect IL-6 and LDH release (Fig. EV5C), suggesting that the caspase-4/-5-mediated signaling triggered by LTA most likely begins at the cell surface via

TLR2, in line with the alternative inflammasome pathway described for various TLR ligands (Gaidt et al, 2016; Unterberger et al, 2023). Finally, to test whether LTA from other Gram-positive bacteria similarly induces caspase-4/5-dependent IL-1β secretion, we used LTA derived from *S. aureus* and, akin to *S. pyogenes* LTA, we observed a caspase-4/-5-dependent reduction in the release of IL-1β, but not IL-6 (Fig. EV5D). Of note, we also detected a significant upregulation of caspase-5 gene expression upon stimulation with both *S. pyogenes* and *S. aureus* LTA, whereas gene expression levels of caspase-4 remained unaffected (Fig. EV5E). In line with our previous results showing *Spy* EV-induced TLR2 activity in a reporter cell-line, LTA from both *S. pyogenes* and *S. aureus* induced TLR2 activity in a dose-dependent manner, albeit *S. aureus* LTA was considerably more potent and showed a measurable response at concentrations as low as 0.01 μg/mL (Figs. 5G and EV5F). As shown before for Spy EVs, LTA-induced IL-1β secretion is dependent on caspase-8 (Fig. EV5G). Furthermore, BafA1 increased IL-1β release in response to *S. pyogenes* LTA, suggesting a vesicular secretion mechanism for IL-1β as observed for *Spy* EVs (Fig. 5H). Since the turnover of autophagosomes was also significantly increased by *S. pyogenes* LTA (Fig. 5I), a positive role of autophagy during IL-1β secretion in human monocytes upon *Spy* EV or LTA encounter may be a plausible conclusion.

## EVs from other Gram-positive bacteria induce IL-1β through caspase-4/-5

Since the synthesis of teichoic acids is moderately conserved among Gram-positive bacteria (Reichmann and Gründling, 2011; Brown et al, 2013), we asked whether EVs derived from other Gram-positive bacteria could trigger IL-1β production through caspase-4 and -5. In fact, monocytes treated with EVs from *S. aureus*, *Streptococcus agalactiae*, or *Bacillus subtilis* released less IL-1β when caspase-4/-5 was inhibited, while IL-6 secretion remained unaffected (Figs. 5J and EV5H,I), demonstrating that caspase-4/-5 are also involved in IL-1β release in response to EVs derived from other Gram-positive bacteria.

## LTA-depleted EVs severely attenuate the IL-1β response of human monocytes

To determine whether surface ligands other than LTA are sensed upstream of caspase-4/-5 activation, we treated *S. pyogenes* with

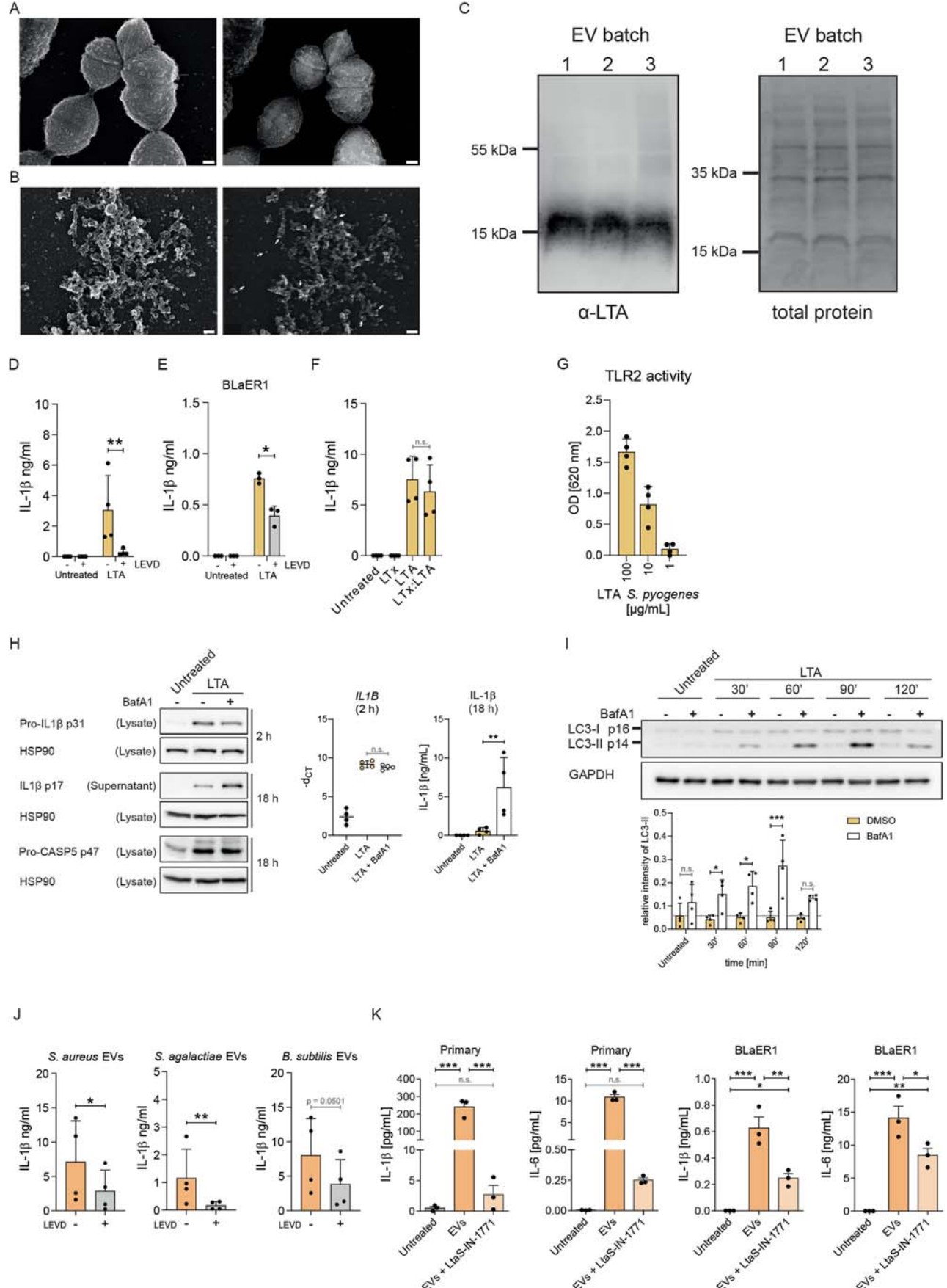

**Figure 5. Lipoteichoic acid conserved in Gram-positive EVs promotes caspase-4/-5-dependent IL-1β secretion.**

(A) *S. pyogenes* cells imaged by Scanning Electron Microscopy and immunogold staining (SEM, scale bar = 200 nm). (B) Immunogold-labelling of surface lipoteichoic acid from purified *Spy* EVs (LTA clusters indicated with white arrows, scale bar = 200 nm). (C) Immunoblot analysis of LTA in *Spy* EVs. (D) IL-1β in supernatants from monocytes either left untreated or preincubated with Ac-LEVD-CHO prior to the addition of *S. pyogenes* LTA after 18 h. Bars represent the mean ± SD of four biological replicates. *P* = 0.0181. (E) IL-1β in supernatants from BLaER1 cells either left untreated or preincubated with Ac-LEVD-CHO prior to the addition of *S. pyogenes* LTA after 18 h. Bars represent the mean ± SD of three biological replicates. *P* = 0.0177. (F) IL-1β release at 18 h from monocytes that were left untreated or treated with *S. pyogenes* LTA, either directly applied (LTA) or transfected using Lipofectamine LTX (LTx:LTA). Bars represent the mean ± SD of four biological replicates. (G) TLR2 activity in HEK-Blue TLR2 cells after stimulation with *Spy* LTA for 18 h. Baseline represents untreated cells. Bars represent the mean ± SD of four biological replicates. (H) Immunoblot, RTqPCR & ELISA analysis of IL-1β released by monocytes after stimulation with *S. pyogenes* LTA in the presence or absence of BafA1 (100 nM) for 2 h or 18 h. Bars represent the mean ± SD of four biological replicates. *P* = 0.0067. (I) Immunoblot analysis of LC3 levels in human monocytes after stimulation with *S. pyogenes* LTA in the presence or absence of BafA1 (100 nM). Bars represent the mean ± SD of four biological replicates. *P* = 0.0412 (30'), *P* = 0.0143 (60'), *P* = 0.0002 (90'). (J) IL-1β release at 18 h from monocytes either left untreated or preincubated with Ac-LEVD-CHO before the addition of EVs from *S. aureus* (left), *S. agalactiae* (middle), or *B. subtilis* (right). Bars represent the mean ± SD of four biological replicates. *P* = 0.0164 (*S. aureus*), *P* = 0.0099 (*S. agalactiae*). (K) IL-1β and IL-6 in supernatants from monocytes or BLaER1 cells either left untreated or stimulated with *Spy* EVs isolated −/+ LtaS-IN-1771 for 18 h. Bars represent the mean ± SD of three biological replicates. (Primary, IL-1β) *P* = 0.0003 (both comparisons). (Primary, IL-6) *P* < 0.0001 (both comparisons). (BLaER1, IL-1β) *P* = 0.0005, *P* = 0.0160, *P* = 0.0044. (BLaER1, IL-6) *P* = 0.0005, *P* = 0.0051, *P* = 0.0161. Data information: (D, E, I) Two-way ANOVA was applied with Holm–Šídák correction for multiple comparisons. (F, H, K) One-way ANOVA was applied with Holm–Šídák correction for multiple comparisons. (J) Statistical significance was assessed using paired *t* tests. \**P* ≤ 0.05, \*\**P* ≤ 0.01, \*\*\**P* ≤ 0.001, n.s. not significant. Source data are available online for this figure.

compound 1771, which was shown to inhibit LTA biosynthesis in *S. aureus* (Richter et al, 2013; Serpi et al, 2023) to produce LTA-depleted *Spy* EVs (Fig. EV5J). Consistent with its reported antimicrobial properties, compound 1771 significantly delayed the growth of *S. pyogenes* compared with DMSO-treated control cultures (Fig. EV5K). When monocytes were treated with LTA-depleted *Spy* EVs, their release of both IL-1β and IL-6 was significantly decreased (Fig. 5K). To a lesser extent, this phenotype was also reproduced in BLaER1 cells, indicating that LTA on Spy EVs is the predominant factor responsible for TLR and inflammasome activation.

## Free LTA in EV-depleted supernatants recapitulates monocyte activation in response to EVs

In addition to EV-bound LTA, supernatants of *S. pyogenes* also contain free LTA that is released during bacterial growth. To compare the effects of EV-bound vs. free LTA on monocyte activation, we first measured the amount of LTA in double-filtered *S. pyogenes* supernatants before and after EV isolation (Fig. 6A). The LTA concentration in the remaining supernatants was reduced by ~45% after removal of EVs using ultracentrifugation. We then determined the amounts of cytokines and LDH released by human monocytes treated either with *Spy* EVs or with EV-depleted supernatant (SUP) (Fig. 6B). Notably, due to the presence of Slo in the EV-depleted SUPs from WT *S. pyogenes*, monocytes displayed high amounts of cell death and no significant release of IL1β or IL-6 could be observed. Therefore, we additionally isolated EVs and EV-depleted SUP from an *S. pyogenes slo* deletion strain. Increasing volumes of Δ*slo* EV-depleted SUP significantly induced IL-1β and IL-6 secretion without affecting cell survival. Immunoblotting confirmed comparable levels of LTA in EV-depleted supernatants from WT and Δ*slo S. pyogenes* (Fig. 6C). Interestingly, THB alone also led to significant IL-6 secretion by human monocytes, which is most likely attributable to the presence of beef heart peptones or high salt concentrations within the concentrated media. Furthermore, protein analysis using mass spectrometry revealed an 84% overlap between EVs and EV-depleted supernatants, suggesting that a high percentage of proteins encapsulated in EVs is also released

into the media (Fig. 6D). Finally, as for monocytes treated with *Spy* EVs, stimulation with EV-depleted SUP prevented caspase-3 activation and induced phosphorylation of the MAP kinase (MAPK) p44/42 (ERK1/2) (Fig. 6D) indicative of monocyte survival pathways being activated.

## Spy EVs isolated using size exclusion chromatography (SEC) confirm that LTA is as driving factor for monocyte activation

Given the growing interest in EVs, isolation methodologies are constantly evolving and refined (Welsh et al, 2024). Since our EV preparations using ultracentrifugation most likely also contain non-vesicular extracellular particles (NVEP) such as pili, phages, or proteins, we attempted to isolate *Spy* EVs via size exclusion chromatography (SEC) as previously described for *E. coli* and other bacteria (Watson et al, 2021). In parallel to SEC, we processed the culture supernatants of *S. pyogenes* by ultracentrifugation to directly compare the EVs obtained. Figure 6E shows the LTA and protein concentration in sequential fractions, which increase successively. Using SDS-PAGE and TRPS, we identified *Spy* EVs enriched within the fractions 8-16 of SEC1, 8-15 of SEC2, and 8-13 of SEC3 (Appendix Fig. S2A). Characteristic blockage traces and size/concentration histograms of EVs UC and the pooled SEC fractions are summarized in Appendix Fig. S2B. Traces from individual SEC fractions are shown in Appendix Fig. S2C. Proteomic characterization revealed a 65% and 80% overlap between EVs derived from ultracentrifugation (EV UC) and SEC fractions 8-16 (EVs) and 17–28 (soluble), respectively. Since NTA analysis was not sensitive enough to detect particles in our SEC fractions, we used TRPS to determine the number, size, and zeta potential of our comparative UC and SEC-isolated EVs (Fig. 6G). We found negligible evidence of EV-aggregation after prolonged storage in PBS, supported by a unimodal size distribution (Appendix Fig. S3A). While the size of UC- and SEC-derived *Spy* EVs was comparable, the concentration of SEC EVs was one order of magnitude lower than that of the EV UC batches ($10^{10}$ vs. $10^{11}$ particles/mL). The zeta-potential of EVs in PBS ranged from −16 to −24 mV (UC mean: −22 ± 2.0, SEC mean: −21 ± 3.3), which is commonly observed for EVs and indicates the presence of

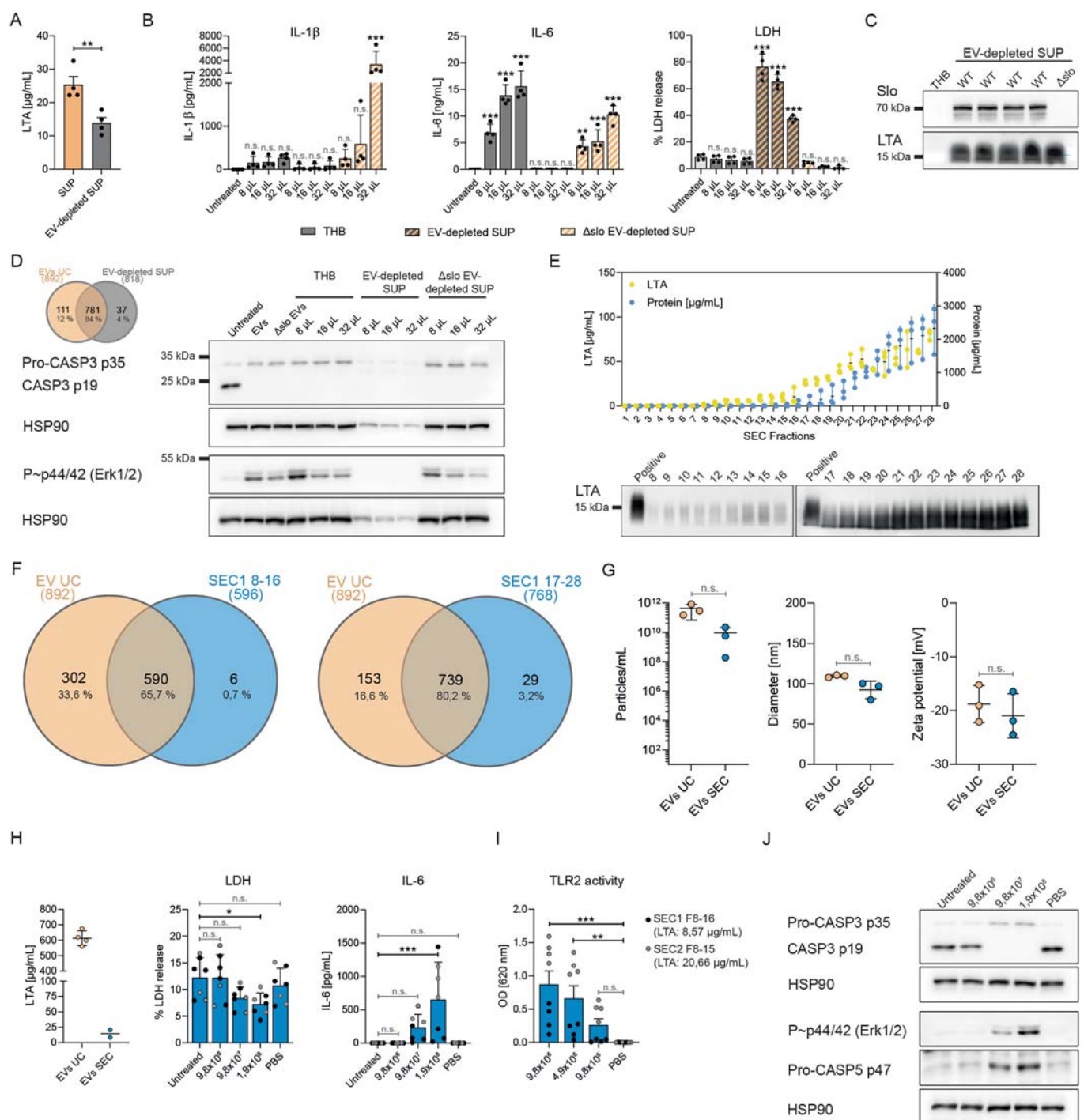

negatively charged cell-wall teichoic acid and membrane-anchored lipoteichoic acid (Rogers et al, 2023). Stimulation of human monocytes with SEC EVs significantly increased cell viability and IL-6 secretion (Fig. 6H). In addition, SEC EVs elicited pronounced TLR2 activity in HEK-Blue TLR2 reporter cells (Fig. 6I). In line with the lower particle and LTA concentration of SEC EVs compared to EVs UC (0.171 and 0.413 µg LTA per $1.9 \times 10^8$ particles for SEC1 and SEC2, respectively), the release of IL-1β was barely measurable (Appendix Fig. S3B). In fact, a dose response

experiment with *Spy* LTA revealed that the minimal concentration of LTA required to achieve measurable IL-1β secretion was 0,5 µg/ mL (Appendix Fig. S3C). However, we also detected inhibition of caspase-3 activation as well as the induction of p44/42 (ERK1/2) MAPK phosphorylation and caspase-5 protein expression upon SEC EV treatment. Conclusively, the SEC EV data corroborate the results of EV UC, with LTA being a major driver of monocyte activation and the subsequent inflammation in response to *Spy* EVs.

◄ **Figure 6.** *Spy* EVs isolated using size exclusion chromatography partially recapitulate the monocyte response against conventional EV preparations.

(A) LTA concentration in double-filtered *S. pyogenes* culture supernatants before and after ultracentrifugation. Bars represent the mean ± SD of four biological replicates. $P = 0.0080$. (B) IL-1β, IL-6, and LDH released by monocytes after 18 h stimulation with either *Spy* EVs, THB medium, or EV-depleted supernatants (SUP) derived from either *S. pyogenes* WT or Δ*slo* mutant strains. The percentage of LDH released from the positive control is shown as a measure of cell death. Bars represent the mean ± standard deviation (SD) of four biological replicates. (IL-1β) $P < 0.0001$. (IL-6) $P < 0.0001$ (THB, all comparisons), $P = 0.0020$, $P = 0.0002$, $P < 0.0001$. (LDH) $P < 0.0001$ (all comparisons). (C) Immunoblot of Slo and LTA in EV-depleted supernatants from *S. pyogenes* WT (4 biological replicates) or Δ*slo* mutant strains. THB media was used as negative control. (D) Venn diagram & immunoblots for EV-depleted supernatants. The Venn diagram shows protein expression in ultracentrifuged EV preparations (EV UC) vs. EV-depleted supernatants. Numbers indicate the total amount of proteins for each category. Immunoblots display caspase-3 cleavage and phospho-p44/42 MAPK in human monocytes after 18 h stimulation with either *Spy* EVs, THB medium, or EV-depleted supernatants (SUP) derived from either *S. pyogenes* WT or Δ*slo* mutant strains. Representative of four biological replicates. (E) LTA and protein concentration in 28 fractions of double-filtered *S. pyogenes* culture supernatant fractionated using size exclusion chromatography. Bars represent the mean ± standard deviation (SD) of three biological replicates. Immunoblot shows LTA present in fractions 8–28. (F) Venn diagrams showing protein expression in ultracentrifuged EV preparations (EV UC) vs. pooled SEC fractions 8–16 (EVs) or 17–28 (soluble), respectively. (G) Concentration, size, and zeta potential of *Spy* EVs analyzed with the exoid instrument. Bars represent the mean ± SD of three biological replicates. (H) LTA concentration in EVs UC vs. EVs SEC (left panel). Bars represent the mean ± SD of four vs. two biological replicates. LDH (middle panel) and IL-6 (right panel) released by human monocytes treated with *Spy* EVs isolated using SEC for 18 h. Bars represent the mean ± SD of seven biological replicates with pooled EV fractions from 2 SECs (black circles: SEC1 F8-16; grey circles: SEC2 F8-15). $P = 0.0299$ (LDH), $P = 0.0003$ (IL-6). (I) TLR2 activity in HEK-Blue TLR2 cells after stimulation with *Spy* EVs isolated using SEC for 18 h. Baseline represents untreated cells. Bars represent the mean ± SD of eight biological replicates with pooled EV fractions from 2 SECs (black circles: SEC1 F8-16; grey circles: SEC2 F8-15). $P = 0.0007$, $P = 0.0071$. (J) Immunoblots displaying caspase-3 cleavage, phospho-p44/42 MAPK, and caspase-5 expression in human monocytes after 18 h stimulation with *Spy* EVs isolated using SEC. Representative of four biological replicates. Data information: (A, G, H) Statistical significance was assessed using unpaired *t* test. (B, H) Owo-way ANOVA was applied with Holm–Šídák correction for multiple comparisons. *$P ≤ 0.05$, **$P ≤ 0.01$, ***$P ≤ 0.001$, n.s. not significant. Source data are available online for this figure.

# Discussion

Murine infection models have been used primarily to understand the role of specific receptors and downstream signaling mediators recognizing the strict human pathogen *S. pyogenes* (Watson et al, 2016). Although these models are extremely informative and enable the study of systemic infections and therapeutic interventions, it remains challenging to assess their contributions due to the intrinsic differences between the innate immune system of mice and humans (Heil et al, 2004; Hasan et al, 2005; Khare et al, 2012; Gaidt et al, 2016).

In the present study, we explored the response of primary human monocytes as well as monocyte-like BLaER1 cells to hypervirulent *S. pyogenes* and its EVs and identified mediators of recognition and downstream signaling events. Strikingly, only one report using murine macrophages has so far addressed the cellular response during *S. pyogenes* infection on a genome-wide level (Goldmann et al, 2007), and to our knowledge, no reports on primary human monocytes or macrophages have been published to date. Our RNA sequencing results revealed that responses triggered by *S. pyogenes* and its EVs strongly overlap, consistent with the observation that *Spy* EVs contain at least half of the proteins of the total proteome of their parental cells (Resch et al, 2016). Nonetheless, *S. pyogenes* and its EVs regulate the expression of distinct sets of genes in monocytes, indicating that unique components of intact bacteria or EVs convey a differential immune response.

In the broad spectrum of *S. pyogenes* clinical manifestations, inflammation plays a critical role in preventing bacterial invasion and limiting an excessive, devastating tissue-destructive immune response. It is well-established that interferons expressed during viral infections are beneficial to the host (Isaacs and Lindenmann, 1957). Notably, interferons produced during bacterial infections have been shown to play both protective and detrimental roles, depending on the bacterial species and the site of infection (Kovarik et al, 2016). In mice, the balance between protection and damage during *S. pyogenes* infection is orchestrated by IL-1β levels, a key immunoregulatory and proinflammatory cytokine matured by type

I interferon-regulated multiprotein complexes, the inflammasomes (Castiglia et al, 2016). Similarly, studies with IL-1β inhibitors suggest a central role of IL-1β in controlling *S. pyogenes* dissemination in the human body (LaRock et al, 2016).

On one side of this balance, *S. pyogenes* triggers an interferon-mediated signature that is distinct from its EVs, as demonstrated here by unbiased RNA-Seq profiling. These results are consistent with a report showing that biopsies from patients with necrotizing streptococcal soft tissue infection (NSTI) displayed a recognizable IFN signature that was absent in polymicrobial NSTI's (Thänert et al, 2019). On the other side, we found that IL-1β production in response to *S. pyogenes* or its EVs converge at the NLRP3 inflammasome, although different sensing and signaling routes are employed. Accordingly, several studies have demonstrated that NLRP3 is a key sensor of Gram-positive bacteria and a recent report indicated that it participates in the recognition of *S. aureus* EVs (Mariathasan et al, 2006; Harder et al, 2009; Wang et al, 2020).

In addition to the canonical inflammasome, two other systems, the non-canonical inflammasome and the alternative inflammasome, contribute to the production of IL-1β in human monocytes (Shi et al, 2014; Gaidt et al, 2016), although little is known about the molecular interactions and crosstalk between the components of these three systems. In this respect, an important observation of this study is the identification of the non-canonical components of the inflammasome, caspase-4/-5, as mediators of IL-1β release induced by *Spy* EVs and to a lesser extent, by *S. pyogenes*. In most tissues, basal expression of caspase-5, like murine caspase-11, is low and transcription has been shown to be induced by LPS and IFN-γ (Lin et al, 2000; Casson et al, 2015; Viganò et al, 2015) through TLR/TRIF signaling, whereas caspase-4 is typically constitutively expressed (Shi et al, 2014). Consequently, we report a significant upregulation of caspase-5, but not caspase-4 upon challenge with *Spy* EVs or purified LTA. Although RNA sequencing revealed a slight increase in caspase-5 transcription in response to parental *S. pyogenes* cells, the expression level was considerably lower than in monocytes treated with *Spy* EVs and could not be validated by RT-qPCR. This could be explained by the differential sensing modes identified in monocytes, TLR8 for intact bacteria and TLR2 for Spy

EVs, the former sensing bacterial RNA and predominantly activating interferon responses and the latter robustly activating NF-kB target genes such as caspase 5. Furthermore, several type III secretion effector proteins from Gram-negative bacterial pathogens have been demonstrated to actively suppress inflammasome activation (Galle et al, 2008; Brodsky et al, 2010; LaRock and Cookson, 2012; Kobayashi et al, 2013). Moreover, Gram-positive *Streptococcus pneumoniae* as well as *Streptococcus oralis* have been shown to prevent inflammasome assembly by introducing oxidative stress into host cells through the production of hydrogen peroxide (Erttmann and Gekara, 2019). Therefore, future work should address whether caspase-5 gene expression is regulated in a temporal- and redox-dependent manner during infection with *S. pyogenes*.

During Gram-negative bacterial infection, binding of LPS to caspase-4/-5/-11 results in Gasdermin D cleavage, cytosolic leakage, and pyroptotic cell death (Shi et al, 2015). In contrast, the activation of caspase-4/-5/-11 leading to IL-1β production by Gram-positive bacteria does not always result in cell death (Hara et al, 2018; Krause et al, 2019), which is in agreement with our findings as we did not observe a reduction in *S. pyogenes*-induced cytotoxicity when caspase-4/-5 were inhibited. Interestingly, a recent report demonstrates that *S. pyogenes* infection induces caspase-4 expression in human neutrophils, indicating that this system participates in *S. pyogenes*-induced IL-1β production in other cell types as well (Williams et al, 2021).

Although TLR2 has traditionally been linked to the recognition of Gram-positive bacteria (Schwandner et al, 1999; Takeuchi et al, 1999; Underhill et al, 1999), *S. pyogenes* infection of mice lacking TLR2 results in variable susceptibility compared to wild-type mice (Gratz et al, 2008; Fieber et al, 2015). Furthermore, TLR2 contributes to the sensing of bacterial EVs (Prados-Rosales et al, 2011; Kim et al, 2012; Shen et al, 2012; Choi et al, 2018) and mediates priming of the NLRP3 inflammasome during infection with *S. aureus* or *S. pneumoniae* in macrophages (Witzenrath et al, 2011; Wang et al, 2020). In monocytes, however, priming is not required for NLRP3 inflammasome activation, allowing direct release of IL-1β/IL-18 in response to TLR stimulation via the alternative inflammasome pathway (Netea et al, 2009; Viganò et al, 2015; Gaidt et al, 2016; Gritsenko et al, 2020). Here, we show that TLR2 is differentially involved in the sensing of *Spy* EVs and *S. pyogenes*: whereas EVs induced IL-6 and IL-1β via this receptor, TLR2 contributed only slightly to IL-1β secretion during bacterial infection. In fact, our results confirm previous findings that endolysosomal TLR8 is the primary receptor for *S. pyogenes* in human monocytes (Eigenbrod et al, 2015) that activates inflammatory responses.

We provide evidence that LTA, decorating the surface of *S. pyogenes* and its EVs, modulates caspase-4/-5-dependent responses. Noteworthily, *S. pyogenes*-derived LTA has already been described to prime the NLRP3 inflammasome in macrophages (Richter et al, 2021). Furthermore, we have shown that caspase-4/-5-dependent IL-1β released in response to EVs is conserved for a variety of Gram-positive species. Nevertheless, in contrast to LPS, it has been demonstrated that caspase-4/-5/-11 do not bind directly to any component present in lysates of several Gram-positive species (Shi et al, 2014), suggesting that other sensors act upstream of these enzymes. In line with this hypothesis, caspase-11 coordinates the sensing of cytosolic LTA from *L. monocytogenes* as a component of

the NLRP6 inflammasome in mice (Hara et al, 2018). In addition, a newly identified sensor (NLRP7) that recognizes acylated Gram-positive lipoproteins, has been described in humans (Khare et al, 2012). Since we were unable to detect the expression of NLRP6 nor NLRP7 in monocytes, it remains to be elucidated whether any other cytosolic sensor acts upstream of caspase-4/-5 during encounters with Gram-positive bacteria. However, given that in our experimental set up, the cytosolic presence of LTA was not required to trigger caspase-4/-5 dependent IL-1β production, we believe that the upstream receptor(s) of this cascade are located on the cell surface. Since LPS stimulation of human monocytes has been reported to activate an alternative TLR4-linked and calcium-flux-dependent, one-step inflammasome pathway, it seems plausible that a similar mechanism might exist for TLR2 and LTA that potentially also involves caspase-4/-5. Indeed, TLR2 agonists have been implicated in the activation of the alternative inflammasome involving TRAF6, TAK1, and IKKβ in monocytes (Chen et al, 2023). Here, we found a dependence on TLR2 and the previously described alternative inflammasome component RIPK1 for IL-1β secretion solely for *Spy* EVs, indicating that inflammasome activation by intact *S. pyogenes* can be triggered by various sensor proteins. Consistent with these results, a previous report demonstrated that *Spy* EVs from a *lgt* mutant strain, characterized by a defect in lipoprotein modification, did not activate a TLR2-reporter cell line (Biagini et al, 2015). Furthermore, contrary to monocytes infected with *S. pyogenes*, monocytes stimulated with *Spy* EVs display hallmarks of the alternative inflammasome pathway (Gaidt et al, 2016) by lacking ASC speck formation or cleavage of GSDMD, and the resulting IL-1β secretion was also independent of potassium efflux. Similarly, previous research suggests that ASC specks are not required for caspase-1 activation in all instances (Nagar et al, 2021). While monocytes infected with *S. pyogenes* partially exhibit characteristics of the classical NLRP3 inflammasome, we found a comparable induction of caspase cleavage in both WT and a *slo* deletion strain, although Slo has been proven to be essential for inflammasome engagement in macrophages in providing signal 2 (Harder et al, 2009; Richter et al, 2021). In monocytes, however, signal 2 is dispensable and a TLR8 mediated one-step inflammasome activation process appears to take place. Moreover, both *S. pyogenes* and its EVs rely on caspase-4/-5 and -8 activities to facilitate IL-1β secretion. Caspase-4/-5 have been identified to facilitate LPS-induced IL-1β release in monocytes (Viganò et al, 2015), but how they fit into the alternative inflammasome pathway described by Gaidt et al, 2016 is currently unknown. Of note, we were unable to find evidence of caspase cleavage in response to *Spy* EVs. Given that the EV-induced IL-1β response is significantly weaker than that induced by *S. pyogenes*, this could be due to a sensitivity issue that prevents us from detecting caspase cleavage products. However, caspase activity does not always result in cleavage (Stennicke et al, 1999; Guey et al, 2014; Krause et al, 2018). In fact, initiator caspases (Muzio et al, 1998; Boatright et al, 2003; Pop et al, 2006), but also caspase-1 (Conos et al, 2016), have been reported to be activated by dimerization, with cleavage serving either to stabilize the dimer or to terminate protease activity (Boucher et al, 2018). In line with the aforementioned missing signs for classical inflammasome activation, it is plausible to assume that *Spy* EV induced-caspase activation is mild and does not necessarily result in cleavage. Finally, the discovery of additional players such as the tyrosine-protein kinase SYK as well

as the adaptor protein SCIMP mediating inflammasome activation through TLR2/4 (Zewinger et al, 2020), suggests that there might not be a unique signaling pathway for the alternative inflammasome. Instead, some degree of exchangeability between signaling molecules might ensure the proper transmission of danger signals depending on their origin. Future studies are necessary to determine the exact sequence and molecules involved in caspase-1 activation and subsequent IL-1β release in response to *S. pyogenes* or its derived EVs.

The leaderless cytokine IL-1β is generally released during pyroptosis, requiring GSDMD-dependent pore formation (Evavold et al, 2018; Heilig et al, 2018). However, the absence of cell death in response to *Spy* EVs points towards a GSDMD-independent route of secretion. Notably, secretory lysosomes have been described to serve as non-classical pathway for IL-1β release upon LPS-induced alternative inflammasome activation in human monocytes (Andrei et al, 1999). Likewise, we observed elevated IL-1β levels in monocytes stimulated with BafA1 to excrete lysosomes. In addition, we noticed increased autophagosome formation and turn over in response to *Spy* EVs and LTA, indicating the induction of autophagic processes. Accordingly, autophagy has been shown to be important for monocyte survival and differentiation (Zhang et al, 2012). Since increased autophagy has also been reported to promote unconventional IL-1β secretion (Dupont et al, 2011), further studies should evaluate the molecular composition of the monocytes soluble and vesicular proteome following *Spy* EV treatment.

In this study, we titrated and evaluated a biologically relevant *Spy* EV dosage based on particle count (and not on protein content, as mostly reported in bacterial EV-host-cell-response experiments) to avoid excessive over-dosing. Moreover, because EV isolation techniques are continuously evolving, we set out to refine our EV-isolation procedures with the aim to localize LTA (being the primary agent for monocyte activation) to vesicular and non-vesicular/soluble entities. To this end, monocytes were incubated with EV-depleted supernatants as well as *Spy* EVs isolated using SEC, presumably depleted from potential non-vesicular secreted entities to a higher extend as compared to EVs UC. Nonetheless, the monocyte´s responses to SEC EVs were comparable to EVs UC. We conclude that the additional sample preparation steps are informative but increase the risk of contamination. Decent EV amounts are hardly feasible to obtain for encapsulated, non-hypervesiculating *S. pyogenes* strains and may not cover the entire size spectrum of EVs present in culture supernatants, as is the case with ultracentrifugation.

Monocytes are usually short-lived cells, that rapidly undergo apoptosis through caspase-3 activation in the absence of adequate survival signals (Fahy et al, 1999; Goyal et al, 2002). We provide evidence that cleavage of caspase-3 is abrogated when monocytes are stimulated with LTA derived from either *Spy* EVs (UC and SEC) or EV-depleted supernatants. In addition, we found phosphorylation of the MAP kinase p44/42 (ERK1/2), previously shown to promote monocyte survival and differentiation (Bianchi et al, 2007; Farzam-Kia et al, 2023). Interestingly, TLR2 has been demonstrated to induce selective autophagy in an ERK1/2-dependent manner (Anand et al, 2011; Chang et al, 2013; Lu et al, 2017). These reports are consistent with our findings of increased LC3-II turn over upon *Spy* EV and LTA treatment. Lastly, we observe that IL-6 and particularly IL-1β secretion in response to *Spy* EVs highly correlates with the LTA content in our EV preparations. Overall, we find free and vesicular *Spy* LTA promoting IL-1β release while inhibiting apoptotic and pyroptotic cell death mechanisms. Since LTA has been demonstrated to induce antibody secretion and memory B cells (Yi et al, 2024) as well as dendritic cell maturation (Wesa and Galy, 2001; Luft et al, 2002), we speculate that vesicular LTA may serve as a "booster" to establish a humoral immune response. As we have previously shown IgG-immune reactive epitopes on *Spy* EV components (Resch et al, 2016), LTA-deficient *S. pyogenes* may be useful for follow-up experiments to investigate whether other factors within *Spy* EVs exhibit immunomodulatory properties.

Altogether, our study highlights a new role for TLR2 and caspase-4/-5 in the recognition of Gram-positive EVs in human monocytes, balancing cell death, proteostasis, and autophagy mechanisms. Whether *Spy* EVs have an overall protective role during *S. pyogenes* infection or, on the contrary, exacerbate the disease in humans, remains to be shown. Given that no causative organism could be identified in 70% of sepsis cases (Paoli et al, 2018), it is possible that systemically distributed microbial products, such as EVs, may induce an inflammation overshoot after the parental strain has been eradicated by antibiotics or trigger flares of inflammation resulting in chronic inflammatory diseases. In fact, for Gram-negative bacteria, OMVs have already been shown to be sensed by caspase-11 in mice (Vanaja et al, 2016) leading to sepsis-like symptoms (Park et al, 2010; Svennerholm et al, 2017; Kim et al, 2018), endothelial cell activation (Soult et al, 2014; Kovarik et al, 2016), and even disseminated intravascular coagulation (DIC) (Wang et al, 2019; Peng et al, 2020). Notably, an OMV-producing *Neisseria meningitidis* strain was the causative agent in a fatal case of meningococcal septicemia (Namork and Brandtzaeg, 2002). In contrast, evidence for the pathophysiological role of EVs from Gram-positive pathogens is scarce. To date, only EVs from *S. aureus* have been implicated in the development of atopic dermatitis as well as asthma (Kim et al, 2012; Hong et al, 2014). However, the discovery of *S. aureus* EVs in house dust supports their persistence. Thus, bacterial EVs circulating in human body fluids or tissues after an overcome infection, may have unknown long-lasting immunomodulatory effects and may contribute to sterile inflammation processes. Strikingly, the expression of the metalloprotease STEAP4, which in our study was upregulated solely in response to *Spy* EVs, has been demonstrated to be associated with chronic inflammation in patients with obesity (Catalán et al, 2013), diabetes (Taylor et al, 2025), or inflammatory bowel disease (Xue et al, 2017). However, despite the identification of cross-reactive antibodies against *S. pyogenes*, that also recognize epitopes in heart and brain tissues, and are responsible for streptococcal post-infectious autoimmune sequelae (Cunningham, 2016), it is not known whether hypervesiculating strains or their EVs alone contribute to the emergence of these pathologies. Our study uncovers *Spy* EV constituent LTA as a critical factor for a non-toxic innate immune response, which may tip the balance in acute situations such as sepsis or have long-lasting effects triggering flares of low-level inflammation during chronic or recurring infections (where increased EV release is warranted). Therefore, future experiments should explore the role of *Spy* EVs in severe complications of rheumatic fever such as rheumatic heart disease or neurological disorders using in vivo models.

# Methods

**Reagents and tools table**

| Reagent/resource | Reference or source | Identifier or catalog number |
|---|---|---|
| **Experimental models** | | |
| BLaER1 wild-type | Prof. Thomas Graf (Rapino et al, 2013) | N/A |
| BLaER1 TLR2⁻/⁻ | Prof. Holger Heine, this study | N/A |
| BLaER1 MYD88⁻/⁻ | Prof. Holger Heine (Vierbuchen et al, 2017) | N/A |
| BLaER1 ASC⁻/⁻ | Prof. Holger Heine (Vierbuchen et al, 2017) | N/A |
| BLaER1 CASP4⁻/⁻ | Prof. Holger Heine, this study | N/A |
| BLaER1 Non-targeting cells | This study | N/A |
| BLaER1 CASP5⁻/⁻ clone 1 and 2 | This study | N/A |
| BLaER1 CASP1⁻/⁻ clone 1 and 2 | This study | N/A |
| BLaER1 NLRP3⁻/⁻ clone 1 and 2 | This study | N/A |
| HEK-Blue™ hTLR2 Cells | Invivogen | hkb-htlr2 |
| ISS3348, M1 serotype Streptococcus pyogenes | G. Teti, University of Messina, Italy | N/A |
| M1T1 5448, M1 serotype Streptococcus pyogenes | M. Walker, University of Queensland, Australia | N/A |
| M1T1 5448 AP, M1 serotype Streptococcus pyogenes | M. Walker, University of Queensland, Australia | N/A |
| ISS3348 Δslo, M1 serotype Streptococcus pyogenes | This study | N/A |
| SA113, Staphylococcus aureus | ATCC | 35556 |
| 168, Bacillus subtilis | ATCC | 23857 |
| NEM316, Streptococcus agalactiae | S. Dramsi, Institute Pasteur, France | N/A |
| **Recombinant DNA** | | |
| pUC19Ωlox71-PermAM/B-ermAM/B-lox66 | Lab collection | pEC454 |
| pEC85ΩPgyrA-cre | Lab collection | pEC455 |
| pRO1600/ColE1, bla | https://doi.org/10.1093/nar/gks1119 | pEC801 |
| pEC801Ωslo(up)-lox71-PermAM/B-ermAM/B-lox66-slo(down) | This study | pEC2898 |
| **Antibodies** | | |
| Anti-S. pyogenes LTA | QEDBio | 15711 |
| Anti-Streptolysin O (Hemolytic streptococcus) antibody | Abcam | ab188539 |
| Gold-conjugated goat anti-mouse IgG | Jackson Immuno | 115-215-146 |
| IL-1β (D3U3E) rabbit mAb | Cell Signaling | 12703 |
| Cleaved-IL-1β (Asp116) (D3A3Z) rabbit mAb | Cell Signaling | 83186 |
| HSP90 | Proteintech | 13171-1-AP |
| anti-Caspase-1 (p20) (human), mAb | Adipogen | AG-20B-0048-C100 |
| Anti-Caspase-4 (Human) mAb | Medical & Biological Laboratories Co., LTD. | M029-3 |
| Anti-Caspase-5 (Human) mAb | Medical & Biological Laboratories Co., LTD. | M060-3 |
| LC3A/B (D3U4C) XP® Rabbit mAb | Cell Signaling | 12741S |
| Caspase-3 (D3R6Y) Rabbit mAb | Cell Signaling | 14220 |
| Cleaved Caspase-3 (Asp175) (5A1E) | Cell Signaling | 9664 |
| Phospho-p44/42 MAPK (Erk1/2) (Thr202/Tyr204) (D13.14.4E) XP® Rabbit mAb | Cell Signaling | 4370 |
| Gasdermin D (E8G3F) Rabbit mAb | Cell Signaling | 97558 |
| Cleaved Gasdermin D (Asp275) (E7H9G) Rabbit mAb | Cell Signaling | 36425 |

| Reagent/resource | | Reference or source | Identifier or catalog number |
|---|---|---|---|
| GAPDH (14C10) Rabbit mAb | | Cell Signaling | 2118 |
| ASC/TMS1 Monoclonal antibody | | Proteintech | 67494-1-Ig |
| Anti-*Streptococcus pyogenes* Group A Carbohydrate antibody | | Abcam | ab9191 |
| Amersham ECL rabbit IgG, HRP-lined whole Ab | | GE Healthcare | NA934-1ML |
| Anti-mouse IgG, HRP-linked Antibody | | Cell Signaling | 7076S |
| Cy™2 AffiniPure™ Donkey Anti-Mouse IgM | | Jackson ImmunoResearch | 715-225-140 |
| Cy™5 AffiniPure™ Donkey Anti-Goat IgG (H + L) | | Jackson ImmunoResearch | 705-175-147 |
| **Oligonucleotides and other sequence-based reagents** | | | |
| **Primers for qRT-PCR** | | **Sigma-Aldrich** | **N/A** |
| **Gene** | **Oligo code** | **Sequence** | **F/R** |
| *TUBB* | OLEC11016 | TGGACTCTGTTCGCTCAGGT | F |
| *TUBB* | OLEC11017 | TGCCTCCTTCCGTACCACAT | R |
| *GAPDH* | OLEC11889 | ACTTTGGTATCGTGGAAGGACT | F |
| *GAPDH* | OLEC11890 | GTAGAGGCAGGGATGATGTTCT | R |
| *IL1A* | OLEC10990 | AGATGCCTGAGATACCCAAAACC | F |
| *IL1A* | OLEC10991 | CCAAGCACACCCAGTAGTCT | R |
| *IL1B* | OLEC11897 | CCAGTGAAATGATGGCTTATTAC | F |
| *IL1B* | OLEC11898 | CTGTAGTGGTGGTCGGAGATT | R |
| *IL6* | OLEC10966 | ACTCACCTCTTCAGAACGAATTG | F |
| *IL6* | OLEC10967 | CCATCTTTGGAAGGTTCAGGTTG | R |
| *CCL4* | OLEC10978 | CTGTGCTGATCCCAGTGAATC | F |
| *CCL4* | OLEC10979 | TCAGTTCAGTTCCAGGTCATACA | R |
| *CASP5* | OLEC10996 | TTCAACACCACATAACGTGTCC | F |
| *CASP5* | OLEC10997 | GTCAAGGTTGCTCGTTCTATGG | R |
| *CASP4* | OLEC10998 | CAAGAGAAGCAACGTATGGCA | F |
| *CASP4* | OLEC10999 | AGGCAGATGGTCAAACTCTGTA | R |
| *WNT5A* | OLEC11000 | ATTCTTGGTGGTCGCTAGGTA | F |
| *WNT5A* | OLEC11001 | CGCCTTCTCCGATGTACTGC | R |
| *IFNG* | OLEC10986 | TCGGTAACTGACTTGAATGTCCA | F |
| *IFNG* | OLEC10987 | TCGCTTCCCTGTTTTAGCTGC | R |
| *IFNB1* | OLEC10988 | ATGACCAACAAGTGTCTCCTCC | F |
| *IFNB1* | OLEC10989 | GGAATCCAAGCAAGTTGTAGCTC | R |
| *PTGFRN* | OLEC11006 | CCTGCAACGTCAGTGACTATG | F |
| *PTGFRN* | OLEC11007 | AGTCCGCCTTAACAGGATCTC | R |
| *CCR2* | OLEC11008 | TACGGTGCTCCCTGTCATAAA | F |
| *CCR2* | OLEC11009 | TAAGATGAGGACGACCAGCAT | R |
| **Guide RNAs** | | **Thermo Fisher Scientific** | **N/A** |
| **Target gene** | **Provider** | **Guide RNA target sequence** | **Primer sequence (5′-3′)** |
| Non-targeting | A35526, Thermo Fisher | AAAUGUGAGAUCAGAGUAAU | n/a |
| *TLR2* | Designed with Benchling software | TGTCCATATTTCCCACTCTC | F: TGTAATTCCGGATGGTTGTGC<br>R: CCTGAAACAAACTTTCATCGGTG |
| *CASP1_C1* | CRISPR917229_SGM, Thermo Fisher | GACAGTATTCCTAGAAGAAC | F: CCATGCATACTCTTGAAGCAGC<br>R: GCACTCTCTCATGGCAAGTTTG |
| *CASP1_C2* | CRISPR937503_SGM, Thermo Fisher | GGCATCTGCGCTCTACCATC | F: CATTTAGTTTCACCCCGCTCC<br>R: ACCATACACATGTACCTGCCC |

| Reagent/resource | | Reference or source | Identifier or catalog number | |
|---|---|---|---|---|
| CASP4 | | Designed with Benchling software | TGCAGCTCATCCGAATATGG | F: TGGGGATTAGGAGGGGACGATT<br>R: ATGGTTACTGTCATCCCCACCC |
| CASP5_C1 | | CRISPR641087_SGM, Thermo Fisher | ACACGATGTTCTGACATTGA | F: TAACACTGCATGGGCCTTGG<br>R: TGTGTGTTATTCGCTGGAGACA |
| CASP5_C2 | | CRISPR718889_SGM, Thermo Fisher | GGGGCTCACTATGACATCGT | F: TTTCATTTGCTAGTGCATGTGCG<br>R: TAGAGGTTAGGGAAGGTGGCA |
| NLRP3_C1 | | CRISPR1098152_SGM, Thermo Fisher | CGCTAATGATCGACTTCAAT | F: CTCATCCGTGTGCCGTGTTC<br>R: CACTATGTGCACTTGGCCACAA |
| NLRP3_C2 | | CRISPR1098154_SGM, Thermo Fisher | GATCGCAGCGAAGATCCACA | F: CTCATCCGTGTGCCGTGTTC<br>R: CACTATGTGCACTTGGCCACAA |
| **Primers for slo deletion** | | Sigma-Aldrich | N/A | |

| Purpose | Code | Sequence 5′–3′ | F/R | Usage* |
|---|---|---|---|---|
| **Generation of slo deletion strain** | | | | |
| pEC801 linearisation | OLEC9943 | CTGCAGGCATGCAAGCTTGCG | F | PCR |
| | OLEC9944 | TCTAGAGGATCCCCGGGTACCGAG | R | PCR |
| slo upstream fragment | OLEC12497 | ACCCGGGGATCCTCTAGAAAGTA ACAATATGTATAAGGTGCCAAAG | F | PCR |
| | OLEC12498 | TGTATGCTATACGAACGGTAGTC CTTCATACCTTTTTATCATTCTAAAATG | R | PCR |
| slo downstream fragment | OLEC12499 | GCATACATTATACGAACGGTAGACTG GTTCAAGAGGTTCGTCAAG | F | PCR |
| | OLEC12500 | AAGCTTGCATGCCTGCAGAGTTGGGG TCAAATCAGTCCAA | R | PCR |
| lox71-P*ermAM/B*-ermAM/B-lox66 | OLEC1943 | TACCGTTCGTATAGCATACATTATA CGAAGTTATCCGTAGC GGTTTTCAAAATTTGCAACC | F | PCR |
| | OLEC1932 | TACCGTTCGTATAATGTATGCTATA CGAAGTTATTTA TTTCCTCCCGTTAAATAATAGATA ACTATTAAA | R | PCR |
| PCR ligation | OLEC11417 | ACCCGGGGATCCTCTAGA | F | LM-PCR |
| | OLEC11418 | AAGCTTGCATGCCTGCAG | R | LM-PCR |
| Validation of locus | OLEC12281 | CATTGCTCCAAATGGTGACC | F | SEQ |
| | OLEC12282 | GAGATTTCCAGCCTTCATTATAACC | R | SEQ |
| **Verification of virulence regulator gene integrity in mutant strains** | | | | |
| covRS | OLEC4856 | TCGCTAGAAGACTATTTGACCAT | F | PCR, SEQ |
| | OLEC4867 | AAGACATCGCGATTGACAGT | R | PCR, SEQ |
| | OLEC3609 | GGCTATGTTCAAGTCTTTCATG | F | SEQ |
| | OLEC3610 | CCAAATAACTCAACAACTAGTAGC | R | SEQ |
| mga | oliRN172 | AGTTGACTAACCAATTGAT CTACGCCTTTT | F | PCR, SEQ |
| | OLEC290 | TTAACCTCTGTTTGATTCGC | R | PCR, SEQ |
| ropB | OLEC4854 | AGCGACTATCATCCGAAACAT | F | PCR, SEQ |
| | OLEC4855 | GCCCTGGAGCTGTTGAGATA | R | PCR, SEQ |
| *LM-PCR PCR-mediated ligation, SEQ sequencing | | | | |
| **Chemicals, enzymes and other reagents** | | | | |
| Recombinant Human IL-3 | | Peprotech | 200-03 | |
| Recombinant Human M-CSF | | Peprotech | 300-25 | |
| Recombinant β-Estradiol | | Sigma-Aldrich | E2257-1MG | |
| Cas9 TrueCutV2 | | Thermo Fisher Scientific | A36498 | |
| S. pyogenes LTA | | Sigma-Aldrich | L3140-5MG | |

| Reagent/resource | Reference or source | Identifier or catalog number |
|---|---|---|
| *S. aureus* LTA | Invivogen | tlrl-pslta |
| Bafilomycin A1 | Enzo lifesciences | BML-CM110-0100 |
| Ac-YVAD-cmk | Invivogen | inh-yvad |
| MCC950 | Invivogen | inh-mcc |
| VX-765 | Invivogen | inh-vx765i-1 |
| Ac-LEVD-CHO | Enzo lifesciences | ALX-260-065 |
| Ac-LEVD-AFC | Enzo lifesciences | ALX-260-084 |
| EasySep Human Monocyte Isolation Kit | Stemcell | 19359 |
| Human GM-CSF | Miltenyi Biotec | 130-093-866 |
| LDH-Cytotoxicity Assay Kit II | Abcam | ab65393 |
| P3 Primary Cell 4D-Nucleofector Kit | Lonza | V4XP-3012 |
| Power SYBR™ Green RNA-to-CT™ 1-Step Kit | Thermo Fisher Scientific | 4367659 |
| Human IL-6 ELISA | R&D Systems | DY206 |
| Human IL-1β ELISA | R&D Systems | DY201 |
| Human IL-1α ELISA | R&D Systems | DY200 |
| Pro-Q™ Emerald 300 Lipopolysaccharide Gel Stain Kit | Thermo Fisher Scientific | P20495 |
| SuperSignal™ West Pico PLUS Chemiluminescent Substrate | Thermo Fisher Scientific | 34580 |
| GSK3145095 | Selleckchem | S8845 |
| TLR2-IN-C29 | Selleckchem | S6597 |
| Cu-CPT22 | Selleckchem | S8677 |
| CU-CPT9a | Invivogen | inh-cc9a |
| CLI-095 | Invivogen | tlrl-cli95-4 |
| M5049 | Invivogen | inh-m5049 |
| TLR9-IN-1 | Hycultec | HY-148045 |
| Ac-IETD-CHO | Cayman Chemical | 27100 |
| HEK-Blue™ Selection | Invivogen | hb-sel |
| Normocin® | Invivogen | ant-nr-1 |
| HEK-Blue™ Detection | Invivogen | hb-det2 |
| Hoechst 33342 Solution (20 mM) | Thermo Fisher Scientific | 62249 |
| LTA (Lipoteichoic Acid) ELISA Kit | Biocat | AKR-5153-CB |
| TRPS Calibration Particles | Izon | CPC100, CPC200 |
| Nanopores | Izon | NP200, NP150, NP100 |
| Izon Reagent kit | Izon | RK3 |
| **Software** | | |
| GraphPad Prism v10.0 | GraphPad | http://www.graphpad.com |
| NTA 3.1 analytical software | Malvern Panalytical | https://www.malvernpanalytical.com |
| Benchling gRNA design tool | Benchling | https://www.benchling.com |
| ICE Analysis | Synthego webtool | https://ice.synthego.com |
| DIA-NN 1.7.12 | Demichev et al, 2020 | N/A |
| DEseq2 (version 1.26.0.) | Love et al, 2014 | N/A |
| STAR (version 2.5.2a) | Dobin et al, 2013 | N/A |
| RNASeQC (version 2.1.0) | DeLuca et al, 2012 | N/A |
| QuPath | Bankhead et al, 2017 | https://qupath.github.io/ |
| Exoid Control Suite (v1.4.0.88) Izon Data Suite v1.0.2.32 v.3.1) | Izon | https://support.izon.com/how-can-i-get-the-latest-software-release |

| Reagent/resource | Reference or source | Identifier or catalog number |
|---|---|---|
| **Other** | | |
| Buffy coats | German Red Cross Blood Transfusion Service | N/A |
| 8 Well Chamber, removable | ibidi | 80841 |
| qEV1 Columns 70 nm | IZON | IC1-70 |
| Nalgene™ Rapid-Flow™ Sterile Disposable Filter Units with PES Nylon Membrane | Thermo Fisher Scientific | 166-0045 |
| Amicon Ultra 15-30 kDa MWCO | Merck, Millipore | UFC9030 |
| Amicon Ultra 4-10 kDa MWCO | Merck, Millipore | UFC201024 |

## Materials availability

BLaER1 knock-out cell lines newly generated for this study in our laboratory can be requested directly from the corresponding authors. The availability of the *TLR2*, *MYD88*, *ASC*, and *CASP4* BLaER1 knock-out cell lines is limited due to an MTA agreement. These cell lines can be requested directly from the laboratories where they were generated.

## Bacterial strains

*S. pyogenes* ISS3348 (G. Teti, University of Messina, Italy), M1T1 5448, M1T1 5448 AP (both M. Walker, University of Queensland, Australia), and *S. agalactiae* NEM316 (S. Dramsi, Institute Pasteur, France) were grown in Todd Hewitt Broth (THB, BD) at 37 °C with 5% $CO_2$ without shaking. Trypticase soy agar (TSA, BD) supplemented with 3% sheep blood (Xebio Diagnostics) was used as a solid medium. *B. subtilis* 168 (ATCC 23857) and *S. aureus* SA113 (ATCC 35556) were grown in Brain Heart Infusion agar or broth (BHI, Sigma-Aldrich) at 37 °C with shaking.

## Eukaryotic cells: BLaER1 cells

BLaER1 wild-type cells were a gift from Thomas Graf (Center for Genomic Regulation, Barcelona, Spain) (Rapino et al, 2013). BLaER1 $ASC^{-/-}$, $MYD88^{-/-}$, $CASP4^{-/-}$, and $TLR2^{-/-}$ knockout cells were kindly provided by Holger Heine (German Center for Lung Research, Borstel, Germany). BLaER1 $CASP1^{-/-}$, $CASP5^{-/-}$, and $NLRP3^{-/-}$ were generated during this study. Cells were maintained in RPMI 1640 medium with Glutamax (Thermo Fisher Scientific) supplemented with 1% penicillin–streptomycin (P/S, Sigma-Aldrich), 10% FBS and 1% sodium pyruvate (both from Thermo Fisher Scientific) at 37 °C in a humidified atmosphere containing 5% $CO_2$. Sex of the cell lines was not a consideration in this study. All cell lines were routinely tested for mycoplasma contamination with the MycoAlertTM Mycoplasma Detection Kit (Lonza).

## Eukaryotic cells: primary monocytes

Human primary monocytes were purified from buffy coats from the German Red Cross Blood Transfusion Service upon ethical approval of the local authorities. For the isolation, buffy coats were diluted 1:1 in PBS and centrifuged (300 × g, 10 min, room temperature). The upper fraction was removed to reduce the final presence of platelets and the remainder was layered and separated in Pancoll density gradients (1.077 g/ml, Pan Biotech; 800× g, 20 min, room temperature, without break). Peripheral Blood Mononuclear Cells (PBMCs) were collected from the interphase and washed three times (500 × g, 10 min, 20 °C) with PBS containing 2% FBS and 0.1% Ethylenediaminetetraacetic acid (EDTA, Thermo Fisher Scientific). PBMCs were counted with CASY (OMNI Life Science) and resuspended at $50 \times 10^6$ cells/ml for monocyte purification. Classical monocytes ($CD14^+CD16^-$) were separated using the EasySep Human Monocyte Isolation Kit (Stemcell) according to the manufacturer's instructions. Monocytes were maintained in RPMI 1640 medium with Glutamax supplemented with 10% FBS at 37 °C in a humidified atmosphere containing 5% $CO_2$. In some experiments, monocytes were allowed to differentiate into adherent macrophages by prolonged cultivation (7 days) in the presence of 100 ng/mL GM-CSF. Sex of the primary cells was not considered in this study.

## Eukaryotic cells: HEK-Blue TLR2 cells

HEK-Blue TLR2 cells (Invivogen) were maintained in DMEM medium supplemented with 1% penicillin–streptomycin (P/S, Sigma-Aldrich), 10% FBS, 100 µg/mL Normocin, and 1× HEK-Blue selection antibiotic mix at 37 °C in a humidified atmosphere containing 5% $CO_2$.

## Bacterial EVs: isolation and characterization

*S. pyogenes* and *S. agalactiae* EVs were isolated from late-logarithmic phase cultures in THB ($OD_{600\ nm}$: 0.4 in an Epoch 2, Agilent microplate reader). *B. subtilis* and *S. aureus* EVs were isolated from late-logarithmic phase cultures in BHI ($OD_{600\ nm}$: 0.8). 450 mL bacterial cultures were first centrifuged (9300 × g, 10 min, 10 °C) using a JLA 10.500 rotor and fitting tubes (both from Beckman Coulter), and cell free supernatants were centrifuged again to remove remaining cell debris (18,600 × g, 15 min, 10 °C). The supernatants were filtered twice through 0.22 µm polyethyl-sulfone membranes (Thermo Fisher Scientific) and EVs purified by ultracentrifugation at 175,000 × g (42,000 rpm) for 4 h at 10 °C using a Ti45 rotor with 70 ml polycarbonate-fitting tubes and an Optima XPN 100 ultracentrifuge. EV-pellets were resuspended and merged in phosphate-buffered saline (PBS, Thermo Fisher Scientific), centrifuged again at 175,000 × g for 3 h at 10 °C to concentrate and remove non-EV soluble proteins. Each final batch was resuspended in 1 ml of PBS and characterized using the NTA Nano Sight 300 device (NS300, Malvern Panalytical). The EV

concentration (particles/mL) and size distribution of the particles in suspension (nm) were determined using the NTA 3.1 analytical software for data analysis (Malvern Panalytical) and revealed an average concentration for *Spy* EVs of ~$7 \times 10^{11}$ EVs/ batch. In addition, size, concentration, and zeta-potential were determined by tunable resistance pulse sensing (TRPS) using the Exoid (ECS v1.4.0.88) using appropriate nanopores (100, 150 and 200 nm) and calibration beads (CPC100, CPC200 and TKP200). 0.22 μm filtered PBS was used as the electrolyte and sample diluent. Size, concentration, or zeta potential were measured at 3 pressures (P1, P2, P3) or 3 voltages (V1, V2, V3) according to Izon-recommendations. Post-acquisition data processing was performed within the Izon Control Suite (v.3.1) in multi-and single pressure or charge analyses modules for size-concentration or zeta determination, respectively. Size-concentration histograms are provided in Appendix Fig. S2B.

For LTA-depleted EVs, *S. pyogenes* cultures were grown in the presence of 5 μM LtaS-IN-1771 (Richter et al, 2013) until reaching $OD_{600 \, nm}$: 0.4 and further processed as described above.

## Bacterial EVs: size exclusion chromatography (SEC) and characterization

A batch of double-filtered *S. pyogenes* supernatant was split into half to compare *Spy* EVs purified by conventional ultracentrifugation or size exclusion chromatography (SEC). For SEC, the supernatant was concentrated using an Amicon Ultra-15 centrifugal filter with a molecular weight cut-off 30 kDa (Merck, Millipore). In total, 1 mL of the concentrate were loaded onto a qEV 70 nm column and 28 fractions of 0.5 mL including void volume (3.5 ml) were collected in PBS. Fractions containing *Spy* EVs were identified by separation on a 12% SDS-PAGE and SYPRO Ruby protein gel staining. The EV containing fractions were pooled and concentrated using an Amicon Ultra-4 centrifugal filter device (MWCO 10 kDa). PBS only was concentrated using an Amicon Ultra-4 centrifugal filter device as well to serve as negative control. *Spy* EVs size, concentration, and zeta potential were determined as described above with the Exoid. SDS-Page protein profiles of SEC, size-concentration histograms, and IL-1β response to SEC are provided in Appendix Figs. S2 and S3.

## EV-depleted supernatants

EV-depleted supernatants were concentrated after EVs have been removed using ultracentrifugation using an Amicon Ultra-15 centrifugal filter device (MWCO 10 kDa). The final volume of the concentrated supernatant was about 3.2 times higher than that of the *Spy* EV batches. THB media alone was concentrated as well and used as control in monocyte experiments.

## Streptolysin O deletion

The chromosomal deletion of *slo* in *S. pyogenes* ISS348 was performed using the Cre-lox recombination approach as previously described (Le Rhun et al, 2017). The up- and downstream flanking regions of *slo* were amplified from *S. pyogenes* ISS3348 genomic DNA and ligated to the lox71-PermAM/B-ermAM/B-lox66 cassette by PCR using the primers listed in Reagents and Tools Table. The slo(up)-lox71-PermAM/B-ermAM/B-lox66-slo(down) fragment

was then integrated into pEC801 using Gibson Assembly® Master Mix generating pEC2898. *S. pyogenes* ISS3348 competent cells were adjusted to an $OD_{620}$ of 1.6 in 100 μL and incubated with 16.5 μg of Cfr42I-linearized pEC2898 on ice for 50 min before electroporation in a 0.1 cm electrode gap cuvette (1.5 kV, 400 Ω and 25 μF pulse; BioRad Gene Pulser Xcell electroporation system). After recovery in THB for 3 h, mutants were selected for two passages on TSA plates with 3% blood and 3 μg/mL erythromycin, cultured in THB without antibiotics, and flash-frozen in liquid nitrogen in THB with 20% glycerol. The obtained strains were validated by Sanger sequencing using OLEC12281 and OLEC12282.

To remove the ermAM/B resistance marker, competent cells of ISS3348 Δslo::lox71-ermAM/B-lox66 were adjusted to an $OD_{620}$ of 2 in 100 μL and incubated with 500 ng of pEC455 on ice for 1 h. After electroporation at 1.5 kV, 400 Ω and 25 μF, cells were recovered for 2–3 h in THB. Transformants were selected on TSA plates with 3% blood and 300 μg/mL kanamycin and then grown in THB with 300 μg/mL kanamycin. To lose the Cre-expressing pEC455, transformants were then passaged four times in the absence of kanamycin, and strains susceptible to both kanamycin and erythromycin were finally cultured in THB and flash-frozen in liquid nitrogen in THB with 20% glycerol. The marker-less deletion strain of *slo* was validated by Sanger sequencing using primers targeting the *slo* locus and the virulence regulators *mga, covRS* and *ropB* (Reagents and Tools Table).

## Transdifferentiation of BLaER1 cells

BLaER1 cells were transdifferentiated into monocyte-like cells for 7 days in six-well plates by seeding $10^6$ cells/well in 3 mL of RPMI 1640 medium with Glutamax, 1% P/S, 10% FBS, 1% sodium pyruvate, 10 ng/mL of human recombinant IL-3, 10 ng/mL of human recombinant Macrophage Colony-stimulating Factor (both from PeproTech) and 100 nM of β-Estradiol (Sigma-Aldrich). Half of the medium was exchanged on days 2 and 5 with fresh medium containing the same supplements. On day 7, cells were harvested, counted with CASY and resuspended in fresh medium without cytokines or P/S at a final density of $1 \times 10^6$ cells/mL. Cells were seeded according to the experiment and maintained in incubators at 37 °C in a humidified atmosphere containing 5% $CO_2$.

## Construction of BLaER1 knock-outs

Guide RNAs (gRNA) designed with CRISPR gRNA design tool (Benchling) or commercially predesigned were selected for high on-target and low off-target scores (Reagents and Tools Table). Clustered Regularly Interspaced Short Palindromic Repeats (CRISPR) editing of BLaER1 cells was performed using the P3 Primary Cell 4D-Nucleofector Kit with strips (Lonza) according to the manufacturer's protocol. Briefly, 1 μL of Cas9 TrueCutV2 (Thermo Fisher Scientific) + 1.6 μL of gRNA (100 μM) were complexed for 15 min at room temperature (RT). The complexed ribonucleoproteins were added to $10^6$ cells previously resuspended in 20 μL of kit buffer and the mix was transferred to an electroporation cuvette. The Amaxa 4D-Nucleofector (Lonza, program EH-140) was used for electroporation and cells were transferred to pre-warmed medium and incubated in a 37 °C with 5% $CO_2$. The whole bulk of targeted or non-targeted cells (control)

was single-cell sorted in 96-well plates using a Sony SH800S Cell Sorter and individual clones were grown at 37 °C with 5% $CO_2$. Genomic DNA was extracted with the NucleoSpin Tissue Kit (Macherey Nagel), PCR amplified (primers listed in the Reagents and Tools Table) and sent for Sanger sequencing (Microsynth). Upon confirmation of the presence of frameshift insertions/deletions (ICE Analysis Webtool Synthego), individual clones were stored in liquid nitrogen.

## Infection assays and cell treatments

Infection assays were performed with *S. pyogenes* from late-logarithmic phase cultures in THB ($OD_{600}$: 0.4). Bacteria were centrifuged ($6000 \times g$, 3 min, 20 °C), washed with PBS and passed ten times through a 27 Gauge syringe insert (NeoLab) in a 1 mL syringe (VWR) to separate the cocci chains. Human monocyte assays were performed either in 96-well plates with $10^5$ cells or 48-well plates with $10^6$ cells per well. Cells were stimulated with a Multiplicity of Infection (MOI) of 5 or $7 \times 10^8$ particles of *Spy* EVs per $10^5$ cells. Infection of BLaER1 cells was performed in 24 well plates with $10^6$ cells per well. BLaER1 cells were stimulated with a MOI of 1 or $3.5 \times 10^{10}$ particles of EVs. For experiments with EVs from other Gram-positive species, the volume of EVs was adjusted to add the same number of particles as for *Spy* EVs. Experiments with SEC EVs were performed in 48-well plates with $10^6$ cells per well. Cells were stimulated with $9.8 \times 10^6$, $9.8 \times 10^7$, or $1.9 \times 10^8$ SEC EVs corresponding to 1, 10, or 20 µL of the SEC EV batch.

Upon bacterial or EV challenge, plates were centrifuged ($300 \times g$, 5 min, 20 °C) for synchronization. For samples collected overnight (18 h post-treatment), 1% P/S was added after 1 h to inhibit bacterial replication and limit cell death. After centrifugation of the plates ($500 \times g$, 5 min, 20 °C), the supernatants were collected at the designated time points and used for Lactate dehydrogenase assay (LDH). The remaining supernatants were stored at −20 °C until used for Enzyme-linked immunosorbent assay (ELISA). For immunoblotting experiments of cell culture supernatants, the FBS concentration of the media was reduced to 2%.

Purified *S. pyogenes* LTA (Sigma-Aldrich) or *S. aureus* LTA (Invivogen) was used at 1 µg/mL. LPS-B5 (lipopolysaccharide from *E. coli* serotype 055:B5, Invivogen) was used at 200 ng/mL. All chemical inhibitors were added 1 h prior to the addition of *S. pyogenes*, bacterial EVs or purified LTA. The inhibitors Ac-LEVD-CHO (Caspase-4/-5, Enzo Lifesciences), Ac-YVAD-cmk (caspase-1), Ac-IETD-CHO (caspase-8), MCC950 (NLRP3), and VX-765 (caspase-1/-4/-5) were used at a final concentration of 20 µM. GSK3145095 (RIPK1) was used at 2.5 µM. TLR2 inhibitors TLR2-IN-C29 and Cu-CPT22 were used at 50 µM and 10 µM. Final concentrations for other TLR inhibitors were: CLI-095 (TLR4; 0.05; 0.5; 5 µM), M5049 (TLR7/8; 0.1 and 1 µM), TLR9-IN-1 (TLR9; 5 µM), and Cu-CPT9a (TLR8; 5 µM).

## RNA isolation

Human monocytes and BLaER1 cells were seeded at $1 \times 10^6$ cells/mL in six-well plates. Cells were left untreated or infected with *S. pyogenes* or treated with *Spy* EVs ($7 \times 10^9$ or $7 \times 10^{10}$ particles for monocytes or BLaER1 cells, respectively). Then, cells were collected by centrifugation ($500 \times g$, 10 min, RT), and RNA was extracted with Trizol reagent (Thermo Fisher Scientific) according to the

manufacturer's protocol. Samples were concentrated and DNAse-treated using the RNA Clean & Concentrator kit (Zymo Research). RNA quality was assessed with Agilent 2100 Bioanalyzer (Agilent Technologies) and a cut-off of RIN > 8 was established.

## Library preparation and RNA sequencing

The Sequencing Core Facility at the Max Planck Institute for Molecular Genetics (Berlin, Germany) prepared cDNA libraries using the KAPA Stranded mRNA-Seq Kit (Roche) according to the manufacturer's instructions. Monocyte and BLaER1 datasets were sequenced in single-end mode (75 base pairs) on a HiSeq 4000 instrument (Illumina) or a NextSeq 500 instrument (Illumina), respectively.

## RNA sequencing: data processing

Fastq files from the Illumina sequencing were aligned with STAR (Dobin et al, 2013) against a combined reference containing the GRCh38 human genome reference (Schneider et al, 2017) and the *S. pyogenes* reference genome NC_002737.2 (Ferretti et al, 2001). Gene expression was quantified by counting overlapping features reads using RNASeQC 2.1.0 (DeLuca et al, 2012) against the Gencode v29 human gene annotation (Frankish et al, 2019). Normalization and differential gene expression (DE) were performed using the DEseq2 package (Love et al, 2014). Pathway analysis was performed using GAGE (Luo et al, 2009) and GO term enrichment analysis was generated using The Gene Ontology resource applying Bonferroni correction for multiple testing (Ashburner et al, 2000; Mi et al, 2019; Carbon et al, 2021). Data were visualized using Python programming environment using the matplotlib (John D. Hunter and Dept. of Computer Science, Univ. of Utah, Salt Lake, 2007), seaborn (Waskom, 2021), and bokeh (https://bokeh.org) libraries. Venn diagrams were generated using https://www.interactivenn.net/.

## Gene expression analysis: quantitative PCR

The Power SYBR™ Green RNA-to-CT™ 1-Step Kit (Thermofisher Scientific) was used according to the manufacturer's instructions. RNA samples were measured in a QuantStudio 5 qPCR machine (Thermo Fisher Scientific). Relative expression was calculated as ΔCt: Ct gene of interest-Ct housekeeping reference gene. All qPCR primers used in this study were selected from PrimerBank (Wang and Seed, 2003) and are summarized in the Reagents and Tools Table. For relative quantification, samples were normalized to *TUBB* and *GAPDH*.

## LDH release assay and ELISA

Release of lactate dehydrogenase (LDH) was quantified using the Cytotoxicity Assay Kit II (Abcam) according to the manufacturer's instructions. The percentage of cytotoxicity was calculated as follows:

$$Cytotoxicity\,(\%) = \frac{(Test\,sample - Low\,control)}{(High\,control - Low\,control)} \times 100$$

Human IL-1α, IL-1β and IL-6 (R&D Systems) were measured by ELISA according to the manufacturer's protocol. LTA was

measured using the LTA ELISA Kit (Biocat) according to the manufacturer's instructions.

## Caspase-4/5 substrate cleavage assay

Monocytes were incubated in Hank's balanced salt solution (HBSS) supplemented with 1% non-essential amino acids, 1% bovine serum albumin, 1% Glutamax, and 10 mM HEPES. At 20 h post-treatment, cell culture supernatants were harvested and cellular debris was removed by centrifugation at $500\times g$. 100 µL of the cell culture supernatant were mixed with 100 µL of caspase assay buffer (20 mM HEPES [pH7,4], 0,1 M NaCl, 10% Glycerol, 5 mM DTT) containing a final concentration of 250 µM Ac-LEVD-AFC in a 96-well black plate followed by incubation at 37 °C in a humidified atmosphere containing 5% $CO_2$ for 2 h. The fluorescence was measured using a Cytation 5 microplate reader (Agilent BioTek) at ex/em 400/505 nm. All readings were normalized to untreated control wells.

## TLR2 activity assay

HEK-Blue TLR2 cells were seeded in 96-well plates with 280,000 cells/200 µL of HEK-Blue Detection medium per well. Cells were stimulated for 18 h and production of the reporter gene secreted embryonic alkaline phosphatase (SEAP) upon TLR2 activation was monitored by absorbance measurement at 620 nm.

## Immunoblot

In total, 30 µl of EVs were boiled with 6 µL of SDS-PAGE buffer (6×, Thermo Fisher Scientific) for 5 min at 95 °C. To determine the presence of LTA in LtaS-IN-1771-treated EV preparations, bacterial pellets of 50 mL cultures were boiled in PBS with 6× SDS-PAGE Buffer. The volume for LtaS-IN-1771-treated cultures was normalized to the $OD_{600}$ measured for the corresponding DMSO-treated control culture for each time point. Protein extraction from monocytes was performed using TRIzol reagent (Thermo Fisher Scientific) according to the manufacturer's instructions. The resulting protein pellet was air-dried and dissolved in lysis buffer (2 M Thiourea, 8 M Urea, 0.065 M CHAPS, 0.13 M DTT, 0.079 M Tris). The Bradford method was used to determine protein concentrations. Equal amounts of protein were separated by 12% SDS-PAGE and transferred to a polyvinylidene fluoride (PVDF) membrane. Membranes were incubated overnight with antibodies against LTA (#15711; QEDBio), pro-IL1B (#12703; Cell Signaling), and HSP90 (13171-1-AP; Proteintech). Corresponding secondary antibodies conjugated with horseradish peroxidase and in combination with SuperSignal West Pico PLUS Chemiluminescent Substrate (#34580; Thermo Fisher Scientific) were used to visualize bands on a Fusion FX imaging system (Vilber).

## MS-based proteomics—monocytes

Cells were lysed and denatured in 4% SDS dissolved in PBS and processed according to the SP3 method (Hughes et al, 2019), digested with trypsin (SERVA) at a 1:50 enzyme-to-protein ratio and peptides desalted using OASIS HLB 96-well cartridges (Waters, #186001828BA). All samples were analyzed using a 90-min

gradient and data-independent acquisition (DIA) on an Orbitrap Exploris 480 (Thermo Fisher Scientific) that was coupled to an RSLC nano UPLC (Thermo Fisher Scientific).

The MS was operated in DIA mode for single-injection quantitative measurements of individual samples with the following settings: 45k MS1 resolution, MS1 scan range 390–1010 $m/z$, 15k MS2 resolution, AGC target of 1000%, maximum injection time 22 ms, fixed normalized collision energy of 30. In all, 12 $m/z$ precursor isolation windows were used in a staggered window pattern with optimized window placements from 394.4 to 1006.7 $m/z$.

DIA data were analyzed by DIA-NN 1.7.12 (Demichev et al, 2020) using the human fasta-database (UP000005640_9606, downloaded on 190111, 20412 entries). N-terminal methionine excision enabled. All data was filtered to 1% FDR. Cross-run normalization was enabled. Oxidation (M) was used as variable modification, and carbamidomethylation (C) was used as fixed modification. Default parameters were used unless specified otherwise. The unique genes table was used to obtain quantitative data on caspase-4 and caspase-5 protein abundance.

## MS-based LPS analysis

All solvents and water used throughout all experiments were Fisher (Waltham, MA) Optima LC/MS grade. Intact LPS-B5 (lipopoly-saccharide from *E. coli* serotype 055:K59(B5)H⁻, Source strain: ATCC 12014; CDC 5624-50 [NCTC 9701] from InvivoGen) was dissolved in a solution of 50% 2-propanol containing 10 mM ammonium acetate at an estimated LPS concentration of 1 µg/µL.

The LPS was directly infused via an ESI source at a flow rate of 5 µL min⁻¹ into an Orbitrap Exploris 480 (Thermo Fisher Scientific) mass spectrometer. The mass spectrometer was operated in positive ionization mode with a capillary potential of 3 kV. MS1 (600–1800 $m/z$) and MS2 (100–1800 $m/z$) scans were acquired at 60 K resolution. Selected precursor ions were fragmented by stepped Higher Energy Collision Dissociation (@15,30,45,50%). LPS composition assignments and annotations are based on the chemical structure of LPS from *E. coli* as reported previously (Pupo et al, 2021). The supernatant obtained from the "bacterial EVs: isolation and quantification" step was also treated similarly to determine potential LPS contamination. MS/MS spectra are provided in Appendix Fig. S1.

## MS-based proteomics—EVs

### Data-independent acquisition (DIA)

EV samples were lysed and denatured in 4% SDS in PBS and processed following the SP3 protocol (Hughes et al, 2019) as described above. Label-free DIA analyses of peptides were acquired over 120 min by an Orbitrap Exploris 480 (Thermo Scientific) coupled to a 3000 RSLC nano UPLC (Thermo Scientific) from 1 µg of peptides based on Pierce™ Quantitative Peptide Assays & Standards (23275). Samples were loaded on a pepmap trap cartridge (300 µm i.d. x 5 mm, C18, Thermo) with 2% acetonitrile, 0.1% TFA at a flow rate of 20 µL/min. Peptides were separated over a 25 cm analytical column (PepSep C18, 75 µm I.D., 1.5 µm). Solvent A consists of 0.1% formic acid in water. Elution was carried out at a constant flow rate of 250 nL/min within 120 min. Initially, a two-step linear gradient was applied: 3–30% solvent B (0.1% formic

acid in 80% acetonitrile) within 70.5 min, 30–45% solvent B within 13 min, followed by column washing and equilibration. The column was kept at a constant temperature of 50 °C.

The MS was operated in DIA mode for single-injection quantitative measurements of individual samples with the following settings: 60k MS1 resolution, MS1 scan range 375–1600 $m/z$, 15k MS2 resolution, MS2 scan range 120–1600 $m/z$, Normalized AGC target of 1000%, maximum injection time 54 ms, and fixed normalized collision energy of 30. In total, 12 $m/z$ precursor isolation windows with optimized window placements from 400.4319 to 1210.8024 $m/z$.

### DIA data analysis

Raw data analysis was performed using Spectronaut® (Biognosys AG, Zurich, Switzerland) version 19.7.250203.62635 in directDIA+ deep mode with reviewed UniProt databases (S. pyogenes serotype M1 – version 2024_02_05, 1,690 canonical entries). Methionine oxidation and Acetyl (Protein N-term) were set as a variable, and carbamidomethylation on cysteine residues was used as a static modification. The FDR for PSM-, peptide-, and protein-level was set to 0.01. All tolerances were set to dynamic for pulsar searches. All downstream data processing was carried out in R.

## Scanning electron microscopy

Ten μl of sample were layered on 12 mm polylysinated coverslips and washed with PBS. For immunogold labelling, samples were blocked briefly with a BSA/cold water fish gelatin solution (Sigma-Aldrich), and reactions were incubated O/N with a 1:500 mouse anti-LTA antibody (QEDBio) diluted in 1% BSA at 4 °C. After three washes with PBS, the samples were incubated with gold-conjugated goat anti-mouse IgG (Jackson Immuno) for 1 h, washed four times with PBS and fixed in 2.5% glutaraldehyde (Electron Microscopy Sciences). Samples were then fixed in 0.5% osmium-tetroxide (Polysciences Inc), tannic acid (Sigma-Aldrich) and again in osmium-tetroxide. The coverslips were dehydrated in a graded ethanol series, dried in carbon dioxide at critical point and vacuum-coated with 3 nm carbon-platinum (Bal-Tec). Imaging was performed using a LEO 1550 scanning-electron microscope (Zeiss) at a 20 kV acceleration voltage. For topographic analysis, secondary electrons were detected with the in-lens secondary electron (SE) detector, while immunogold labelling was imaged using a detector for backscattered electrons (BSE).

## Transmission electron microscopy

Overall, 10 μl of sample was applied to freshly glow discharged carbon-film-coated copper grids and allowed to adsorb for 6 min. After three washes with distilled water, the grids were contrasted with 4% phospho-tungstic-acid / 1% trehalose, touched on filter paper and air-dried. The grids were examined in a LEO 906 (Zeiss AG, Oberkochen) electron microscope operated at 100 kV and images were recorded with a Morada (SIS-Olympus, Münster) digital camera.

## ASC speck formation

Monocytes were grown in removable 8-well chambers in slide format (ibidi). After treatment, cells were fixed with 4% PFA in PBS overnight at RT, permeabilized with 0.5% Triton X-100 in PBS for 1 min at RT, and non-specific binding was blocked for 30 min using a blocking buffer containing 3% normal donkey serum, 3% cold water fish gelatin, 1% BSA, and 0.05% Tween20 in PBS. Samples were incubated for 1 h at 37 °C with primary antibodies diluted in block buffer (mouse anti-ASC, 1:500 and goat anti-S. pyogenes Group A Carbohydrate antibody 1:1000). After washing, samples were incubated with the fluorophore-conjugated secondary antibodies a-mouse-Cy™2 and a-goat-Cy™5 both at 15 μg/mL. Hoechst 33342 at 5 μg/mL was added for DNA staining for 45 min at 37 °C. After washing, the removable silicon chambers were removed, and coverslips were mounted using Mowiol. Slides were imaged with a Zeiss AxioScan Z1 slide scanner (Zeiss, Jena, Germany) at ×20 magnification. Images were quantified using QuPath (Bankhead et al, 2017). Selected samples were imaged in more detail using an SP8 confocal microscope (Leica). Image series were prepared as collapsed confocal stacks using Arivis software.

## Statistical analysis

Statistical tests were performed with GraphPad Prism 10.0.0 software, except for the RNA sequencing data. All quantitative data represent mean and standard deviation (SD). Figure legends indicate the number of biological replicates. Student's $t$ test was applied for pairwise comparisons. One-way or two-way ANOVA with Holm–Šídák's multiple comparisons test were used for comparisons with more than two groups or for comparisons between groups with more than one variable, respectively. A $P$ value of $<0.05$ or $<0.01$ (RNAseq) was defined as statistically significant. No blinding procedures were implemented in any of the experiments presented.

## Data availability

The datasets produced in this study are available in the following databases: The code developed to analyze the RNA sequencing datasets of monocytes and BLaER1 cells has been deposited at https://github.com/MPUSP/human_RNA_seq_pipeline/releases/tag/v1.0.0. The raw RNA sequencing datasets are available under the NCBI-BioProject accession number: PRJNA753844. The mass spectrometry proteomics data have been deposited to the ProteomeXchange Consortium via the PRIDE (Perez-Riverol et al, 2022) partner repository (https://www.ebi.ac.uk/pride/profile/reviewer_pxd046570) with the dataset identifier PXD046570. Confocal images for ASC speck formation were deposited on BioImage Archive (Accession number: S-BIAD1725).

The source data of this paper are collected in the following database record: biostudies:S-SCDT-10_1038-S44319-025-00558-7.

## Peer review information

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

## Acknowledgements

We thank Karin Hahnke for measuring RNA concentrations with the Bioanalyzer and Dominik Rigo for measuring the protein content of EVs using the Bradford assay. We acknowledge the Max Planck Institute for Molecular Genetics for library preparation and RNA sequencing of the monocyte and BLaER1 samples. We thank Thomas Graf for providing BLaER1 wild-type cells. We acknowledge Knut Finstermeier for assistance with data analysis. We are much obliged to Thomas F Wulff for assistance in generating the *S. pyogenes slo* deletion strain. We are grateful to the members of the Charpentier Lab for their critical reading and discussion of the manuscript. This work was supported by the Max Planck Society (to EC), the German Research Foundation (DFG, Leibniz Prize to EC), Louis-Jeantet Foundation (Louis-Jeantet Price to EC), and the Tang Prize Foundation (Tang Prize to EC).

## Author contributions

**Kathrin Krause**: Conceptualization; Formal analysis; Investigation; Visualization; Methodology; Writing—original draft; Project administration; Writing—review and editing. **Sandra Franch Arroyo**: Conceptualization; Formal analysis; Investigation; Visualization; Methodology; Writing—original draft; Project administration; Writing—review and editing. **Matteo Ugolini**: Investigation; Methodology. **Tonya Kueck**: Investigation; Methodology. **Timothy J Sullivan**: Formal analysis; Visualization. **Eric J C Gálvez**: Formal analysis; Visualization. **Matthias Muenzner**: Formal analysis; Investigation. **Christian Goosmann**: Investigation; Visualization. **Volker Brinkmann**: Formal analysis; Investigation; Visualization. **Christian K Frese**: Formal analysis; Investigation. **Kathirvel Alagesan**: Formal analysis; Investigation; Visualization. **Tim Vierbuchen**: Resources. **Holger Heine**: Resources. **Ulrike Resch**: Conceptualization; Resources; Formal analysis; Supervision; Investigation; Writing—review and editing. **Leif E Sander**: Resources; Supervision; Funding acquisition; Writing—review and editing. **Emmanuelle Charpentier**: Conceptualization; Supervision; Funding acquisition; Writing—review and editing.

Source data underlying figure panels in this paper may have individual authorship assigned. Where available, figure panel/source data authorship is listed in the following database record: biostudies:S-SCDT-10_1038-S44319-025-00558-7.

## Funding

## Disclosure and competing interests statement

The authors declare no competing interests.

# Expanded View Figures

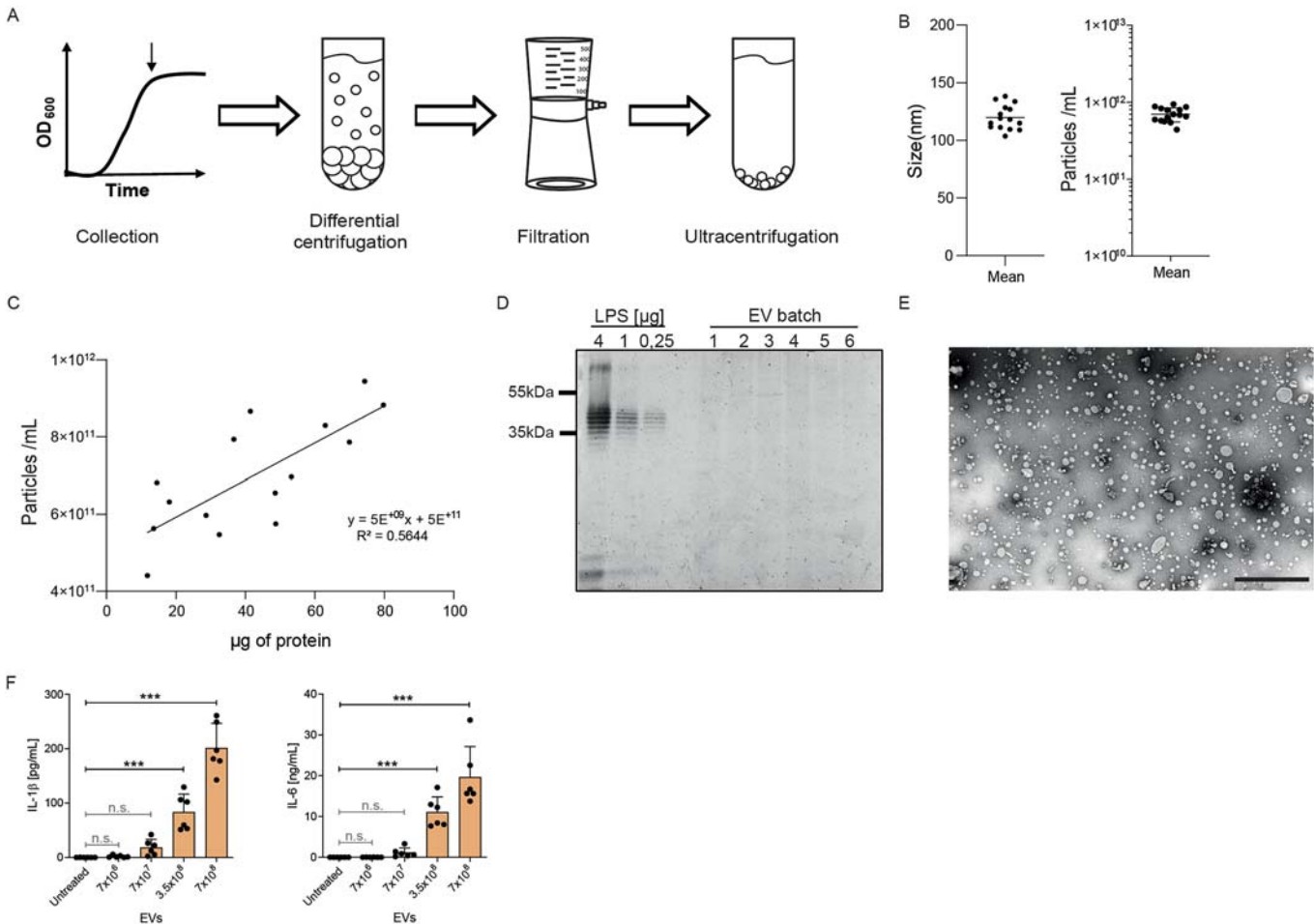

**Figure EV1. Purification, quantification, and characterization of bacterial EVs used in this study.**

(**A**) Schematic protocol of the purification strategy used for EVs. (**B**) Mean size (left) and concentration (right) of *Spy* EVs ($n = 15$) characterized by nanoparticle tracker analysis (NTA). (**C**) Linear correlation between particles/mL (measured by NTA) and the amount of protein of the same batches (Bradford assay, $n = 15$). (**D**) LPS analysis of EV preparations separated by SDS PAGE followed by Pro-Q Emerald 300 staining. Smooth LPS standard from *E. coli* serotype O55:B5 with characteristic ladder pattern was used as positive control. (**E**) *Spy* EV preparation imaged by transmission electron microscopy (TEM, scale bar: 1 μm). (**F**) IL-1β and IL-6 released by human monocytes stimulated with increasing amounts of *Spy* EVs for 18 h. Bars represent the mean ± SD of six biological replicates. *P < 0.0001* (for all comparisons). Data information: (**F**) One-way ANOVA was applied with Holm–Šídák correction for multiple comparisons. ***$P \le 0.001$, n.s. not significant.

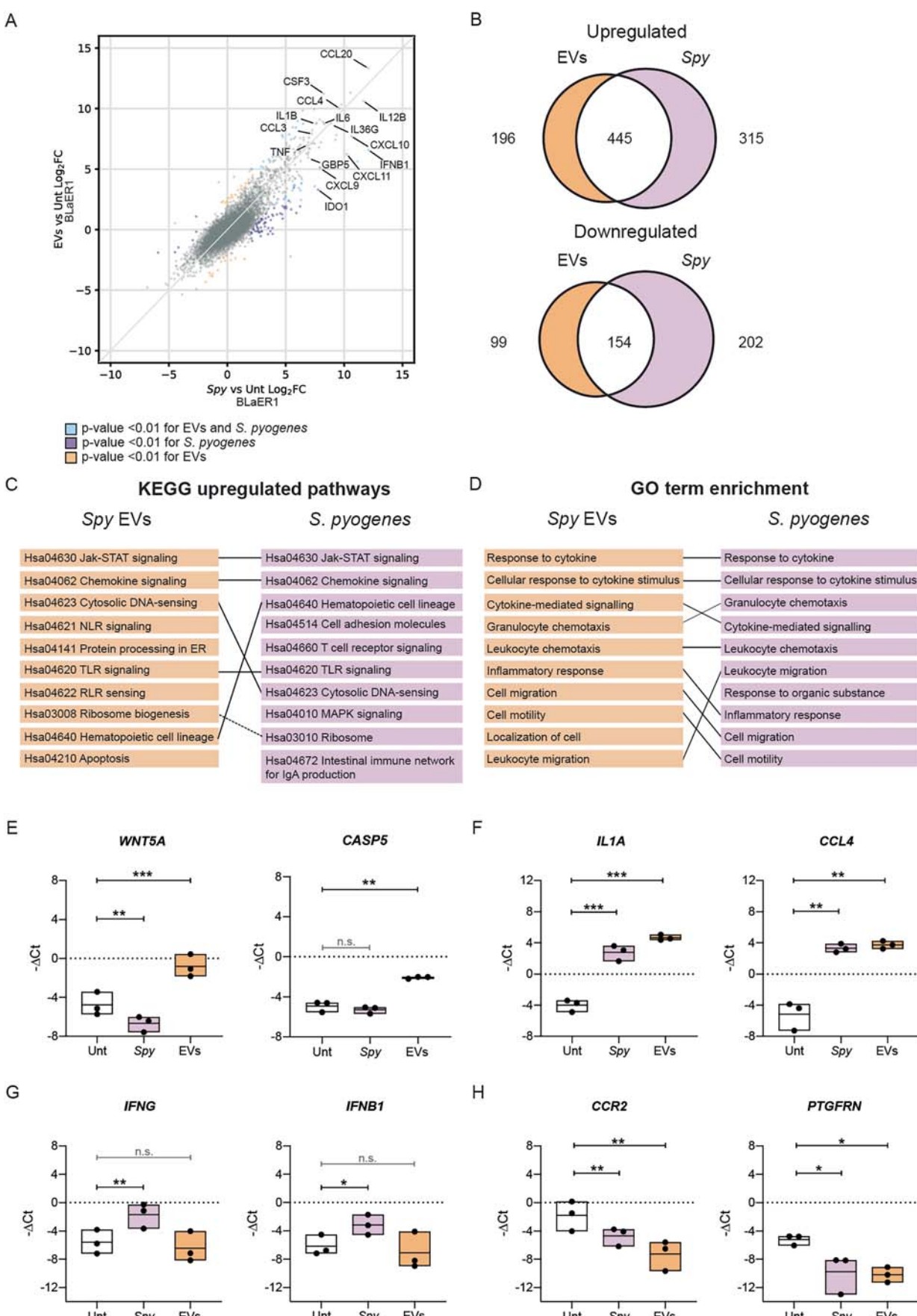

**Figure EV2.   RNA sequencing of monocytes: pathway analysis and RTqPCR of specific genes.**

(A) Scatterplot comparison of $Log_2$ fold change (FC) (treatment vs untreated control, Unt) in transcript abundance for *S. pyogenes*-infected (x-axis) or EV-treated samples (y-axis) in BLaER1 cells (related to Fig. 1). Grey dots represent transcripts commonly regulated but with an FC < |2|. Blue dots denote commonly regulated transcripts with a FC ≥ |2|. Orange and purple dots indicate differentially expressed genes for either *Spy* EVs or *S. pyogenes*, respectively. (B) Venn diagrams of differentially transcribed genes in BLaER1 cells after stimulation with *Spy* EVs or *S. pyogenes* for 4 h. Numbers indicate the total amount of genes for each category. (C, D) Top ten upregulated KEGG pathways (C) or GO terms (D) are shown for *Spy* EV-treated versus untreated monocytes and *S. pyogenes*-treated versus untreated monocytes. Solid lines indicate same pathway and dashed lines indicate a related pathway. (E–H) Quantitative real time PCR of selected genes. Floating bar plots displaying the negative ΔCt values of three biological replicates. The central line represents the mean, the borders represent the minimum and the maximum values. A dotted line indicates the mean expression of the housekeeping genes (*GAPDH/TUBB*). (E) Genes upregulated by *Spy* EV treatment. (*WNT5A*) $P = 0.0029$, $P = 0.0004$. (*CASP5*) $P = 0.0028$. (F) Genes upregulated by *S. pyogenes* and *Spy* EV treatments. (*IL1A*) $P = 0.0003$ (for both comparisons). (*CCL4*) $P = 0.0014$ (for both comparisons). (G) Genes upregulated by *S. pyogenes* treatment. (*IFNG*) $P = 0.0025$. (*IFNB*) $P = 0.0308$. (H) Genes downregulated for both *S. pyogenes* and *Spy* EV treatments. (*CCR2*) $P = 0.0055$ (*Spy*), $P = 0.0010$ (EVs). (*PTGRFN*) $P = 0.0130$ (for both comparisons). Data information: (E–H) One-way ANOVA with Holm–Šídák correction for multiple comparisons was applied for statistical analyses. *$P \le 0.05$, **$P \le 0.01$, ***$P \le 0.001$, n.s. not significant.

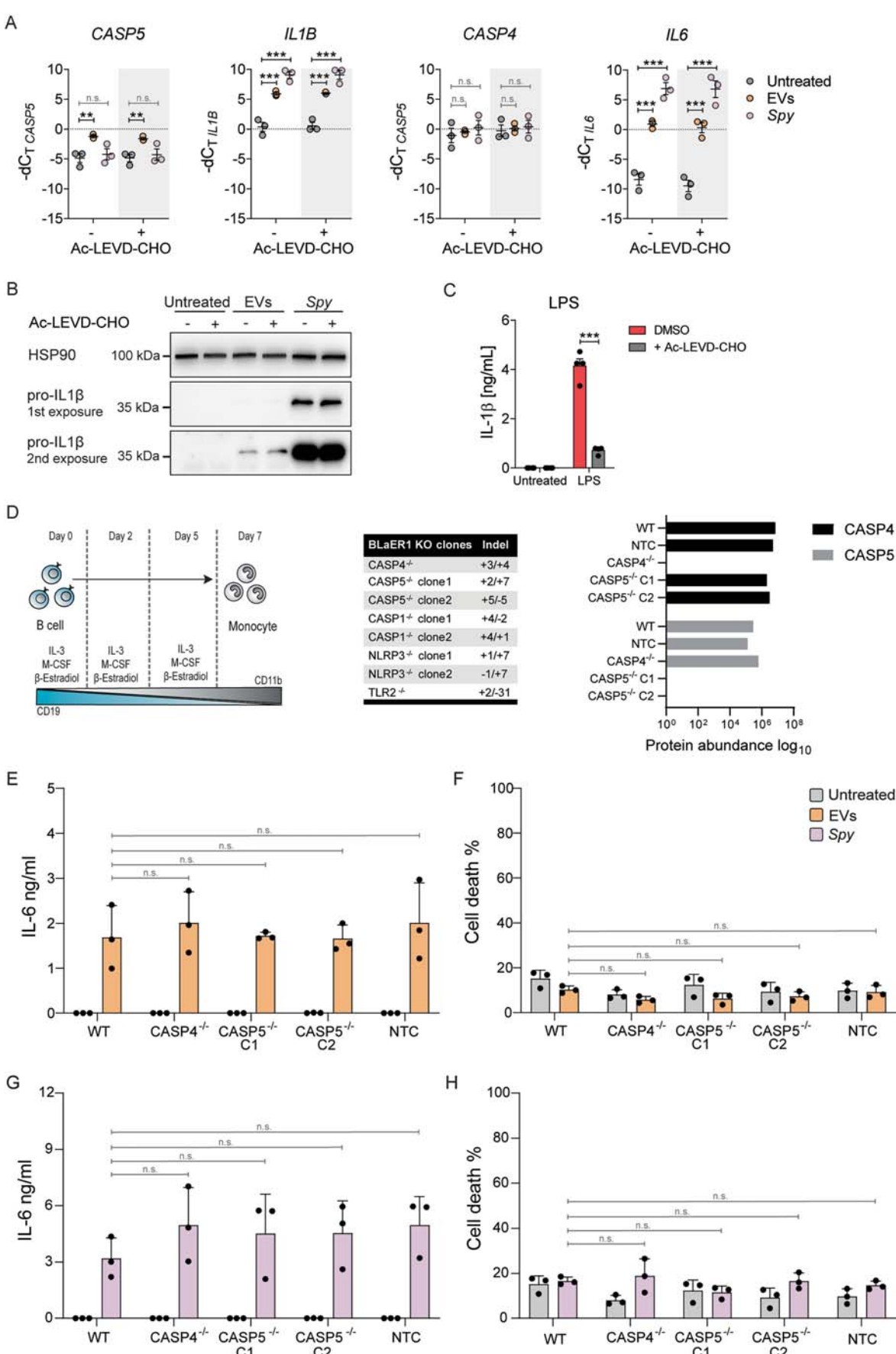

**Figure EV3.  BLaER1 knock-out characterization.**

(A) Quantitative real time PCR of selected genes in human monocytes. Shown are mean ± SD of three biological replicates. A dotted line indicates the mean expression of the housekeeping genes (*GAPDH/TUBB*). (*CASP5*) P = 0.0013 (−), P = 0.0018 (+). (*IL1B*) P = 0.0006 (EVs for both comparisons), P = 0.0001 (*Spy*, for both comparisons). (*IL6*) P = 0.0001 (EVs for both comparisons), P < 0.0001 (*Spy*, for both comparisons). (B) Immunoblot analysis of pro-IL-1β in human monocytes. Cells were either left untreated or preincubated with the caspase-4/-5 inhibitor Ac-LEVD-CHO. Representative of three biological replicates. (C) IL-1β in response to LPS in supernatants from human monocytes either left untreated or preincubated with Ac-LEVD-CHO at 18 h. Shown are mean ± SD of three biological replicates. P < 0.0001 (D) Graphical representation of the differentiation protocol of BLaER1 cells as well as CRISPR-Cas9 mutagenesis of BLaER1 cells. The existence of frameshift mutations on each allele of the $CASP4^{-/-}$, $CASP5^{-/-}$, $TLR2^{-/-}$, $CASP1^{-/-}$, and $NLRP3^{-/-}$ clones was assessed using next generation sequencing. The guide RNAs used for genome editing are shown in the Reagents and Tools Table. Mass spectrometry analysis was used to confirm the absence of caspase-4 and caspase-5 from their respective KO clones compared to wild-type or non-targeted control (NTC) BLaER1 cells. (E–H) IL-6 and LDH released from BLaER1 WT, caspase-4 KO ($CASP4^{-/-}$), caspase-5 KO clones 1 and 2 ($CASP5^{-/-}$ C1 or C2), and non-targeted cells (NTC). BLaER1 cells were left untreated or were stimulated for 18 h with either *Spy* EVs or *S. pyogenes*. The percentage of LDH released from the positive control is shown as a measure of cell death. Bars represent the mean ± SD of three biological replicates. Data information: (A, C, E–H) Two-way ANOVA with Holm–Šídák correction for multiple comparisons was applied for statistical analyses. **P ≤ 0.01, ***P ≤ 0.001, n.s. not significant.

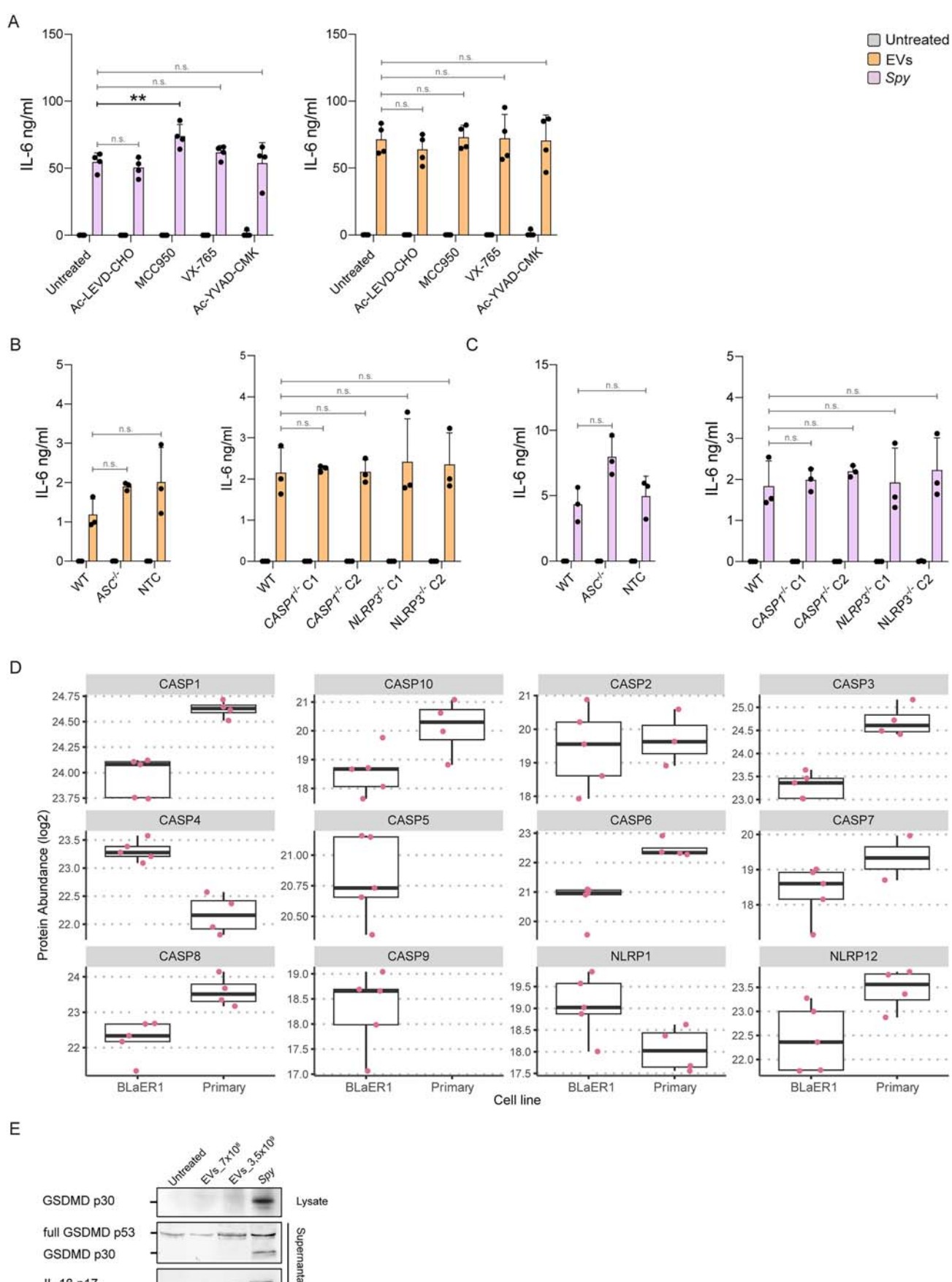

**Figure EV4. Cytokine controls and cell death of monocytes and BLaER1 cells: canonical inflammasome.**

(A) IL-6 in supernatants from human monocytes either left untreated or preincubated with Ac-LEVD-CHO, MCC950, VX-765, or Ac-YVAD-CMK. Cells were then left unstimulated or challenged with *S. pyogenes* or its EVs for 18 h. Bars represent the mean ± SD of four biological replicates. *P* = *0.0011*. (B, C) IL-6 released from BLaER1 WT, PYCARD KO (*ASC*$^{-/-}$), NLRP3 KO clones 1 and 2 (*NLRP3*$^{-/-}$), Caspase-1 KO clones 1 and 2 (*CASP1*$^{-/-}$), and non-targeted control (NTC) cells stimulated with *Spy* EVs or infected with *S. pyogenes* for 18 h. Bars represent the mean ± SD of three biological replicates. (D) Box plot showing caspase and NLRP protein levels differentially expressed in BLaER1 cells (5 biological replicates) and primary human monocytes (4 biological replicates). Each dot represents the protein level of an individual sample. The boxplot displays the distribution of the data with the box representing the interquartile range (IQR) between the 25th (Q1) and 75th (Q3) percentiles. The line inside the box indicates the median (50th percentile). Whiskers extend to the smallest and largest values within 1.5 × IQR below Q1 and above Q3, respectively. Data points outside this range are shown individually as outliers. In total, 6472 proteins were quantified in at least 3 biological replicates of BLaER1 cells, and 6092 proteins were quantified in primary monocytes. (E) Immunoblots displaying GSDMD and IL-1β in human monocytes either left untreated or challenged with *Spy* EVs or *S. pyogenes* for 18 h. Representative of four biological replicates. Data information: (A–C) Two-way ANOVA with Holm–Šídák correction for multiple comparisons was applied for statistical analyses. **$P \leq 0.01$, n.s. not significant.

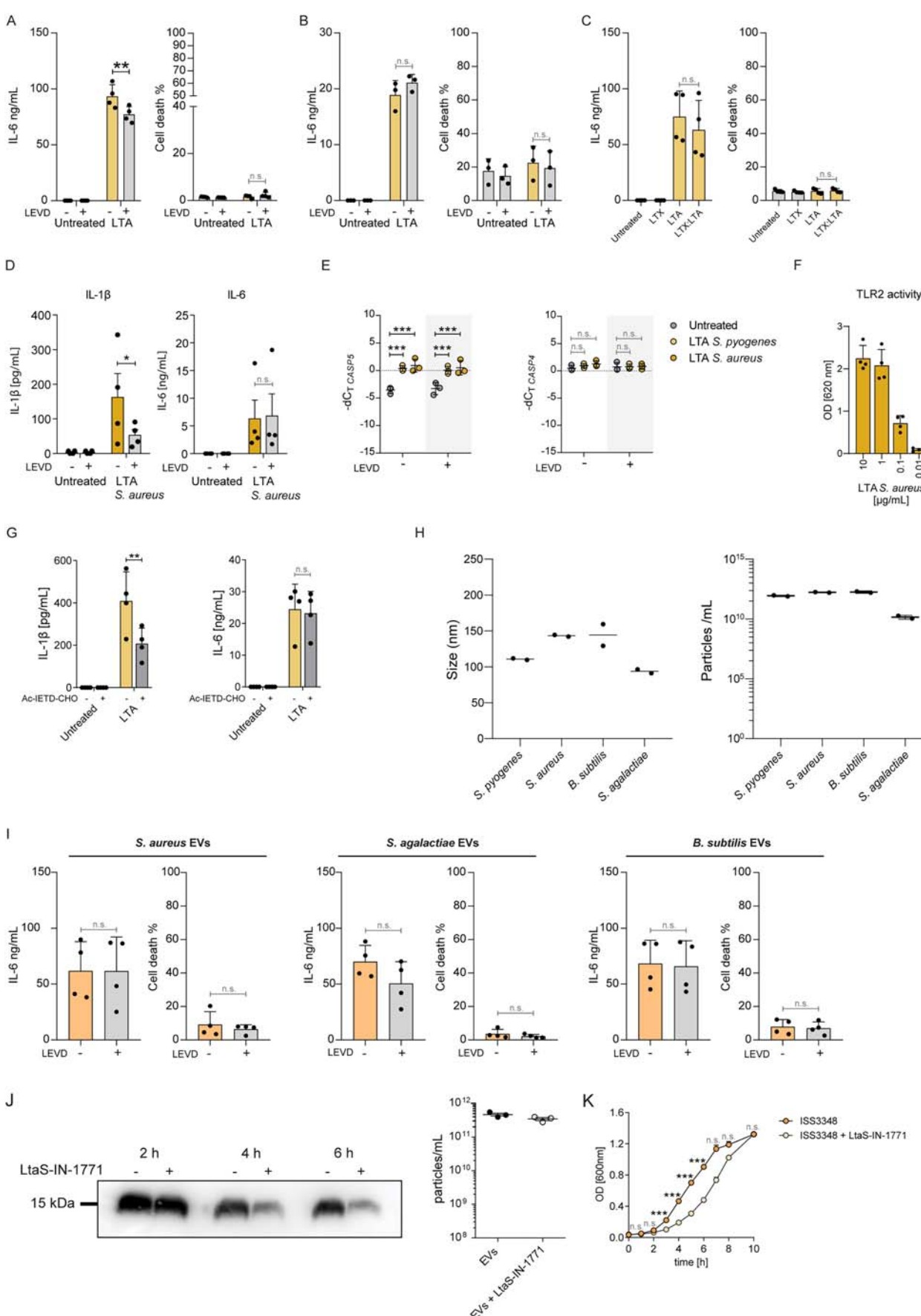

**Figure EV5. Cytokine controls and cell death of monocytes and BLaER1 cells: role of LTA in EV-dependent monocyte activation.**

(A) IL-6 and LDH release at 18 h from monocytes that were left untreated or incubated with Ac-LEVD-CHO before the addition of *S. pyogenes* LTA. Shown are mean ± SD of four biological replicates. *P* = 0.0069. (B) IL-6 and LDH release at 18 h from BLaER1 cells that were left untreated or incubated with Ac-LEVD-CHO prior to addition of purified *S. pyogenes* LTA. Shown are mean ± SD of three biological replicates. (C) IL-6 and LDH release at 18 h from monocytes that were left untreated or treated with *S. pyogenes* LTA, either directly applied (LTA) or transfected using Lipofectamine LTX (LTx:LTA). Shown are mean ± SD of four biological replicates. (D) IL-1β and IL-6 release at 18 h from monocytes that were left untreated or incubated with Ac-LEVD-CHO before the addition of *S. aureus* LTA. Shown are mean ± SD of four biological replicates. *P* = 0.0374. (E) Quantitative real time PCR of *CASP5* and *CASP4*. Shown are mean ± SD of three biological replicates. A dotted line indicates the mean expression of the housekeeping genes (*GAPDH/TUBB*). *P* < 0.0001 (for all comparisons). (F) TLR2 activity in HEK-Blue TLR2 cells after stimulation with *S. aureus* LTA for 18 h. Baseline represents untreated cells. Bars represent the mean ± SD of four biological replicates. (G) IL-1β and IL-6 released by human monocytes that were either left untreated or preincubated with the caspase-8 inhibitor Ac-IETD-CHO. Cells were then left unstimulated or treated with *S. pyogenes* LTA for 18 h. Bars represent the mean ± SD of four biological replicates. *P* = 0.0066. (H) Mean size and particles/mL of *S. pyogenes*, *S. aureus*, *B. subtilis*, and *S. agalactiae* EVs (*n* = 2). (I) IL-6 and LDH release from monocytes either left untreated or preincubated with Ac-LEVD-CHO before the addition of EVs from *S. aureus*, *S. agalactiae*, or *B. subtilis* for 18 h. Shown are mean ± SD of four biological replicates. (J) Immunoblot analysis of LTA in bacterial pellets treated −/+ LtaS-IN-1771 and particles/mL of corresponding *S. pyogenes* EVs. Representative image of 2 biological replicates is shown. (K) *S. pyogenes* growth in the presence of LtaS-IN-1771 in THB media. Shown are mean ± SD of three biological replicates. (3 h) *P* = 0.0010, (4h-8h) *P* < 0.0001. Data information: (ABDEGK) Two-way ANOVA was applied with Holm–Šídák correction for multiple comparisons. (C) One-way ANOVA was applied with Holm–Šídák correction for multiple comparisons. (I–H) Statistical significance was assessed using paired *t* tests. *\*P* ≤ 0.05, *\*\*P* ≤ 0.01, *\*\*\*P* ≤ 0.001, n.s. not significant.

