## [Peer Review File · EMBO Reports]

Streptococcus pyogenes EVs induce the alternative inflammasome via Caspase-4/-5 in human monocytes

Kathrin Krause, Sandra Franch Arroyo, Matteo Ugolini, Tonya Kueck, Timothy Sullivan, Eric Gálvez, Matthias Muenzner, Christian Goosmann, Volker Brinkmann, Christian Frese, Kathirvel Alagesan, Tim Vierbuchen, Holger Heine, Ulrike Resch, Leif Sander, and Emmanuelle Charpentier

Corresponding author(s): Emmanuelle Charpentier (research@emmanuelle-charpentier.org), Kathrin Krause (krause@mpusp.mpg.de), Ulrike Resch (ulrike.resch@meduniwien.ac.at)

Review Timeline:

Submission Date:	9th Aug 24
Editorial Decision:	16th Sep 24
Appeal Received:	18th Oct 24
Editorial Decision:	22nd Oct 24
Revision Received:	6th Apr 25
Editorial Decision:	3rd Jun 25
Revision Received:	18th Jul 25
Accepted:	31st Jul 25

Editor: Achim Breiling

Transaction Report:

Dear Prof. Charpentier,

Thank you for the submission of your manuscript to EMBO reports. I have now received the reports from the three referees that were asked to evaluate your study, which can be found at the end of this email.

As you will see, all referees have major concerns and note substantial technical and experimental shortcomings, in particular referee #1. Moreover, referees #1 and #3 indicate novelty concerns.

Given the comments of the referees, the amount of work required to address them, and the fact that EMBO reports can only invite revision of papers that receive overall positive support from all the referees upon initial assessment, I cannot offer to publish your manuscript.

I am sorry to have to disappoint you this time. I nevertheless hope that the referee comments will be helpful in your continued work in this area, and I thank you once more for your interest in our journal.

Yours sincerely

Referee #1:

The study investigates how human monocytes recognize and respond to extracellular vesicles derived from Gram-positive bacteria, specifically *Streptococcus pyogenes*. While the manuscript explores an interesting aspect of the immune response, I have three major concerns that I believe significantly impact the manuscript's suitability for publication.

1. Lack of Novelty: The authors aim to describe the activation of the inflammasome via TLR2, NLRP3, and caspase 4/5 in response to Gram-positive bacterial EVs. However, the potential of Gram-positive EVs to activate macrophages (PMID: 34253061, 36978183, 34959485), TLR2 (PMID: 34281154, 36077142), and the inflammasome (PMID: 31988111) has already been documented in the literature. As such, the novelty of this study is limited, and it does not significantly advance our current understanding of these processes.

2. Inadequate Isolation and Characterization of EVs: The isolation and characterization of EVs in this study do not meet the latest guidelines (MISEV 2023; PMID: 38326288). The methods used do not ensure the purity of the EV preparations, which undermines the validity of the conclusions drawn by the authors. Specifically, the lack of TEM imaging to confirm the presence of only EVs and the absence of other extracellular particles is a critical omission. Moreover, the authors have not demonstrated the presence of intact spherical nanostructures, which is essential for confirming the identity of the isolated EVs. Performing EV isolations with additional techniques (i.e., SEC or density gradients), characterizing them and repeating key experiments would be beyond the scope of a major revision.

The use of bacterial culture media, which is a known source of extracellular particles, raises further concerns. Given that the authors isolated EVs via ultracentrifugation, they cannot exclude the possibility of co-isolating contaminating particles from the media. To address this issue, the authors would have to repeat key experiments using EV preparations from plain bacterial media (without bacterial cultivation) to assess the immunogenicity of the media components. Only by doing so can they conclusively determine that the observed immune responses are due to the EVs and not contaminating particles.

3. Inappropriate Dosing: The dosing of EVs applied to human monocytes in this study is excessively high. For example, the authors applied 7×10^8 particles to 1×10^5 monocytes in a 96-well format, resulting in a ratio of 7,000 EVs per cell. This number was even higher (10 times more) for RNA isolation experiments, and for BLaER1 cells, the ratio was 35,000 EVs per cell. These doses are far beyond physiological EV ratios, making it unsurprising that the cells responded to the EVs.

Furthermore, there is a lack of justification for the comparison of infection with an MOI of 5 to stimulations with 7×10^8 particles for monocytes and an MOI of 1 with 3.5×10^{10} particles for BLaER1 cells. The manuscript does not provide sufficient rationale for these dosing choices, which raises concerns about the validity of the conclusions drawn.

Given these significant concerns, I suggest that the manuscript be rejected. It does not meet the high standards expected of this journal, lacks novelty, and fails to adhere to the standards of the EV community in terms of isolation and characterization.

Referee #2:

The study investigates the immune response to *Streptococcus pyogenes* (Spy) extracellular vesicles (EVs) and their parental cells in human monocytes, revealing distinct transcriptional signatures and pathways activated by each. The research demonstrates that Spy EVs trigger IL-1 β release through the alternative inflammasome pathway involving TLR2, NLRP3, and caspases-4 and -5, with caspase-4/-5 activation partly dependent on lipoteichoic acid (LTA) from Gram-positive bacterial membranes. Additionally, the study highlights that caspase-4/-5 also plays a role in IL-1 β production in response to EVs from other Gram-positive species, indicating a conserved immune mechanism.

Comments:

- 1) The purification strategy, size, and abundance of the bacterial EVs used in this study are detailed in Figures EV1A and B. EV quantification was carried out using Nanoparticle Tracking Analysis (NTA), which demonstrated higher reproducibility compared to protein quantification. Although NTA provided size measurements of EVs ranging from 100-150 nm, to better illustrate any heterogeneous features in the EV population and to visualise actual ultrastructural characteristics of the EVs, it is advisable that the authors also include transmission electron microscopy (TEM) analysis and show electron micrographs.
- 2) The authors suggest that EVs isolated from Spy and the Spy bacterial cell itself differentially modulate inflammasome responses. However, have the authors directly compared the effects of EV-associated LTA with freely released LTA on immune cells? It would be beneficial to compare the key effects of EV-associated LTA with those of culture supernatant after vesicle preparation (containing non-vesicle-associated LTA) to exclude the influence of other surface structures present on the bacteria. Additionally, what proportion of the total released LTA is associated with EVs?
- 3) Pathway analysis of primary monocytes revealed a high similarity between KEGG pathways and GO terms for each treatment, with common upregulation of cytokines (IL1A, IL1B, IL6), chemokines (CCL3, CCL4, CXCL8), and growth factors (CSF3), and downregulation of immune receptors (TLR1, TLR6, CCR2) and adhesion factors (PECAM1, VCAN) across both stimuli. While cytokines such as IL1 and IL6 are upregulated, CCR2 levels are significantly downregulated. Given that CCR2 is the receptor for the chemokine CCL2 (MCP-1), it would be logical to assess the levels of CCL2 in the sample, as the CCR2-CCL2 pathway is directly and indirectly linked to IL1B and IL6 signalling. However, based on the information provided, it is unclear if the authors have assessed CCL2 (MCP-1) levels.
- 4) Could you explain why the metalloendopeptidase STEAP4 is induced in macrophages treated with extracellular vesicles (EVs) but not when treated with whole bacterial cells, even though both treatments increase IL-6 and IL1B levels, which are known regulators of STEAP4?
- 5) Did the authors examine whether IL-1 β release in their study occurs through gasdermin D pores, as typically observed in response to LPS stimulation, or if it might be mediated by some alternative pathway(s)? Specifically, have they considered analyzing gasdermin D activation, or explored the possibility of IL-1 β secretion through non-canonical pathways, such as microvesicle shedding, exosomes, or secretory lysosomes? Additionally, have they investigated the potential for IL-1 β release via inflammasome-independent or autophagy-dependent mechanisms?
- 6) Discussion: The discussion is well-structured and comprehensive, providing a detailed analysis of the differential immune responses elicited by *Streptococcus pyogenes* and its extracellular vesicles (EVs) in human monocytes. The study's focus on TLR2 and caspase-4/-5 as key mediators adds valuable insight into the innate immune response to this pathogen. The comparison with existing literature, particularly the novel findings related to EVs, highlights the contribution of this study to the field. The discussion effectively addresses the complexity of immune responses, acknowledging the dual role of inflammation in both protecting against bacterial invasion and preventing excessive immune responses. However, there are some areas where the discussion could be strengthened. The exploration of the mechanistic differences between *S. pyogenes* and its EVs is somewhat limited, leaving a gap in understanding how these differences translate into specific immune responses. The speculative nature of the role of EVs in disease progression could be supported by more detailed experimental proposals. Moreover, the clinical implications of the findings, particularly regarding sepsis, could be further elaborated to clarify their potential impact. Finally, the roles of TLR2 and caspase-4/-5, though highlighted, could be better defined in terms of their biological significance under different conditions, making the discussion more precise and actionable.

Referee #3:

The manuscript by Franch-Arroyo et al presents results demonstrating the activation of NLRP3 non-canonical inflammasome by EVs released from Gram-positive *Streptococcus pyogenes*. The results are interesting though activation by EVs from Gram-positive *Staphylococcus* has been demonstrated previously (Wang et al., PNAS 2020). Consequently, the novelty of the results cannot be unambiguously established. I also have a few other comments which the authors will hopefully find useful.

1. Fig. EV3 - The levels of IL-1b secretion are too low even at the highest concentration of EVs used here. Two of the data points (out of 4) have values around approx. 50 pg/mL only. Thus, how reproducible the data is needs to be made clear.
2. Related to the above, how are Fig. EV3 and Fig. 2A different so as to result in drastically different levels of IL-1b secretion?
3. It is important to show Casp-5 expression by WB. Can the authors also show Casp-5 cleaved form with Spy EVs?
4. Immunoblot showing pro-IL-1b expression should be moved to main figures. Do the authors also see cleaved IL-1b expression by EVs.
5. Experiment demonstrating the requirement (or not) of potassium flux should be conducted.

6. Line 225 - the description of results examining the role of RIP1K can be improved.
7. Line 241 - "infection with MCC950 and ..." needs to be rephrased for clarity.
8. Fig. 5F - The authors must expand further why LTA transfection wouldn't result in lower IL-1b levels as compared to direct LTA stimulation, particularly as LTA is packaged within Lipofectamine and may not be fully accessible for TLR2 stimulation.
9. Further assays such as ASC oligomerisation may be conducted to fully validate the results obtained.

** As a service to authors, EMBO Press provides authors with the ability to transfer a manuscript that one journal cannot offer to publish to another journal, without the author having to upload the manuscript data again. To transfer your manuscript to another EMBO Press journal using this service, please click on Link Not Available

We thank the editor and reviewers for reading and evaluating our manuscript. We would like to take the opportunity to respond to the constructive reviewer's comments, as we do not feel that a rejection of our manuscript is entirely justified. Please find our point-by-point responses below.

Referee #1:

1. Lack of Novelty: The authors aim to describe the activation of the inflammasome via TLR2, NLRP3, and caspase 4/5 in response to Gram-positive bacterial EVs. However, the potential of Gram-positive EVs to activate macrophages (PMID: 34253061, 36978183, 34959485), TLR2 (PMID: 34281154, 36077142), and the inflammasome (PMID: 31988111) has already been documented in the literature. As such, the novelty of this study is limited, and it does not significantly advance our current understanding of these processes.

We politely disagree with the reviewer's assessment that there is lack of novelty. While the activation of macrophages by Gram-positive EVs has been studied more extensively, there are currently no publications examining the alternative pathway of inflammasome activation in human monocytes in response to Gram-positive bacterial species or their derived EVs. The studies cited by the reviewer have in fact all been carried out in macrophages. For human monocytes, however, there are only 6 publications to date describing the alternative inflammasome: Netea et al. 2009¹ showed that monocytes can release processed IL-1 β in a 1-step pathway after stimulation by TLR2 and TLR4 using Pam3Cys, LPS, or heat-killed *S. epidermidis*. Then, Gaidt et al. 2016² elegantly characterized the LPS-induced alternative inflammasome pathway, which is a particular feature of human monocytes and signals through the TLR4-TRIF-RIPK1-FADD-CASP8 axis to activate CASP1. A role for CASP8 in LPS-induced IL-1 β secretion by human monocytes was also reported by Oliva-Martin et al. 2016³. The study by Gritsenko et al. 2020⁴ corroborated the previous findings by demonstrating that priming is dispensable for NLRP3 inflammasome activation in human monocytes but not in macrophages, underlining the uniqueness of inflammasome activation in human monocytes. Gao et al. 2023⁵ also demonstrated the activation of the alternative inflammasome pathway and its dependency on CASP8 in response to different heat-inactivated Gram-negative bacteria. Finally, using different TLR ligands, Unterberger et al. 2023⁶ provided evidence that aside from TLR4, various TLRs can engage the alternative inflammasome in human monocytes. However, none of these publications investigated activation of the alternative inflammasome in the presence of live bacteria or their derived EVs.

We apologize if this was not understandable in the manuscript and we can certainly work on the text to make it clearer. In addition, the contribution of the non-canonical inflammasome to Gram-positive infection has only been demonstrated in mice for *S. aureus* and *L. monocytogenes*. However, no study to date has reported a role for caspase-4/-5 during Gram-positive infection in human cells. Furthermore, the involvement of caspase-4/-5 in alternative inflammasome activation has only been reported for LPS^{7,8}.

2. Inadequate Isolation and Characterization of EVs: The isolation and characterization of EVs in this study do not meet the latest guidelines (MISEV 2023; PMID: 38326288). The methods used do not ensure the purity of the EV preparations, which undermines the validity of the conclusions drawn by the authors. Specifically, the lack of TEM imaging to confirm the presence of only EVs and the absence of other extracellular particles is a critical omission. Moreover, the authors have not demonstrated the presence of intact spherical nanostructures, which is essential for confirming the identity of the isolated EVs. Performing EV isolations with additional techniques (i.e., SEC or density gradients), characterizing them and repeating key experiments would be beyond the scope of a major revision.

The use of bacterial culture media, which is a known source of extracellular particles, raises further

concerns. Given that the authors isolated EVs via ultracentrifugation, they cannot exclude the possibility of co-isolating contaminating particles from the media. To address this issue, the authors would have to repeat key experiments using EV preparations from plain bacterial media (without bacterial cultivation) to assess the immunogenicity of the media components. Only by doing so can they conclusively determine that the observed immune responses are due to the EVs and not contaminating particles.

The reviewer has raised an important point and we believe that the suggestion to use “EV preparations” from plain bacterial media as control to test whether there is any effect on the monocytes is very feasible and would strengthen our claims. We are currently conducting experiments using this control. The isolation and characterization of *S. pyogenes* EVs derived from strain ISS3348 (also used in this study) were published by our laboratory in 2016⁹, where EVs were visualized via SEM, TEM, and STED microscopy. While we acknowledge that our EV preparations via ultracentrifugation are not very pure, we still believe that our conclusions regarding LipoTeichoic Acid (LTA) being the major PAMP that drives alternative inflammasome activation in human monocytes are valid, as we were able to demonstrate IL-1 β secretion from human monocytes also using LTA purified from *S. pyogenes* and *S. aureus* (ultrapure LTA). Furthermore, inhibition of LTA synthesis in our bacterial cultures significantly reduces the immunogenic properties of the derived EV preparations, despite the presence of potential co-isolated contaminants from the media. Nevertheless, we are ready to repeat the key experiments of this manuscript with a new preparation of *S. pyogenes* EVs.

3. Inappropriate Dosing: The dosing of EVs applied to human monocytes in this study is excessively high. For example, the authors applied 7e8 particles to 1e5 monocytes in a 96-well format, resulting in a ratio of 7,000 EVs per cell. This number was even higher (10 times more) for RNA isolation experiments, and for BLaER1 cells, the ratio was 35,000 EVs per cell. These doses are far beyond physiological EV ratios, making it unsurprising that the cells responded to the EVs.

Furthermore, there is a lack of justification for the comparison of infection with an MOI of 5 to stimulations with 7e8 particles for monocytes and an MOI of 1 with 3.5e10 particles for BLaER1 cells. The manuscript does not provide sufficient rationale for these dosing choices, which raises concerns about the validity of the conclusions drawn.

We apologize for this oversight. All bacterial infections for primary monocytes as well as BLaER1 cells were carried out with an MOI 5. We will correct the corresponding “Methods” section.

The dose of EVs for stimulation of human monocytes was originally assessed by comparing the IL-6 response against increasing concentrations of EVs with different concentrations of purified *S. pyogenes* LTA, where 7x10⁸ EV particles induced a similar response as 1 μ g/mL LTA. The IL-1 β response of alternative inflammasome activation has been reported to be lower compared to the canonical/non-canonical inflammasome², consistent with our observations and therefore requiring a higher dose of EVs whereas live *S. pyogenes* can potentially engage multiple inflammasome pathways and overall induces higher levels of IL-1 β .

With regard to the increased dosage used for BLaER1 cells, they have been shown to elicit a weaker cytokine response when stimulated with LPS compared to classical primary monocytes, which can be attributed to a defect in GPI-anchor biosynthesis¹⁰.

Referee #2:

1. The purification strategy, size, and abundance of the bacterial EVs used in this study are detailed in Figures EV1A and B. EV quantification was carried out using Nanoparticle Tracking Analysis (NTA),

which demonstrated higher reproducibility compared to protein quantification. Although NTA provided size measurements of EVs ranging from 100-150 nm, to better illustrate any heterogeneous features in the EV population and to visualise actual ultrastructural characteristics of the EVs, it is advisable that the authors also include transmission electron microscopy (TEM) analysis and show electron micrographs.

The isolation and characterization of *S. pyogenes* EVs derived from strain ISS3348 (also used in this study) were published by our laboratory in 2016⁹, where EVs were visualized via SEM, TEM, and STED microscopy. For the present manuscript, we included only NTA for EV quantification since this method is more reliable for determining the EV number, but all purification steps were kept as previously described. If required, further TEM images of our EV preparations could be added to the manuscript.

2. The authors suggest that EVs isolated from Spy and the Spy bacterial cell itself differentially modulate inflammasome responses. However, have the authors directly compared the effects of EV-associated LTA with freely released LTA on immune cells? It would be beneficial to compare the key effects of EV-associated LTA with those of culture supernatant after vesicle preparation (containing non-vesicle-associated LTA) to exclude the influence of other surface structures present on the bacteria. Additionally, what proportion of the total released LTA is associated with EVs?

The reviewer raises an interesting point. We already included in the manuscript experiments with purified LTA from *S. pyogenes* as well as *S. aureus*. To treat monocytes with culture supernatants after EV preparation, we would first have to assess the amount of LTA remaining in the media, which could be done and would answer the questions regarding the proportions between released and EV-associated LTA.

3. Pathway analysis of primary monocytes revealed a high similarity between KEGG pathways and GO terms for each treatment, with common upregulation of cytokines (IL1A, IL1B, IL6), chemokines (CCL3, CCL4, CXCL8), and growth factors (CSF3), and downregulation of immune receptors (TLR1, TLR6, CCR2) and adhesion factors (PECAM1, VCAN) across both stimuli. While cytokines such as IL1 and IL6 are upregulated, CCR2 levels are significantly downregulated. Given that CCR2 is the receptor for the chemokine CCL2 (MCP-1), it would be logical to assess the levels of CCL2 in the sample, as the CCR2-CCL2 pathway is directly and indirectly linked to IL1B and IL6 signalling. However, based on the information provided, it is unclear if the authors have assessed CCL2 (MCP-1) levels.

We did not measure the levels of CCL2 (MCP-1) via ELISA since we were primarily focussing on the role of caspase-4/-5 and the alternative inflammasome. According to our RNA-seq dataset, CCL2 is upregulated in primary monocytes by $\log_2 = 2,47$ upon EV treatment and $\log_2 = 3,18$ upon *S. pyogenes* infection. In BLaER1 cells, CCL2 is upregulated by $\log_2 = 4,37$ upon EV treatment and $\log_2 = 6,28$ upon *S. pyogenes* infection. Since CCL2 is upregulated by IL-1 β , it is plausible that we detect an increase in gene expression for CCL2 in *S. pyogenes*-infected cells, which also secrete higher levels of IL-1 β compared to EV treated cells. While we agree that this would be an interesting follow-up, we believe this is outside the scope of the present manuscript.

4. Could you explain why the metalloredutase STEAP4 is induced in macrophages treated with extracellular vesicles (EVs) but not when treated with whole bacterial cells, even though both treatments increase IL-6 and IL1B levels, which are known regulators of STEAP4?

We can only speculate regarding the specific upregulation of STEAP4 mRNA by EVs but not by *S. pyogenes*. Indeed, STEAP4 expression has been shown to be induced by various cytokines such as TNF, IL-6, or IL-1 β ¹¹. On the other hand, STEAP4 also negatively regulates the transcription of IL-6 and IL-8¹². It is therefore plausible that STEAP4 is upregulated in a sterile inflammatory setting such as EV treatment to prevent excessive cellular stress, whereas during bacterial infection, extensive inflammation is required to combat the invading pathogen. A paragraph can be added to the discussion if requested by the reviewer.

5. Did the authors examine whether IL-1 β release in their study occurs through gasdermin D pores, as typically observed in response to LPS stimulation, or if it might be mediated by some alternative pathway(s)? Specifically, have they considered analyzing gasdermin D activation, or explored the possibility of IL-1 β secretion through non-canonical pathways, such as microvesicle shedding, exosomes, or secretory lysosomes? Additionally, have they investigated the potential for IL-1 β release via inflammasome-independent or autophagy-dependent mechanisms?

Infection of human monocytes with *S. pyogenes* leads to cleavage of GSDMD and subsequent cell death. Here, IL-1 β secretion will be dependent on GSDMD. In contrast, treatment with EVs does not result in any measurable cell death or GSDMD cleavage, so IL-1 β secretion is most likely independent of GSDMD. Interestingly, we found a reduction in IL-1 β as well as IL-6 secretion in response to EVs when cells were treated with Bafilomycin A1, an inhibitor of the vacuolar-type H⁺ ATPase, suggesting a potential role for acidic compartments in cytokine secretion.

6) Discussion: The discussion is well-structured and comprehensive, providing a detailed analysis of the differential immune responses elicited by *Streptococcus pyogenes* and its extracellular vesicles (EVs) in human monocytes. The study's focus on TLR2 and caspase-4/-5 as key mediators adds valuable insight into the innate immune response to this pathogen. The comparison with existing literature, particularly the novel findings related to EVs, highlights the contribution of this study to the field. The discussion effectively addresses the complexity of immune responses, acknowledging the dual role of inflammation in both protecting against bacterial invasion and preventing excessive immune responses. However, there are some areas where the discussion could be strengthened. The exploration of the mechanistic differences between *S. pyogenes* and its EVs is somewhat limited, leaving a gap in understanding how these differences translate into specific immune responses. The speculative nature of the role of EVs in disease progression could be supported by more detailed experimental proposals. Moreover, the clinical implications of the findings, particularly regarding sepsis, could be further elaborated to clarify their potential impact. Finally, the roles of TLR2 and caspase-4/-5, though highlighted, could be better defined in terms of their biological significance under different conditions, making the discussion more precise and actionable.

We can revise the sections of the discussion addressing clinical implications as well as the roles of TLR2 and caspase-4/-5.

Referee #3:

The manuscript by Franch-Arroyo et al presents results demonstrating the activation of NLRP3 non-canonical inflammasome by EVs released from Gram-positive *Streptococcus pyogenes*. The results are interesting though activation by EVs from Gram-positive *Staphylococcus* has been demonstrated previously (Wang et al., PNAS 2020). Consequently, the novelty of the results cannot be unambiguously established. I also have a few other comments which the authors will hopefully find useful.

Regarding novelty issues, please see our response to point #1 of Referee 1.

1. Fig. EV3 - The levels of IL-1b secretion are too low even at the highest concentration of EVs used here. Two of the data points (out of 4) have values around approx. 50 pg/mL only. Thus, how reproducible the data is needs to be made clear.

We agree with the reviewer's observation and will repeat the experiment since when conducting it originally, we experienced unusual high variability in the monocyte response toward EV treatment between individual donors, with 2 donors showing very poor responses.

2. Related to the above, how are Fig. EV3 and Fig. 2A different so as to result in drastically different levels of IL-1b secretion?

As mentioned in our statement to point #1, the donor variability for Fig. EV3 was unusual, so we will repeat the experiment.

3. It is important to show Casp-5 expression by WB. Can the authors also show Casp-5 cleaved form with Spy EVs?

We will be able to provide Western Blots for cleaved caspases 4/-5 in response to *S. pyogenes* infection. However, for EV stimulation we were unable to detect a cleavage product for these caspases. This could be due to a sensitivity issue, since the EV-induced IL-1 β response is weaker than that induced by *S. pyogenes*, and cleaved caspases are notoriously difficult to detect. Alternatively, caspase activity does not always result in cleavage^{13,14}. Since we also do not observe any cell death in response to EV treatment, it is plausible to assume that caspase activation is mild and does not result in cleavage.

4. Immunoblot showing pro-IL-1b expression should be moved to main figures. Do the authors also see cleaved IL-1b expression by EVs.

The blots can be moved if requested by the reviewer. Western Blots for IL-1 β cleaved in response to EV treatment can be generated.

5. Experiment demonstrating the requirement (or not) of potassium flux should be conducted.

In agreement with what has been reported for the alternative inflammasome², we have data from BLaER1 monocytes, showing no impact of extracellular potassium on IL- β secretion for EV treatment. These can be added to the manuscript.

6. Line 225 - the description of results examining the role of RIP1K can be improved.

In agreement with the reviewer's suggestion, the section can be elaborated.

7. Line 241 - 'infection with MCC950 and ...' needs to be rephrased for clarity.

We apologize for this oversight and will rephrase this passage of the text.

8. Fig. 5F - The authors must expand further why LTA transfection wouldn't result in lower IL-1b levels as compared to direct LTA stimulation, particularly as LTA is packaged within Lipofectamine and may not be fully accessible for TLR2 stimulation.

Given that activation of the alternative inflammasome is typically weak compared to the canonical inflammasome², we would have assumed that the presence of another receptor recognizing cytosolic LTA would further enhance the IL-1 β response. Because we do not see any decrease in IL-1 β , we assume that there is still free LTA left in the mixture that causes inflammasome activation from the cell surface.

9. Further assays such as ASC oligomerisation may be conducted to fully validate the results obtained.

The manuscript contains experiments using ASC knockout BLaER1 cells to validate the role of ASC in alternative inflammasome activation. However, Gaidt et al. 2016² demonstrated that stimulation of the alternative inflammasome pathway does not result in ASC oligomerization.

References

1. Netea, M. G. *et al.* Differential requirement for the activation of the inflammasome for processing and release of IL-1 β in monocytes and macrophages. *Blood* **113**, 2324–2335 (2009).
2. Gaidt, M. M. *et al.* Human Monocytes Engage an Alternative Inflammasome Pathway. *Immunity* **44**, 833–846 (2016).
3. Oliva-Martin, M. J. *et al.* Caspase-8 inhibition represses initial human monocyte activation in septic shock model. *Oncotarget* **7**, 37456–37470 (2016).
4. Gritsenko, A. *et al.* Priming Is Dispensable for NLRP3 Inflammasome Activation in Human Monocytes In Vitro. *Front. Immunol.* **11**, 1–14 (2020).
5. Gao, Y. cFLIP S regulates alternative NLRP3 in inflammasome activation in human monocytes. (2023). doi:10.1038/s41423-023-01077-y
6. Unterberger, S., Mullen, L., Flint, M. S. & Sacre, S. Multiple TLRs elicit alternative NLRP3 inflammasome activation in primary human monocytes independent of RIPK1 kinase activity. *Front. Immunol.* **14**, 1092799 (2023).
7. Viganò, E. *et al.* Human caspase-4 and caspase-5 regulate the one-step non-canonical inflammasome activation in monocytes. *Nat. Commun.* **6**, 8761 (2015).
8. Schmid-Burgk, J. L. *et al.* Caspase-4 mediates non-canonical activation of the NLRP3 inflammasome in human myeloid cells. *Eur. J. Immunol.* **45**, 2911–2917 (2015).
9. Resch, U. *et al.* A two-component regulatory system impacts extracellular membrane-derived vesicle production in group a streptococcus. *MBio* **7**, 1–10 (2016).
10. Wegner, J. *et al.* An epigenetic GPI anchor defect impairs TLR4 signaling in the B cell transdifferentiation model for primary human monocytes. *Sci. Rep.* 1–15 (2021). doi:10.1038/s41598-021-94386-z
11. Scarl, R. T., Lawrence, C. M., Gordon, H. M. & Nunemaker, C. S. STEAP4: its emerging role in metabolism and homeostasis of cellular iron and copper. *J. Endocrinol.* **234**, R123–R134 (2017).
12. Tanaka, Y. *et al.* Six-transmembrane epithelial antigen of prostate4 (STEAP4) is a tumor necrosis factor alpha-induced protein that regulates IL-6, IL-8, and cell proliferation in synovium from patients with rheumatoid arthritis. *Mod. Rheumatol.* **22**, 128–136 (2012).
13. Stennicke, H. R. *et al.* Caspase-9 Can Be Activated without Proteolytic Processing. *J. Biol. Chem.* **274**, 8359–8362 (1999).
14. Guey, B., Bodnar, M., Manié, S. N., Tardivel, A. & Petrilli, V. Caspase-1 autoproteolysis is differentially required for NLRP1b and NLRP3 inflammasome function. *Proc. Natl. Acad. Sci.* **111**, 17254–17259 (2014).

Dear Dr. Charpentier,

Thanks for your letter asking me to re-consider my decision on your manuscript. I now went through your rebuttal letter (revision plan), and it seems that your plan addresses adequately the referee concerns (copied again below).

I thus would like to invite you to revise your manuscript with the understanding that the concerns of the referees must be addressed in the revised manuscript and in a detailed point-by-point response, as indicated in your revision plan. Acceptance of your manuscript will depend on a positive outcome of a second round of review. It is EMBO reports policy to allow a single round of revision only and acceptance of the manuscript will therefore depend on the completeness of your responses included in the next, final version of the manuscript.

Revised manuscripts should be submitted within three months of a request for revision. Please contact me to discuss the revision (also by video chat) if you have further questions or comments regarding the revision, or should you need additional time.

1) a .docx formatted version of the final manuscript text (including legends for main figures, EV figures and tables), but without the figures included. Figure legends should be compiled at the end of the manuscript text.

2) individual production quality figure files as .eps, .tif, .jpg (one file per figure), of main figures and EV figures. Please upload these as separate, individual files upon re-submission.

4) a complete author checklist, which you can download from our author guidelines

(<https://www.embopress.org/page/journal/14693178/authorguide>). Please insert page numbers in the checklist to indicate where the requested information can be found in the manuscript. The completed author checklist will also be part of the RPF.

5) that primary datasets produced in this study (e.g. RNA-seq, ChIP-seq, structural and array data) are deposited in an appropriate public database. If no primary datasets have been deposited, please also state this in a dedicated section (e.g. 'No primary datasets have been generated and deposited'), see below.

The accession numbers and database should be listed in a formal "Data Availability" section (placed after Materials & Methods) that follows the model below. This is now mandatory (like the COI statement). Please note that the Data Availability Section is restricted to new primary data that are part of this study. This section is mandatory. As indicated above, if no primary datasets have been deposited, please state this in this section

Data availability

8) Regarding data quantification and statistics, please make sure that the number "n" for how many independent experiments were performed, their nature (biological versus technical replicates), the bars and error bars (e.g. SEM, SD) and the test used to calculate p-values is indicated in the respective figure legends (also for EV figures and all those in an Appendix). Please also check that all the p-values are explained in the legend, and that these fit to those shown in the figure. Please provide statistical testing where applicable. Please avoid the phrase 'independent experiment', but clearly state if these were biological or technical replicates. Please also indicate (e.g. with n.s.) if testing was performed, but the differences are not significant. In case n=2, please show the data as separate datapoints without error bars and statistics. See also: <http://www.embopress.org/page/journal/14693178/authorguide#statisticalanalysis>

9) Please add scale bars of similar style and thickness to microscopic images, using clearly visible black or white bars (depending on the background). Please place these in the lower right corner of the images themselves. Please do not write on or near the bars in the image but define the size in the respective figure legend.

10) Please also note our reference format:

12) We now use CRediT to specify the contributions of each author in the journal submission system. CRediT replaces the author contribution section. Please use the free text box to provide more detailed descriptions and do NOT provide your final manuscript text file with an author contributions section. See also our guide to authors: <https://www.embopress.org/page/journal/14693178/authorguide#authorshipguidelines>

13) All Materials and Methods need to be described in the main text using our 'Structured Methods' format, which is required for all research articles. According to this format, the Methods section should include a Reagents and Tools Table (listing key reagents, experimental models, software, and relevant equipment and including their sources and relevant identifiers), uploaded as separate file, followed by a Methods section in which we encourage the authors to describe their methods using a step-by-

step protocol format with bullet points, to facilitate the adoption of the methodologies across labs. More information on how to adhere to this format as well as downloadable templates (.doc) for the Reagents and Tools Table can be found in our author guidelines (section 'Structured Methods'):

14) Please order the sections like this, using these names:

Title page - Abstract - Keywords - Introduction - Results - Discussion - Methods - Data availability section - Acknowledgements - Disclosure and Competing Interests Statement - References - Figure legends - Expanded View Figure legends

15) Please make sure that all the funding information is also entered into the online submission system and that it is complete and similar to the one in the acknowledgement section of the manuscript text file.

Finally, please note that all corresponding authors are required to supply an institutional e-mail address and an ORCID ID for their name upon submission of a revised manuscript. Please provide an institutional e-mail address for co-corresponding author Jin Peng and ORCID IDs for both corresponding authors.

Please find instructions on how to link the ORCID ID to the account in our manuscript tracking system in our Author guidelines: <http://www.embopress.org/page/journal/14693178/authorguide#authorshipguidelines>

I look forward to seeing a revised version of your manuscript when it is ready. Please let me know if you have questions or comments regarding the revision.

Best,

Referee #1:

The study investigates how human monocytes recognize and respond to extracellular vesicles derived from Gram-positive bacteria, specifically *Streptococcus pyogenes*. While the manuscript explores an interesting aspect of the immune response, I have three major concerns that I believe significantly impact the manuscript's suitability for publication.

1. Lack of Novelty: The authors aim to describe the activation of the inflammasome via TLR2, NLRP3, and caspase 4/5 in response to Gram-positive bacterial EVs. However, the potential of Gram-positive EVs to activate macrophages (PMID: 34253061, 36978183, 34959485), TLR2 (PMID: 34281154, 36077142), and the inflammasome (PMID: 31988111) has already been documented in the literature. As such, the novelty of this study is limited, and it does not significantly advance our current understanding of these processes.

2. Inadequate Isolation and Characterization of EVs: The isolation and characterization of EVs in this study do not meet the latest guidelines (MISEV 2023; PMID: 38326288). The methods used do not ensure the purity of the EV preparations, which undermines the validity of the conclusions drawn by the authors. Specifically, the lack of TEM imaging to confirm the presence of only EVs and the absence of other extracellular particles is a critical omission. Moreover, the authors have not demonstrated the presence of intact spherical nanostructures, which is essential for confirming the identity of the isolated EVs. Performing EV isolations with additional techniques (i.e., SEC or density gradients), characterizing them and repeating key experiments would be beyond the scope of a major revision.

The use of bacterial culture media, which is a known source of extracellular particles, raises further concerns. Given that the authors isolated EVs via ultracentrifugation, they cannot exclude the possibility of co-isolating contaminating particles from the media. To address this issue, the authors would have to repeat key experiments using EV preparations from plain bacterial media (without bacterial cultivation) to assess the immunogenicity of the media components. Only by doing so can they conclusively determine that the observed immune responses are due to the EVs and not contaminating particles.

3. Inappropriate Dosing: The dosing of EVs applied to human monocytes in this study is excessively high. For example, the authors applied 7×10^8 particles to 1×10^5 monocytes in a 96-well format, resulting in a ratio of 7,000 EVs per cell. This number was even higher (10 times more) for RNA isolation experiments, and for BLaER1 cells, the ratio was 35,000 EVs per cell. These doses are far beyond physiological EV ratios, making it unsurprising that the cells responded to the EVs.

Furthermore, there is a lack of justification for the comparison of infection with an MOI of 5 to stimulations with 7×10^8 particles for

monocytes and an MOI of 1 with 3.5×10^{10} particles for BLaER1 cells. The manuscript does not provide sufficient rationale for these dosing choices, which raises concerns about the validity of the conclusions drawn.

Given these significant concerns, I suggest that the manuscript be rejected. It does not meet the high standards expected of this journal, lacks novelty, and fails to adhere to the standards of the EV community in terms of isolation and characterization.

Referee #2:

The study investigates the immune response to *Streptococcus pyogenes* (Spy) extracellular vesicles (EVs) and their parental cells in human monocytes, revealing distinct transcriptional signatures and pathways activated by each. The research demonstrates that Spy EVs trigger IL-1 β release through the alternative inflammasome pathway involving TLR2, NLRP3, and caspases-4 and -5, with caspase-4/-5 activation partly dependent on lipoteichoic acid (LTA) from Gram-positive bacterial membranes. Additionally, the study highlights that caspase-4/-5 also plays a role in IL-1 β production in response to EVs from other Gram-positive species, indicating a conserved immune mechanism.

Comments:

- 1) The purification strategy, size, and abundance of the bacterial EVs used in this study are detailed in Figures EV1A and B. EV quantification was carried out using Nanoparticle Tracking Analysis (NTA), which demonstrated higher reproducibility compared to protein quantification. Although NTA provided size measurements of EVs ranging from 100-150 nm, to better illustrate any heterogeneous features in the EV population and to visualise actual ultrastructural characteristics of the EVs, it is advisable that the authors also include transmission electron microscopy (TEM) analysis and show electron micrographs.
- 2) The authors suggest that EVs isolated from Spy and the Spy bacterial cell itself differentially modulate inflammasome responses. However, have the authors directly compared the effects of EV-associated LTA with freely released LTA on immune cells? It would be beneficial to compare the key effects of EV-associated LTA with those of culture supernatant after vesicle preparation (containing non-vesicle-associated LTA) to exclude the influence of other surface structures present on the bacteria. Additionally, what proportion of the total released LTA is associated with EVs?
- 3) Pathway analysis of primary monocytes revealed a high similarity between KEGG pathways and GO terms for each treatment, with common upregulation of cytokines (IL1A, IL1B, IL6), chemokines (CCL3, CCL4, CXCL8), and growth factors (CSF3), and downregulation of immune receptors (TLR1, TLR6, CCR2) and adhesion factors (PECAM1, VCAN) across both stimuli. While cytokines such as IL1 and IL6 are upregulated, CCR2 levels are significantly downregulated. Given that CCR2 is the receptor for the chemokine CCL2 (MCP-1), it would be logical to assess the levels of CCL2 in the sample, as the CCR2-CCL2 pathway is directly and indirectly linked to IL1B and IL6 signalling. However, based on the information provided, it is unclear if the authors have assessed CCL2 (MCP-1) levels.
- 4) Could you explain why the metalloendopeptidase STEAP4 is induced in macrophages treated with extracellular vesicles (EVs) but not when treated with whole bacterial cells, even though both treatments increase IL-6 and IL1B levels, which are known regulators of STEAP4?
- 5) Did the authors examine whether IL-1 β release in their study occurs through gasdermin D pores, as typically observed in response to LPS stimulation, or if it might be mediated by some alternative pathway(s)? Specifically, have they considered analyzing gasdermin D activation, or explored the possibility of IL-1 β secretion through non-canonical pathways, such as microvesicle shedding, exosomes, or secretory lysosomes? Additionally, have they investigated the potential for IL-1 β release via inflammasome-independent or autophagy-dependent mechanisms?
- 6) Discussion: The discussion is well-structured and comprehensive, providing a detailed analysis of the differential immune responses elicited by *Streptococcus pyogenes* and its extracellular vesicles (EVs) in human monocytes. The study's focus on TLR2 and caspase-4/-5 as key mediators adds valuable insight into the innate immune response to this pathogen. The comparison with existing literature, particularly the novel findings related to EVs, highlights the contribution of this study to the field. The discussion effectively addresses the complexity of immune responses, acknowledging the dual role of inflammation in both protecting against bacterial invasion and preventing excessive immune responses. However, there are some areas where the discussion could be strengthened. The exploration of the mechanistic differences between *S. pyogenes* and its EVs is somewhat limited, leaving a gap in understanding how these differences translate into specific immune responses. The speculative nature of the role of EVs in disease progression could be supported by more detailed experimental proposals. Moreover, the clinical implications of the findings, particularly regarding sepsis, could be further elaborated to clarify their potential impact. Finally, the roles of TLR2 and caspase-4/-5, though highlighted, could be better defined in terms of their biological significance under different conditions, making the discussion more precise and actionable.

Referee #3:

The manuscript by Franch-Arroyo et al presents results demonstrating the activation of NLRP3 non-canonical inflammasome by EVs released from Gram-positive *Streptococcus pyogenes*. The results are interesting though activation by EVs from Gram-positive *Staphylococcus* has been demonstrated previously (Wang et al., PNAS 2020). Consequently, the novelty of the results cannot be unambiguously established. I also have a few other comments which the authors will hopefully find useful.

1. Fig. EV3 - The levels of IL-1b secretion are too low even at the highest concentration of EVs used here. Two of the data

- points (out of 4) have values around approx. 50 pg/mL only. Thus, how reproducible the data is needs to be made clear.
2. Related to the above, how are Fig. EV3 and Fig. 2A different so as to result in drastically different levels of IL-1b secretion?
 3. It is important to show Casp-5 expression by WB. Can the authors also show Casp-5 cleaved form with Spy EVs?
 4. Immunoblot showing pro-IL-1b expression should be moved to main figures. Do the authors also see cleaved IL-1b expression by EVs.
 5. Experiment demonstrating the requirement (or not) of potassium flux should be conducted.
 6. Line 225 - the description of results examining the role of RIP1K can be improved.
 7. Line 241 - "infection with MCC950 and ..." needs to be rephrased for clarity.
 8. Fig. 5F - The authors must expand further why LTA transfection wouldn't result in lower IL-1b levels as compared to direct LTA stimulation, particularly as LTA is packaged within Lipofectamine and may not be fully accessible for TLR2 stimulation.
 9. Further assays such as ASC oligomerisation may be conducted to fully validate the results obtained.

We thank the editor and reviewers for taking the time to read and evaluate our manuscript. Changes made to the manuscript file have been marked in color. Please find our point-by-point responses below.

Referee #1:

1. Lack of Novelty: The authors aim to describe the activation of the inflammasome via TLR2, NLRP3, and caspase 4/5 in response to Gram-positive bacterial EVs. However, the potential of Gram-positive EVs to activate macrophages (PMID: 34253061, 36978183, 34959485), TLR2 (PMID: 34281154, 36077142), and the inflammasome (PMID: 31988111) has already been documented in the literature. As such, the novelty of this study is limited, and it does not significantly advance our current understanding of these processes.

We politely disagree with the reviewer's assessment that there is lack of novelty. While the activation of macrophages by Gram-positive EVs has been studied more frequently (but still much less than studies on Gram-negative EVs), there are currently no publications examining the alternative pathway of inflammasome activation in primary human monocytes in response to Gram-positive bacterial species or their derived EVs. The studies cited by the reviewer were in fact all performed in macrophages, emanating upon monocyte differentiation, naturally in vivo in a tissue- and microenvironment-dependent manner or in vitro, mostly after several days of culture in plastic plates and possibly the addition of recombinant M-CSF/GM-CSF. For the experiments presented here, we carefully purified monocytes via CD14+/CD16- negative selection in order to obtain as many "naïve" monocytes as possible, which are naturally present circulating for 1-3 days in the blood. Given the organizational efforts needed for such studies, it is not surprising that for human monocytes, there are only 6 publications to date describing the alternative inflammasome: Netea et al. 2009¹ showed that monocytes can release processed IL-1beta in a single step pathway after stimulation by TLR2 and TLR4 using Pam3Cys, LPS, or heat-killed *S. epidermidis*. Then, Gaidt et al. 2016² elegantly characterized the LPS-induced alternative inflammasome pathway, which is a particular feature of human monocytes and signals through the TLR4-TRIF-RIPK1-FADD-CASP8 axis to activate CASP1. A role for CASP8 in LPS-induced IL-1β secretion by human monocytes was also reported by Oliva-Martin et al. 2016³. The study by Gritsenko et al. 2020⁴ corroborated the previous findings by demonstrating that priming is dispensable for NLRP3 inflammasome activation in human monocytes but not in macrophages, underlining the uniqueness of inflammasome activation in human monocytes. Gao et al. 2023⁵ also demonstrated the activation of the alternative inflammasome pathway and its dependence on CASP8 in response to different heat-inactivated Gram-negative bacteria. Finally, using different TLR ligands, Unterberger et al. 2023⁶ provided evidence that aside from TLR4, various TLRs can engage the alternative inflammasome in human monocytes. However, none of these publications investigated the activation of the alternative inflammasome in the presence of live bacteria or their derived EVs. In addition, the contribution of the non-canonical inflammasome to Gram-positive infection has only been demonstrated in mice for *S. aureus* and *L. monocytogenes*. However, no study to date has reported a role for caspase-4/-5 during Gram-positive infection in human cells. And lastly, the involvement of caspase-4/-5 in the alternative activation of the inflammasome has only been reported for LPS^{7,8}.

We apologize if this was not understandable in the manuscript and we have worked on the text to raise awareness of the host cell type and host response, but also of the pathogen and its secretome to make this clearer.

2. Inadequate Isolation and Characterization of EVs: The isolation and characterization of EVs in this study do not meet the latest guidelines (MISEV 2023; PMID: 38326288). The methods used do not ensure the purity of the EV preparations, which undermines the validity of the conclusions drawn by the authors. Specifically, the lack of TEM imaging to confirm the presence of only EVs and the absence

of other extracellular particles is a critical omission. Moreover, the authors have not demonstrated the presence of intact spherical nanostructures, which is essential for confirming the identity of the isolated EVs. Performing EV isolations with additional techniques (i.e., SEC or density gradients), characterizing them and repeating key experiments would be beyond the scope of a major revision.

The use of bacterial culture media, which is a known source of extracellular particles, raises further concerns. Given that the authors isolated EVs via ultracentrifugation, they cannot exclude the possibility of co-isolating contaminating particles from the media. To address this issue, the authors would have to repeat key experiments using EV preparations from plain bacterial media (without bacterial cultivation) to assess the immunogenicity of the media components. Only by doing so can they conclusively determine that the observed immune responses are due to the EVs and not contaminating particles.

The reviewer has raised a relevant point and we agree with the suggestion to use “EV preparations” from plain bacterial media as control to test whether there is any effect of media components alone on the monocytes. Therefore, we have now included in the manuscript data showing TLR2 activity (Fig. 3D) in response to *S. pyogenes* EVs and also to “EV” preparations from THB media alone (Mock). We found TLR2 activation in response to *S. pyogenes* EVs but not to the Mock treatment.

Regarding the argument that our study is lacking imaging of the EV preparations: The isolation and extensive characterization of *S. pyogenes* EVs derived from strain ISS3348 (also used in this study) were published by our laboratory in 2016⁹, where EVs were visualized via SEM, TEM, and STED microscopy. However, in line with the reviewer’s suggestion, we have repeated TEM imaging of our EV preparations and added them to the manuscript (Fig. EV1E).

As for the concern of inadequate EV isolation: While we acknowledge that our EV preparations via ultracentrifugation may still contain soluble protein contaminants despite of the extensive wash-step (i.e. a 1-3mm pellet resuspended in 70ml PBS), we are confident that our conclusions regarding Lipoteichoic Acid (LTA) being the major PAMP that drives alternative inflammasome activation in human monocytes are valid based on experiments showing that firstly, commercially available LTA from both *S. pyogenes* (Sigma Aldrich) and *S. aureus* (Invivogen, ultrapure) induced IL-1 β secretion by human monocytes. Secondly, inhibition of LTA synthesis in our bacterial cultures using the small molecule inhibitor LtaS-IN-1771 significantly reduced the immunogenic properties of the derived EV preparations, despite the presence of potential co-isolated contaminants from the media. Thirdly, we also isolated EVs using SEC and repeated the key experiments (Fig. 6E-J). We were able to show a similar effect of SEC EVs on TLR2 activation, monocyte survival and IL-6 secretion compared to our conventional EV preparations using ultracentrifugation (EV UC). Of note, the inability to detect significant amounts of IL-1 β in response to SEC EVs correlated with and is explained by the reduced concentrations of EV and LTA obtained after SEC compared to our EVs UC. Therefore, future studies warrant the processing of larger volumes via SEC following tangential flow filtration to avoid protein-overloading of SEC columns and ensure adequate EV and LTA concentrations, which we agree would be outside the scope of the present study.

3. Inappropriate Dosing: The dosing of EVs applied to human monocytes in this study is excessively high. For example, the authors applied 7e8 particles to 1e5 monocytes in a 96-well format, resulting in a ratio of 7,000 EVs per cell. This number was even higher (10 times more) for RNA isolation experiments, and for BLaER1 cells, the ratio was 35,000 EVs per cell. These doses are far beyond physiological EV ratios, making it unsurprising that the cells responded to the EVs.

Furthermore, there is a lack of justification for the comparison of infection with an MOI of 5 to

stimulations with 7×10^8 particles for monocytes and an MOI of 1 with 3.5×10^{10} particles for BLaER1 cells. The manuscript does not provide sufficient rationale for these dosing choices, which raises concerns about the validity of the conclusions drawn.

We apologize for this oversight. All bacterial infections for primary monocytes as well as BLaER1 cells were carried out with a MOI 5. We corrected the corresponding "Methods" section.

The dose of EVs for stimulation of human monocytes was originally assessed by comparing the IL-6 response against increasing concentrations of EVs with different concentrations of purified *S. pyogenes* LTA, where 7×10^8 EV particles induced a response similar to 1 $\mu\text{g}/\text{mL}$ LTA. However, the IL-1 β response of alternative inflammasome activation has been reported to be lower compared to the canonical/non-canonical inflammasome², consistent with our observations and therefore requiring a higher dose of EVs, while live *S. pyogenes* can potentially engage multiple inflammasome pathways and overall induces higher levels of IL-1 β . Furthermore, our data show that *S. pyogenes* LTA is less potent in activating TLR2 compared to LTA from *S. aureus*.

With regard to the increased dosage used for BLaER1 cells, they have been shown to elicit a weaker cytokine response when stimulated with LPS compared to classical primary monocytes, which can be attributed to a defect in GPI-anchor biosynthesis¹⁰.

Referee #2:

1. The purification strategy, size, and abundance of the bacterial EVs used in this study are detailed in Figures EV1A and B. EV quantification was carried out using Nanoparticle Tracking Analysis (NTA), which demonstrated higher reproducibility compared to protein quantification. Although NTA provided size measurements of EVs ranging from 100-150 nm, to better illustrate any heterogeneous features in the EV population and to visualise actual ultrastructural characteristics of the EVs, it is advisable that the authors also include transmission electron microscopy (TEM) analysis and show electron micrographs.

The isolation and characterization of *S. pyogenes* EVs derived from strain ISS3348 (also used in this study) were published by our laboratory in 2016⁹, where EVs were visualized via SEM, TEM, and STED microscopy. For the present manuscript, we included only NTA for EV quantification since this method is more reliable for determining the EV number, but all purification steps were kept as previously described. In agreement with the reviewer's suggestion, we repeated TEM imaging and added recordings of our EV preparations to the manuscript (Fig. EV1E).

2. The authors suggest that EVs isolated from *Spy* and the *Spy* bacterial cell itself differentially modulate inflammasome responses. However, have the authors directly compared the effects of EV-associated LTA with freely released LTA on immune cells? It would be beneficial to compare the key effects of EV-associated LTA with those of culture supernatant after vesicle preparation (containing non-vesicle-associated LTA) to exclude the influence of other surface structures present on the bacteria. Additionally, what proportion of the total released LTA is associated with EVs?

The reviewer raises an interesting point. We have added data to the manuscript (Fig. 6A-D) showing that free LTA in EV-depleted supernatants also promotes IL-1 β release and monocyte survival comparable to *Spy* EVs. In addition, we performed experiments with commercially available LTA from *S. pyogenes* (Sigma Aldrich) and *S. aureus* (Invivogen, ultrapure), together strengthening our claim that LTA is the main driver of human monocyte activation.

3. Pathway analysis of primary monocytes revealed a high similarity between KEGG pathways and GO terms for each treatment, with common upregulation of cytokines (IL1A, IL1B, IL6), chemokines (CCL3,

CCL4, CXCL8), and growth factors (CSF3), and downregulation of immune receptors (TLR1, TLR6, CCR2) and adhesion factors (PECAM1, VCAN) across both stimuli. While cytokines such as IL1 and IL6 are upregulated, CCR2 levels are significantly downregulated. Given that CCR2 is the receptor for the chemokine CCL2 (MCP-1), it would be logical to assess the levels of CCL2 in the sample, as the CCR2-CCL2 pathway is directly and indirectly linked to IL1B and IL6 signaling. However, based on the information provided, it is unclear if the authors have assessed CCL2 (MCP-1) levels.

Initially, we did not measure the levels of CCL2 (MCP-1) via ELISA because we were primarily focusing on the role of caspase-4/-5 and the alternative inflammasome. According to our RNA-seq dataset, CCL2 is upregulated in primary monocytes by $\log_2 = 2,47$ upon EV treatment and $\log_2 = 3,18$ upon infection with *S. pyogenes*. In BLaER1 cells, CCL2 is upregulated by $\log_2 = 4,37$ upon EV treatment and $\log_2 = 6,28$ upon *S. pyogenes* infection. Given that CCL2 is upregulated by IL-1 β , it is plausible that we could detect an increase in gene expression for CCL2 in *S. pyogenes*-infected cells, which also secrete higher levels of IL-1 β compared to EV treated cells.

During the revision process, we also measured CCL2 in some cell culture supernatants from monocytes stimulated with EVs and could detect significant CCL2 release. While we agree that this would be an interesting follow-up, given the emphasis on IL-1 β throughout the manuscript, we believe that this is beyond the scope of the present study.

4. Could you explain why the metalloredutase STEAP4 is induced in macrophages treated with extracellular vesicles (EVs) but not when treated with whole bacterial cells, even though both treatments increase IL-6 and IL1B levels, which are known regulators of STEAP4?

We can only speculate on the specific upregulation of STEAP4 mRNA by EVs but not by *S. pyogenes*. Indeed, it has been shown that STEAP4 expression is induced by various cytokines such as TNF, IL-6, or IL-1 β ¹¹. On the other hand, STEAP4 also negatively regulates the transcription of IL-6 and IL-8¹². It can be assumed that STEAP4 may be upregulated in a sterile inflammatory setting such as EV treatment to prevent excessive cellular stress, whereas during bacterial infection, extensive inflammation is required to combat the invading pathogen.

5. Did the authors examine whether IL-1 β release in their study occurs through gasdermin D pores, as typically observed in response to LPS stimulation, or if it might be mediated by some alternative pathway(s)? Specifically, have they considered analyzing gasdermin D activation, or explored the possibility of IL-1 β secretion through non-canonical pathways, such as microvesicle shedding, exosomes, or secretory lysosomes? Additionally, have they investigated the potential for IL-1 β release via inflammasome-independent or autophagy-dependent mechanisms?

Infection of human monocytes with *S. pyogenes* leads to cleavage of GSDMD and subsequent cell death. Here, IL-1 β secretion is dependent on GSDMD. In contrast, treatment of monocytes with EVs does not result in any measurable cell death or GSDMD cleavage, so IL-1 β secretion is independent of GSDMD. Interestingly, we found an increase in IL-1 β secretion in response to EVs when cells were treated with Bafilomycin A1, an inhibitor of the vacuolar-type H⁺ ATPase, suggesting a potential role for secretory lysosomes as described in the literature. In addition, we observed an increase in the autophagosome turnover in response to both *S. pyogenes* EVs and LTA, suggesting a role for autophagy in facilitating IL-1 β release. Additional data have been added to Fig.2 and Fig. 5.

6) Discussion: The discussion is well-structured and comprehensive, providing a detailed analysis of the differential immune responses elicited by *Streptococcus pyogenes* and its extracellular vesicles (EVs) in human monocytes. The study's focus on TLR2 and caspase-4/-5 as key mediators adds valuable insight into the innate immune response to this pathogen. The comparison with existing literature,

particularly the novel findings related to EVs, highlights the contribution of this study to the field. The discussion effectively addresses the complexity of immune responses, acknowledging the dual role of inflammation in both protecting against bacterial invasion and preventing excessive immune responses. However, there are some areas where the discussion could be strengthened. The exploration of the mechanistic differences between *S. pyogenes* and its EVs is somewhat limited, leaving a gap in understanding how these differences translate into specific immune responses. The speculative nature of the role of EVs in disease progression could be supported by more detailed experimental proposals. Moreover, the clinical implications of the findings, particularly regarding sepsis, could be further elaborated to clarify their potential impact. Finally, the roles of TLR2 and caspase-4/-5, though highlighted, could be better defined in terms of their biological significance under different conditions, making the discussion more precise and actionable.

We thank the reviewer for this comment and have revised the discussion to address the differences in inflammasome activation between *S. pyogenes* and its EVs as well as the role of TLR2. In addition, we have placed greater emphasis on the clinical implications of EV-induced inflammation.

Referee #3:

The manuscript by Franch-Arroyo et al presents results demonstrating the activation of NLRP3 non-canonical inflammasome by EVs released from Gram-positive *Streptococcus pyogenes*. The results are interesting though activation by EVs from Gram-positive *Staphylococcus* has been demonstrated previously (Wang et al., PNAS 2020). Consequently, the novelty of the results cannot be unambiguously established. I also have a few other comments which the authors will hopefully find useful.

Regarding novelty issues, please see our response to point #1 of Referee 1.

1. Fig. EV3 - The levels of IL-1b secretion are too low even at the highest concentration of EVs used here. Two of the data points (out of 4) have values around approx. 50 pg/mL only. Thus, how reproducible the data is needs to be made clear.

We agree with the reviewer's observation and have repeated these experiments with additional donors because we have observed an unusually high variability in the monocyte response to EV treatment between individual donors, with two donors showing very poor responses. In fact, careful examination of these donor-specific, individual EV-responses in a clinical setting may be the subject of future follow up studies.

2. Related to the above, how are Fig. EV3 and Fig. 2A different so as to result in drastically different levels of IL-1b secretion?

As mentioned in our statement in point #1, the variability of donors for Fig. EV3 (now Fig. EV1F) was unusual, so we repeated these experiments. However, we still occasionally observe a high degree of variability between donors.

3. It is important to show Casp-5 expression by WB. Can the authors also show Casp-5 cleaved form with Spy EVs?

We were able to provide Western Blots for cleaved caspases 4/-5 in response to infection with *S. pyogenes* (Fig. 2E). However, for EV stimulation, we were unable to detect a cleavage product for these caspases. This could be due to a sensitivity issue, since the EV-induced IL-1 β response is weaker than that induced by *S. pyogenes*, and cleaved caspases are notoriously difficult to detect. Alternatively, caspase activity does not always result in cleavage^{13,14}. Since we also do not observe any cell death in response to EV treatment, it is plausible to assume that caspase activation is mild and does not result in cleavage. Immunoblots showing Pro-CASP5 expression in response to *Spy* EVs and *Spy* LTA have been added to Figures 2F and 5H.

4. Immunoblot showing pro-IL-1b expression should be moved to main figures. Do the authors also see cleaved IL-1b expression by EVs.

Due to space limitations we did not move the immunoblot showing pro-IL-1 β to the main figures. However, we have generated immunoblots for cleaved IL-1 β , which can be found for *Spy* EVs in main Fig. 2F and for *Spy* LTA in main Fig. 5H.

5. Experiment demonstrating the requirement (or not) of potassium flux should be conducted.

In agreement with what has been reported for the alternative inflammasome², we have added data from BLAER1 monocytes, showing no impact of extracellular potassium on IL- β secretion or LDH release for EV treatment.

6. Line 225 - the description of results examining the role of RIP1K can be improved.

According to the reviewer's suggestion, the discussion regarding the alternative inflammasome and the molecules involved has been revised.

7. Line 241 - 'infection with MCC950 and ...' needs to be rephrased for clarity.

We apologize for this oversight. The passage has been reworded.

8. Fig. 5F - The authors must expand further why LTA transfection wouldn't result in lower IL-1b levels as compared to direct LTA stimulation, particularly as LTA is packaged within Lipofectamine and may not be fully accessible for TLR2 stimulation.

Indeed, given that the activation of the alternative inflammasome is typically weak compared to the canonical inflammasome², we would have assumed that the presence of another receptor recognizing cytosolic LTA would further enhance the IL-1 β response. We can only speculate as to why we do not see any decrease in IL-1 β , but we hypothesize that there might be still free LTA remaining in the mixture that causes inflammasome activation from the cell surface. In addition, TLR2 has been reported to be expressed in early/late endosomes, lysosomes and Rab11⁺ compartments¹⁵. Therefore, the transfected cytosolic LTA could be delivered to the lysosome network for degradation and presumably activate TLR2 from these compartments.

9. Further assays such as ASC oligomerization may be conducted to fully validate the results obtained.

Gaidt et al. 2016² demonstrated that stimulation of the alternative inflammasome pathway does not result in ASC oligomerization. In agreement with these findings, we did not observe ASC speck formation in response to *Spy* EVs (Fig. 4G).

References

1. Netea, M. G. *et al.* Differential requirement for the activation of the inflammasome for processing and release of IL-1 β in monocytes and macrophages. *Blood* **113**, 2324–2335 (2009).
2. Gaidt, M. M. *et al.* Human Monocytes Engage an Alternative Inflammasome Pathway. *Immunity* **44**, 833–846 (2016).
3. Oliva-Martin, M. J. *et al.* Caspase-8 inhibition represses initial human monocyte activation in septic shock model. *Oncotarget* **7**, 37456–37470 (2016).
4. Gritsenko, A. *et al.* Priming Is Dispensable for NLRP3 Inflammasome Activation in Human Monocytes In Vitro. *Front. Immunol.* **11**, 1–14 (2020).
5. Gao, Y. cFLIP S regulates alternative NLRP3 in inflammasome activation in human monocytes. (2023) doi:10.1038/s41423-023-01077-y.
6. Unterberger, S., Mullen, L., Flint, M. S. & Sacre, S. Multiple TLRs elicit alternative NLRP3 inflammasome activation in primary human monocytes independent of RIPK1 kinase activity. *Front. Immunol.* **14**, 1092799 (2023).
7. Viganò, E. *et al.* Human caspase-4 and caspase-5 regulate the one-step non-canonical inflammasome activation in monocytes. *Nat. Commun.* **6**, 8761 (2015).
8. Schmid-Burgk, J. L. *et al.* Caspase-4 mediates non-canonical activation of the NLRP3 inflammasome in human myeloid cells. *Eur. J. Immunol.* **45**, 2911–2917 (2015).
9. Resch, U. *et al.* A two-component regulatory system impacts extracellular membrane-derived vesicle production in group a streptococcus. *MBio* **7**, 1–10 (2016).
10. Wegner, J. *et al.* An epigenetic GPI anchor defect impairs TLR4 signaling in the B cell transdifferentiation model for primary human monocytes. *Sci. Rep.* 1–15 (2021).
11. Scarl, R. T., Lawrence, C. M., Gordon, H. M. & Nunemaker, C. S. STEAP4: its emerging role in metabolism and homeostasis of cellular iron and copper. *J. Endocrinol.* **234**, R123–R134 (2017).
12. Tanaka, Y. *et al.* Six-transmembrane epithelial antigen of prostate4 (STEAP4) is a tumor necrosis factor alpha-induced protein that regulates IL-6, IL-8, and cell proliferation in synovium from patients with rheumatoid arthritis. *Mod. Rheumatol.* **22**, 128–136 (2012).
13. Stennicke, H. R. *et al.* Caspase-9 Can Be Activated without Proteolytic Processing. *J. Biol. Chem.* **274**, 8359–8362 (1999).
14. Guey, B., Bodnar, M., Manié, S. N., Tardivel, A. & Petrilli, V. Caspase-1 autoproteolysis is differentially required for NLRP1b and NLRP3 inflammasome function. *Proc. Natl. Acad. Sci.* **111**, 17254–17259 (2014).
15. Nilsen, N. J. *et al.* Cellular trafficking of lipoteichoic acid and Toll-like receptor 2 in relation to signaling: role of CD14 and CD36. *J. Leukoc. Biol.* **84**, 280–291 (2008).

Dear Prof. Charpentier,

Thank you for the submission of your revised manuscript to our editorial offices. I have already forwarded to you the reports from the three referees that were asked to re-evaluate the study, you will find also below. As you know, referees #2 and #3 now support the publication of your manuscript in EMBO reports, whereas referee #1 still has several concerns, or states that previous concerns have not been adequately addressed, and maintains the opinion that the manuscript should not be published in its present form. However, I have shared your further p-b-p-response (revision plan) with referee #1 and s/he indicates that the responses adequately address the remaining concerns.

I thus invite you to revise your manuscript further with the understanding that the remaining concerns of the referees will be addressed in the revised manuscript as indicated in your rebuttal letter. I agree with your statement that the suggestion of referee #2 regarding the expression and role of CCL2 in the context of CCR2 downregulation is beyond the scope of the present manuscript.

Please also provide a final detailed point-by-point response to all remaining referee points.

- Please order the sections like this, using these names:

Title page - Abstract - Keywords - Introduction - Results - Discussion - Methods - Data availability section - Acknowledgements - Disclosure and Competing Interests Statement - References - Figure legends - Expanded View Figure legends

- Please check again that the number "n" for how many independent experiments were performed, their nature (biological versus technical replicates), the bars and error bars (e.g. SEM, SD) and the test used to calculate p-values is indicated in the respective figure legends. Please also check that all the p-values are explained in the legend, and that these fit to those shown in the figure. Please provide statistical testing where applicable. Please avoid the phrase 'independent experiment' but clearly state if these were biological or technical replicates. Please also indicate (e.g. with n.s.) if testing was performed, but the differences are not significant. In case n=2, please show the data as separate datapoints without error bars and statistics. See also:

<http://www.embopress.org/page/journal/14693178/authorguide#statisticalanalysis>

If n<5, please show single datapoints for diagrams. Presently some diagrams have no statistics (e.g. 5G, 5H left panel, 6G or EV5F) or miss the 'n.s.'. Please check. Moreover:

- Please note that the exact p values are not provided in the legends of figures 2A, B, C, D, F, G; 3A, B, C, E, F, G; 4A-H; 5D, E, H, I, J, K; 6A, B, H; EV1 F; EV2 E, F, H; EV3 A, C; EV4 A, EV5 A, D, E, G, K.

- Please indicate the statistical test used for data analysis in the legend of figure 1C

- Please note that the box plots need to be defined in terms of minima, maxima, centre, bounds of box and whiskers, and percentile in the legends of figures EV2 E, F, G, H; EV4 D.

- Please note that information related to n is missing in the legend of figure 3B

- Please note that the error bars are not defined in the legends of figures 2G, 3B

- Please add to each legend (main, EV and Appendix figures, where applicable) a 'Data Information' section explaining the statistics used or providing information regarding replicates and scales. See:

- Please add scale bars of similar style and thickness to microscopic images (main and EV figures), using clearly visible black or white bars (depending on the background). Please place these in the lower right corner of the images themselves. Please do not write on or near the bars in the image but define the size in the respective figure legend. Presently, some scale bars are too small and all have text nearby. Please check.

- Please use our reference format (et al needs to be used after 10 author names; DOIs should only be used for preprints and datasets that have not been published yet):

- Please make sure that all the funding information is also entered into the online submission system and that it is complete and similar to the one in the acknowledgement section of the manuscript text file. Grants from the Max Planck Society and the Tang Prize Foundation are missing in the submission system.

- There are 2 datasets uploaded. Please correct the nomenclature (source file names, titles in the system, and callouts) to 'Dataset EV1' and 'Dataset EV2'. Moreover, please provide their title and legends only on the first TAB of the respective excel file. Thus please remove the dataset legends from the Appendix file.

- Please add/move the primer information (Appendix Tables S1-S3) to the Reagents and Tools Table and remove the tables from the Appendix. Please change any callouts (see 'Reagents and Tools Table').
- It is not necessary to repeat all the author information on the title page of the Appendix. Please just state there 'Appendix for ...' (followed by the title of the paper), followed by the TOC.
- Please add the specific URLs for PRJNA753844, PXD046570 and S-BIAD1725 to the data availability statement.
- In case you add new data to the main figures during final revision, please also provide related source data.

In addition, I would need from you uploaded separately:

Best,

Referee #1:

Thank you for the opportunity to re-evaluate the revised version of this manuscript. While the authors have addressed several points raised in the initial round of review and made additional efforts (e.g., inclusion of SEC-purified EV data and TEM imaging), I find that a number of substantial concerns remain unaddressed or insufficiently resolved. As such, I still do not support the publication of this manuscript in EMBO Reports in its current form.

1. Insufficient detail and validation for SEC EV characterization

While the inclusion of SEC data is appreciated, the methodology remains insufficiently described and validated:

- The presence of EVs in SEC fractions was determined by SYPRO Ruby staining, but no gel image or fractionation profile is provided.
- Protein content alone does not confirm the presence of EVs, and without accompanying nanoparticle tracking data (NTA/Exoid/...), TEM, or EV marker analysis per fraction, the conclusions remain speculative.
- The pooling strategy for SEC fractions is not transparent: it is unclear which pooled fractions were used for Exoid measurements. This information is critical for reproducibility and interpretation of the data.

2. Lack of clarity on LTA contamination and implications for EV purity and immunogenicity

The newly included data using SEC-purified EVs (Fig. 6H) clearly show a marked reduction in LTA content compared to ultracentrifugation (UC)-derived EVs. This strongly suggests significant LTA co-isolation in the UC-preparations. Moreover, the authors acknowledge that IL-1 β induction by SEC-EVs is minimal or absent, although this is not shown in the figures. This discrepancy between UC- and SEC-derived EV responses argues against an EV-mediated effect and instead implicates contaminating LTA as the primary immune activator. Despite these findings, the authors do not adequately discuss this critical point or reconsider their conclusions accordingly.

3. Unexplained effects of EV-depleted controls (Fig. 6B)

The IL-6 response observed in cells treated with THB or EV-depleted supernatant is puzzling and unexplained. Even more concerning is the cytotoxic effect of EV-depleted supernatants. This raises the possibility that non-vesicular soluble factors or media components could contribute to the observed responses and undermines the claim that EVs alone are responsible.

4. Unphysiological dosing and missing justification

The applied EV dose (7×10^8 particles per 1×10^5 monocytes) results in approximately 7,000 particles per cell, which is far from physiological. The authors try to justify this by matching the IL-6 response to 1 $\mu\text{g/mL}$ of purified LTA, yet this benchmark is itself artificial. What is the amount of LTA per EV coming from SEC-purification?

Crucially, SEC-EVs, even at similar or higher doses, fail to elicit comparable IL-6 or IL-1 β responses. This inconsistency further

supports the notion that the inflammatory response is driven by LTA or other contaminants in UC-preparations rather than by bona fide EVs. The authors have still not addressed whether such high numbers of vesicles would be released by bacteria under physiological conditions or during infection, nor do they provide a vesicle-per-bacterium estimate from their cultures which could justify their dosing decision.

5. Inconclusive evidence for caspase-4/-5 activation

The claim that caspase-4/-5 mediates inflammasome activation in response to Gram-positive EVs is not substantiated by the presented data. The immunoblots now included show only pro-caspase-5 but do not demonstrate cleavage or activation. This is a critical shortcoming, especially given that caspase-4/-5 cleavage is central to the manuscript's main hypothesis.

6. Lack of LC3-II induction and misinterpretation of data (Fig. 2G)

Despite the authors' claim of increased LC3-II levels upon EV stimulation, this is not supported by the presented immunoblot or quantification. No visible difference is observed between untreated and EV-treated monocytes. This undermines the interpretation that autophagy is induced by EVs and casts doubt on the relevance of this pathway in the context of the current study.

Conclusion

Although the manuscript addresses a timely and interesting topic, the central claims regarding EV-specific activation of the non-canonical and alternative inflammasome pathways in primary human monocytes remain unconvincing. The excessive EV dosing, unresolved contamination concerns, lack of robust caspase-4/-5 activation data, and incomplete EV characterization collectively limit the strength and novelty of the conclusions. I therefore maintain my recommendation against publication of this manuscript in EMBO Reports.

Referee #2:

Comment 1: Include TEM images to illustrate EV heterogeneity and ultrastructure

The authors included TEM analysis of *S. pyogenes* EVs, which visually confirms their structure and supports the NTA data. This addition addresses my comment on visualizing EV heterogeneity.

Comment 2: Compare EV-associated LTA with free LTA and quantify LTA distribution

The authors compared the effects of purified LTA and EV-depleted supernatants, including from a Δ slo mutant to avoid confounding cell death. They measured LTA concentrations before and after EV isolation and found that ~45% of total LTA is EV-associated. These experiments directly respond to my request for comparison and quantification.

Comment 4: Explain STEAP4 induction in response to EVs but not bacteria

The unique upregulation of STEAP4 by EVs is highlighted as an EV-specific signature and connected to chronic inflammation, obesity, and diabetes. While the precise mechanism isn't fully dissected, its EV-specific induction is recognized and discussed in the context of chronic inflammatory responses.

Comment 5: GSDMD activation and non-canonical IL-1 β release

The authors evaluated GSDMD cleavage and ASC speck formation. They found no cleavage or cell death with EVs, suggesting an alternative secretion route (e.g., via secretory lysosomes or autophagy), supported by increased LC3-II and BafA1-induced IL-1 β release. This fulfills my request for mechanistic exploration.

Comment 6: Improve discussion on differential sensing, mechanistic depth, and clinical relevance

The revised discussion is significantly expanded. It includes deeper mechanistic insights, especially on TLR2 vs. TLR8 roles, caspase-4/5 activation routes, and speculative implications for inflammatory diseases and sepsis. Potential clinical relevance and future directions are clearly emphasized.

The authors have satisfactorily addressed nearly all of my comments, particularly those related to the experimental and mechanistic aspects. The only point that may still benefit from further clarification concerns the expression and role of CCL2 in the context of CCR2 downregulation (Comment #3). While I agree that this topic may require further investigation, it could offer valuable insights into the potential link between CCL2-CCR2 signaling, IL1B/IL6 pathways and *Streptococcus pyogenes* EVs.

Referee #3:

This is a substantially improved revision of the previously submitted manuscript. The authors have provided clear and thoughtful responses to all reviewer comments, including new experimental data where appropriate. Importantly, they have now clarified the novelty of their findings, particularly regarding extracellular vesicles (EVs) derived from Gram-positive bacteria, which represents a significant contribution to the field.

As articulated in the revised manuscript and the rebuttal, the EV-related findings are novel and well-supported by robust experimental evidence. However, I remain only slightly convinced that the mechanistic insights into inflammasome activation are equally novel. The signalling pathways involved, while competently studied, appear to reinforce rather than extend current understanding.

One minor but important suggestion: In the final paragraph of the Introduction, the authors should also summarise their key findings related to inflammasome activation following infection with the parental *Streptococcus pyogenes* strain, in addition to the EV results. This will provide a more complete context for the study and better frame the comparison between EV-mediated and *Streptococcus* infection-driven responses.

Overall, the manuscript is now much stronger and suitable for publication, pending the minor revision noted above.

We thank the editor and reviewers for re-evaluating our revised manuscript. Additional changes made to the manuscript file have been marked in yellow. Please find our point-by-point responses below.

Reviewer #1

Thank you for the opportunity to re-evaluate the revised version of this manuscript. While the authors have addressed several points raised in the initial round of review and made additional efforts (e.g., inclusion of SEC-purified EV data and TEM imaging), I find that a number of substantial concerns remain unaddressed or insufficiently resolved. As such, I still do not support the publication of this manuscript in EMBO Reports in its current form.

1. Insufficient detail and validation for SEC EV characterization

While the inclusion of SEC data is appreciated, the methodology remains insufficiently described and validated:

- The presence of EVs in SEC fractions was determined by SYPRO Ruby staining, but no gel image or fractionation profile is provided.

We thank Reviewer 1 for acknowledging our additional efforts in regard to *Spy* EV characterization. As anticipated from TEM, UC EVs are heterogenous in size, which was confirmed by our initial NTA-based quantification.

This methodology is often criticized due to the aggregation/fusion of EVs under high g-forces, such as those used here (175.000xg) and co-precipitation of soluble proteins that may also occur. To minimize co-purification of non-apo-EV-proteins, SEC was performed and representative gel-images from 3 independent *S. pyogenes* cultures were provided in the Appendix (Appendix Figure S2A), as shown below.

A

- Protein content alone does not confirm the presence of EVs, and without accompanying nanoparticle tracking data (NTA/Exoid/...), TEM, or EV marker analysis per fraction, the conclusions remain speculative.

We fully agree with Reviewer 1. On the basis of SDS-PAGE-protein profiles, we would not have been able to determine which fractions contained EVs (SYPRO-Ruby has a sensitivity of 0.25-1ng) and also NTA was not sensitive enough to measure size and concentration in individual fractions. Therefore, we opted for the more sensitive TRPS-methodology as described first by Rienk Nieuwland and colleagues¹. Using Exoid, we were able to determine size and concentration using 100 and 150 nm nanopores.

The characteristic blockage traces and size-concentration histograms of UC-EVs and the pooled SEC fractions are summarized below in Figure 1-Reviewer 1. In addition, we provide the reviewer with traces from individual SEC-fractions, as shown in Figure 2-Reviewer 1. **Both figures were added to the appendix (Appendix Fig. S2B and C).** We also include early eluting non-EV-containing trace and traces already containing soluble proteins, characterized by high background currents (rms) due to charged soluble proteins and salts, the main feature of non-EV containing SEC-fractions. Furthermore, as soluble proteins are sticking to the polyurethan-fabricated nanopores, they become usually irreversibly blocked. Note that UC-EVs did not display higher background currents, meaning that they are depleted from soluble proteins to a large extent and capture the continuum-sized-spectrum of naturally released EVs.

Figure 1-Reviewer 1: UC- and pooled SEC-EV-quantification with TPRS/ Exoid. Plots of representative blockage events over time, termed traces with size&concentration histograms and quantification results inscribed in respective tables (UC-EVs on the left, pooled SEC-EVs on the right). Note that with the Exoid, traces are acquired at 3 different pressures (P1,2,3), P1 is 200Pa lower than P2 and P3 is 300Pa higher than P2 in one duty cycle. Appropriately diluted samples were measured with a NP100 and 47mm stretch and CPC100 calibration particles.

Figure 2-Reviewer 1: Characterization of Individual SEC-fractions. Trace-plots of individual SEC-fractions at one pressure (2500Pa, 1200mV, 60sec on NP100) with available quantification results summarized in a table.

• The pooling strategy for SEC fractions is not transparent: it is unclear which pooled fractions were used for Exoid measurements. This information is critical for reproducibility and interpretation of the data.

Following TRPS and protein estimations, the EV fractions were pooled as indicated in Appendix Figure S2A (red frame, see above). We performed SEC using 70nm qEV-columns from Izon and manually collected 0.5 mL fractions, which may result in some variability in SEC-fractions containing EVs. Because *S. pyogenes* and Gram-positive bacteria in general release much less EVs as compared to Gram-negative bacteria such as *E. coli* or *V. cholerae*, large culture volumes are needed and the concentration of culture supernatants adds an additional variable in the EVs isolated (we used 30kDa

cut-off-filters, but some EVs may still be lost in this way). Lastly, an additional SEC-EV concentration step (10 kDa cut-off-filter) is required to obtain EV-concentrations suitable for experiments with PBMCs/monocytes. We added a short paragraph regarding SEC EV isolation to the discussion (line 591-596).

2. Lack of clarity on LTA contamination and implications for EV purity and immunogenicity

The newly included data using SEC-purified EVs (Fig. 6H) clearly show a marked reduction in LTA content compared to ultracentrifugation (UC)-derived EVs. This strongly suggests significant LTA co-isolation in the UC-preparations. Moreover, the authors acknowledge that IL-1 β induction by SEC-EVs is minimal or absent, although this is not shown in the figures. This discrepancy between UC- and SEC-derived EV responses argues against an EV-mediated effect and instead implicates contaminating LTA as the primary immune activator. Despite these findings, the authors do not adequately discuss this critical point or reconsider their conclusions accordingly.

We kindly draw attention to the fact that evidence for IL-1 β secretion in response to SEC EVs was provided in Appendix Figure S2C (now Appendix Fig. S3B).

c

As stated in lines 401-402 (subheading of SEC EV results section), 432-434, 588-589, and 605-608 (discussion), we clearly acknowledge that LTA is the major driver of monocyte activation. Likewise, cytokine release from monocytes in response to our LTA-depleted EVs using the LTA synthesis inhibitor 1771 (line 370-379, Figure 5J) is significantly reduced, further confirming the critical role of *S. pyogenes* LTA -akin to LTA from other Gram-positive bacteria - in monocyte's activation, IL-6 and IL-1 β release, as well as caspase-4/-5/-8-dependent IL-1 β secretion. The corresponding experimental evidence is shown in Fig. 5D and EV5D (commercially available *S. pyogenes* and *S. aureus* LTA induces caspase-4/-5-dependent IL-1 β but independent IL-6 secretion). Similarly, the immunogenicity of EVs isolated from *S. aureus*, *B. subtilis* and *S. agalactiae* is shown (Fig. 5J and EV5H-I).

Since EVs originating from Gram-positive bacteria have been described as being produced by cytoplasmic membrane protrusions that pass through the cell wall either via turgor pressure, proteases or cell-wall weakening substances², LTA should always be a component of Gram-positive EVs. Therefore, vesicular LTA should activate immune cells and is not a contaminant - by analogy, LPS is a constituent of all Gram-negative OMVs. Furthermore, we even hypothesize that LTA in EVs (tightly attached or integral and thus not part of a labile EV-corona) could be more stable and persist longer in the host, as has been described for other virulence factors such as *Listeria*'s LLO³ or EHEC hemolysin⁴ for example.

3. Unexplained effects of EV-depleted controls (Fig. 6B)

The IL-6 response observed in cells treated with THB or EV-depleted supernatant is puzzling and unexplained. Even more concerning is the cytotoxic effect of EV-depleted supernatants. This raises the possibility that non-vesicular soluble factors or media components could contribute to the observed responses and undermines the claim that EVs alone are responsible.

We politely disagree with the assessment that the effects of EV-depleted supernatants (SUPs) are unexplained. We have provided evidence in Figure 6B that the cytotoxic effect of EV-depleted SUPs from WT *S. pyogenes* is due to the presence of Streptolysin O (Slo). Therefore, we additionally prepared EVs and EV-depleted SUP from a *slo* deletion (Δslo) strain. As indicated in Figure 6B, EV-depleted SUP from the *slo* deletion strain does not induce cell death (similar amounts of LDH release as observed in untreated or THB-treated control cells). However, EV-depleted SUP from the *slo* deletion strain, induces the secretion of IL-1 β (whereas THB alone does not) and IL-6. The LTA content was similar in EV-depleted SUPs derived from WT and Δslo *S. pyogenes* (Figure 6C).

Although we recognize that THB also induces IL-6, we can only speculate to explain this phenomenon. For the experiment, the medium was concentrated using Amicon Ultra-15 (10 kDa cut-off-filter) approximately 120-fold. This leads to increased concentrations of salt, peptones and beef heart infusion ingredients. Indeed, salt has been shown to activate immune cells⁵, including monocytes⁶, leading to IL-6 expression.

4. Unphysiological dosing and missing justification

The applied EV dose (7×10^8 particles per 1×10^5 monocytes) results in approximately 7,000 particles per cell, which is far from physiological. The authors try to justify this by matching the IL-6 response to 1 μ g/mL of purified LTA, yet this benchmark is itself artificial. What is the amount of LTA per EV coming from SEC-purification?

Crucially, SEC-EVs, even at similar or higher doses, fail to elicit comparable IL-6 or IL-1 β responses. This inconsistency further supports the notion that the inflammatory response is driven by LTA or other contaminants in UC-preparations rather than by bona fide EVs. The authors have still not addressed whether such high numbers of vesicles would be released by bacteria under physiological conditions or during infection, nor do they provide a vesicle-per-bacterium estimate from their cultures which could justify their dosing decision.

Based on our CFU calculations, the ratio between bacteria and EVs in the THB culture is approximately $4.4 \times 10^{10} : 7.8 \times 10^{11}$ (corresponding to ~20 (17.7) EVs/bacterial cell, line 142-143). However, as shown in our dose response experiment in Figure EV1F, we were unable to detect IL-1 β below EV concentrations of 7×10^8 EVs per 100,000 monocytes (corresponding to 7000EVs/cell).

S. pyogenes LTA has been used at various concentrations in cell culture experiments, such as 5 $\mu\text{g}/\text{mL}$ ⁷, 30 $\mu\text{g}/\text{mL}$ ⁸ or even 250 $\mu\text{g}/\text{mL}$ ⁹. However, our own dose response experiments in human monocytes revealed that the minimal concentration of LTA required to achieve measurable IL-1 β secretion was 0,5 $\mu\text{g}/\text{mL}$ (Figure 3-Reviewer 1). The figure was added to the Appendix (Appendix Fig. S3C). In our experimental setup, the 1 $\mu\text{g}/\text{mL}$ dose of LTA yielded robust results and, compared to other studies, is not excessive. Additionally, matching the LTA to the EV response allowed us to establish a detection limit without which it would have been difficult to draw meaningful conclusions. Measurement of LTA concentration in SEC EVs revealed 0.171 and 0.413 μg LTA per 1.9×10^8 particles (190 EVs per cell) for SEC1 and SEC2, respectively (line 427). Given that LTA content in SEC EVs is below our established detection threshold for IL-1 β secretion, it is not surprising that we observed only minimal cytokine release from monocytes treated with SEC EVs. Nevertheless, similar to UC EVs, stimulation with SEC EVs abrogated caspase-3 activation and induced p44/42 phosphorylation as well as TLR 2 activation (Fig. 6H-J), indicating that higher dosages of SEC EVs would yield a stronger cytokine response. Thus, massive or recurrent stimulation of immune cells by LTA-containing bacterial EVs could very well contribute to certain disease conditions, as we highlighted in the discussion. Overall, we do not deny that the immune response observed throughout the manuscript are LTA driven. In fact, we state this several times throughout the manuscript, as mentioned above in response to comment 2.

Figure 3- Reviewer 1: IL-1 β secretion from human monocytes treated with different doses of *S. pyogenes* LTA.

With regard to the reviewer's comment on unphysiological dosing, we would kindly ask the reviewer to please define what is meant by "physiological"? To our knowledge, there is no published report showing quantitative data on the number of EVs released by a bacterium during an in vivo infection. Such quantitation might be possible from TEM images or probably from extrachromosomal encoded engineered proteins targeted for EV-secretion, a strategy used for example in EV-based immunotherapy. For *S. pyogenes*, EV release has been shown to be regulated by the bacterial two-component system (TCS) CovRS¹⁰ as well as by penicillin treatment¹¹. However, *S. pyogenes* possesses several TCSs that have not yet been studied in detail¹² and their role in EV production is unknown. Therefore, the release of EVs from *S. pyogenes* into the host could be influenced by several signals that could potentially result in a higher number of EV than when the bacteria are cultured in THB medium. Furthermore, it is known from bacterial infections that host cells vary in their ability to internalize bacteria, indicating that there is always intrinsic heterogeneity^{13,14} in immune responses. Since we do not have a specific marker for bacterial EVs, it is not feasible to measure vesicle uptake on a single cell level. Nevertheless, as stated in the discussion, EV-induced immune responses (which,

as indicated several times throughout the manuscript, are primarily dependent on LTA), albeit significantly less than those elicited by bacteria, still may have long-lasting effects in certain disease conditions (such as sepsis or potentially chronic/recurring infections where increased EV release is warranted). We added this to the discussion (line 608-611 and line 640-643).

5. Inconclusive evidence for caspase-4/-5 activation

The claim that caspase-4/-5 mediates inflammasome activation in response to Gram-positive EVs is not substantiated by the presented data. The immunoblots now included show only pro-caspase-5 but do not demonstrate cleavage or activation. This is a critical shortcoming, especially given that caspase-4/-5 cleavage is central to the manuscript's main hypothesis.

Indeed, we did not find conclusive evidence for the cleavage of caspases-4/-5 in response to EVs via immunoblotting. We assembled the presented main figure (upper left) and the uncropped immunoblot exposures (upper right) in Figure 4-Reviewer 1, as shown below. The same applies when cells were stimulated with LTA (bottom Figure 5H and bottom right, uncropped exposures).

Supplementary information to Figure 2F: Immunoblot analysis of caspase-5 in monocyte cell lysates after stimulation with Spy EVs.

F

Monocyte lysates in RIPA (replicate 1) or Trizol-protein pellet (replicate 2-6). Lane 1: untreated, lane2: Spy EVs, lane 3: EVs+BafA1, lane 4: Spy EVs+ATG4 inhibitor. Red rectangle: Bands recognized by HSP90 antibody

Supplementary information to Figure 5H. Immunoblot analysis of caspase-5 in monocyte cell lysates after stimulation with Spy LTA

H

Lane 1: untreated, lane2: LTA, lane 3: LTA+BafA1, lane 4: LTA+ATG4 Inhibitor

Figure 4-Reviewer 1: Immunoblot analysis of caspase-5 in monocyte cell lysates following stimulation with Spy EVs or LTA. Left: Assembled exposures presented in main figures 2F (upper) or 5H (lower). Right: uncropped exposures of immunoblots incubated with anti-caspase-5 antibody showing bands for inducible, full-length pro-caspase-5 (p47) but no detectable cleaved caspase-5 fragment(s).

In fact, only stimulation with *S. pyogenes* led to detectable amounts of cleaved caspase-4/-5 in the supernatants of infected monocytes, as shown in Figure 2E and the uncropped exposures provided for Reviewer 1 below (Figure 5-Reviewer 1).

Supplementary information on caspase-4/-5 in monocytes-supernatants following incubation with *S. pyogenes* cells or *Spy* EVs

Figure 5-Reviewer 1: (Left) Immunoblot analysis of caspase-4/-5 in monocyte culture supernatants following stimulation with *S. pyogenes* or *Spy* EVs. (Right) ELISA for caspase-4 in monocyte culture supernatants.

Yet, slightly increased levels of active caspase-4/-5 in response to *Spy* EVs were detected using the fluorogenic substrate Ac-LEVD-AFC (Figure 2C). Of note, cleavage products for inflammatory caspases are notoriously difficult to detect. Given that the EV-induced IL-1 β response is significantly weaker than that induced by *S. pyogenes*, this could be due to a sensitivity issue that prevents us from detecting caspase cleavage in response to EVs with currently available methods. However, caspase activity does not always result in cleavage¹⁴⁻¹⁶. In fact, initiator caspases¹⁷, but also caspase-1¹⁸, have been reported to be activated by dimerization with cleavage either serving to stabilize the dimer or to terminate protease activity¹⁹. Since we also do not observe any cell death, GSDMD cleavage or ASC speck formation in response to EV treatment, it is plausible to assume that caspase activation is mild and does not necessarily result in cleavage. **A paragraph was added to the discussion (line 557-566).**

6. Lack of LC3-II induction and misinterpretation of data (Fig. 2G)

Despite the authors' claim of increased LC3-II levels upon EV stimulation, this is not supported by the presented immunoblot or quantification. No visible difference is observed between untreated and EV-treated monocytes. This undermines the interpretation that autophagy is induced by EVs and casts doubt on the relevance of this pathway in the context of the current study.

The experiment was performed in response to a request from Reviewer 2, who presumably considered it a valuable information for the current study. The immunoblot in Figure 2G shows an increase in the levels of LC3-II in the BafA1-negative samples for 60', 90' and 120' EV treatment (orange frame) compared to BafA1-negative untreated cells. Likewise, at the same time points, LC3-II levels are also increasing in the BafA1-positive samples (black frame) compared to BafA1-positive untreated cells. In the quantification graph below the dotted line represents the mean LC3-II intensity in untreated BafA1-negative cells. The mean LC3-II intensities for the EV-treatment are above the dotted line. In addition, the difference between BafA1-negative and -positive samples (defined as autophagic flux) is significant at 30' and 120' for all 4 replicates, indicating an increase in autophagic

flux upon EV treatment. Since autophagy is a dynamic & highly variable process, the time course was chosen to draw sufficient conclusions rather than limiting the analysis to a single time point. Differences between biological replicates in the timing of autophagy induction are to be expected. However, the results were verified by repeating the same experiment with LTA instead of EV treatment (Figure 5I). A significant increase in autophagic flux was also observed for 30', 60' and 90' of LTA treatment.

Conclusion

Although the manuscript addresses a timely and interesting topic, the central claims regarding EV-specific activation of the non-canonical and alternative inflammasome pathways in primary human monocytes remain unconvincing. The excessive EV dosing, unresolved contamination concerns,

lack of robust caspase-4/-5 activation data, and incomplete EV characterization collectively limit the strength and novelty of the conclusions. I therefore maintain my recommendation against publication of this manuscript in EMBO Reports.

With regard to the statement of excessive EV-dosing: based on publicly available and sufficiently detailed information on particles/cell ratios used by other research groups, we do not agree with the reviewer's assessment. For example, Liam-Or et al.²⁰ reported 2×10^9 EVs/ 1.5×10^5 J77.4/HepG2 cells (1.3×10^4 particles/cell). Xu L et al.²¹ reported 2×10^{10} particles/ 3×10^5 Panc-1 cells (6.6×10^5 particles/cell). In a recent study by Hagey DW et al.²², the transcriptional response of fibroblasts to EVs from different sources was tested at doses ranging from 20 to 2×10^5 EVs per cell. Therein, lysosomal pathway signatures were observed with the highest EV-dosage, but this systematic study provided a wealth of novel insights that needs to be considered in EV (sender) – cell (receiver) experimental design. Here, we titrated and evaluated a biologically relevant *Spy* EVs dosage based on particle count (and not on protein content, as mostly reported in bacterial-EV – host cell response experiments, line 585-587) and on the basis of measurements of IL-1 β release, TLR2 activation and cytotoxicity (LDH).

Reviewer #2

Comment 1: Include TEM images to illustrate EV heterogeneity and ultrastructure

The authors included TEM analysis of *S. pyogenes* EVs, which visually confirms their structure and supports the NTA data. This addition addresses my comment on visualizing EV heterogeneity.

Comment 2: Compare EV-associated LTA with free LTA and quantify LTA distribution

The authors compared the effects of purified LTA and EV-depleted supernatants, including from a Δ slo mutant to avoid confounding cell death. They measured LTA concentrations before and after EV isolation and found that ~45% of total LTA is EV-associated. These experiments directly respond to my request for comparison and quantification.

Comment 4: Explain STEAP4 induction in response to EVs but not bacteria

The unique upregulation of STEAP4 by EVs is highlighted as an EV-specific signature and connected to chronic inflammation, obesity, and diabetes. While the precise mechanism isn't fully dissected, its EV-specific induction is recognized and discussed in the context of chronic inflammatory responses.

Comment 5: GSDMD activation and non-canonical IL-1 β release

The authors evaluated GSDMD cleavage and ASC speck formation. They found no cleavage or cell death with EVs, suggesting an alternative secretion route (e.g., via secretory lysosomes or autophagy), supported by increased LC3-II and BafA1-induced IL-1 β release. This fulfils my request for mechanistic exploration.

Comment 6: Improve discussion on differential sensing, mechanistic depth, and clinical relevance

The revised discussion is significantly expanded. It includes deeper mechanistic insights, especially on TLR2 vs. TLR8 roles, caspase-4/5 activation routes, and speculative implications for

inflammatory diseases and sepsis. Potential clinical relevance and future directions are clearly emphasized.

The authors have satisfactorily addressed nearly all of my comments, particularly those related to the experimental and mechanistic aspects. The only point that may still benefit from further clarification concerns the expression and role of CCL2 in the context of CCR2 downregulation (Comment #3). While I agree that this topic may require further investigation, it could offer valuable insights into the potential link between CCL2-CCR2 signaling, IL1B/IL6 pathways and *Streptococcus pyogenes* EVs.

We thank the reviewer for taking the time to re-evaluate our revised manuscript. We feel this concern is beyond the scope of the manuscript, but we ask the reviewer if she/he has a specific experimental setup in mind to address the request regarding the last comment (just above).

Reviewer #3

This is a substantially improved revision of the previously submitted manuscript. The authors have provided clear and thoughtful responses to all reviewer comments, including new experimental data where appropriate. Importantly, they have now clarified the novelty of their findings, particularly regarding extracellular vesicles (EVs) derived from Gram-positive bacteria, which represents a significant contribution to the field.

As articulated in the revised manuscript and the rebuttal, the EV-related findings are novel and well-supported by robust experimental evidence. However, I remain only slightly convinced that the mechanistic insights into inflammasome activation are equally novel. The signalling pathways involved, while competently studied, appear to reinforce rather than extend current understanding.

One minor but important suggestion: In the final paragraph of the Introduction, the authors should also summarise their key findings related to inflammasome activation following infection with the parental *Streptococcus pyogenes* strain, in addition to the EV results. This will provide a more complete context for the study and better frame the comparison between EV-mediated and *Streptococcus* infection-driven responses.

Overall, the manuscript is now much stronger and suitable for publication, pending the minor revision noted above.

We thank the reviewer for taking the time to re-evaluate our revised manuscript. In agreement with the reviewer's suggestion, we modified the last paragraph of the introduction and included key findings of inflammasome activation following *S. pyogenes* infection (line 111-120).

References

1. Coumans, F. A. W. *et al.* Reproducible extracellular vesicle size and concentration determination with tunable resistive pulse sensing. *J. Extracell. vesicles* **3**, 25922 (2014).
2. Cao, Y. & Lin, H. Characterization and function of membrane vesicles in Gram-positive bacteria. *Appl. Microbiol. Biotechnol.* **105**, 1795–1801 (2021).
3. Coelho, C. *et al.* *Listeria monocytogenes* virulence factors, including listeriolysin O, are

- secreted in biologically active extracellular vesicles. *J. Biol. Chem.* **294**, 1202–1217 (2019).
4. Bielaszewska, M. *et al.* Enterohemorrhagic *Escherichia coli* hemolysin employs outer membrane vesicles to target mitochondria and cause endothelial and epithelial apoptosis. *PLoS Pathog.* **9**, e1003797 (2013).
 5. Balan, Y., Packirisamy, R. M. & Mohanraj, P. S. High dietary salt intake activates inflammatory cascades via Th17 immune cells: impact on health and diseases. *Arch. Med. Sci.* **18**, 459–465 (2022).
 6. Ruggeri Barbaro, N. *et al.* Sodium activates human monocytes via the NADPH oxidase and isolevuglandin formation. *Cardiovasc. Res.* **117**, 1358–1371 (2021).
 7. Hasty, D. L. *et al.* Monocyte and Macrophage Activation by Lipoteichoic Acid Is Independent of Alanine and Is Potentiated by Hemoglobin1. *J. Immunol.* **176**, 5567–5576 (2006).
 8. Blease, K., Chen, Y., Hellewell, P. G. & Burke-Gaffney, A. Lipoteichoic Acid inhibits Lipopolysaccharide-Induced Adhesion Molecule Expression and IL-8 Release in Human Lung Microvascular Endothelial Cells1. *J. Immunol.* **163**, 6139–6147 (1999).
 9. Sela, S., Marouni, M. J., Perry, R. & Barzilai, A. Effect of lipoteichoic acid on the uptake of *Streptococcus pyogenes* by HEp-2 cells. *FEMS Microbiol. Lett.* **193**, 187–193 (2000).
 10. Resch, U. *et al.* A two-component regulatory system impacts extracellular membrane-derived vesicle production in group a streptococcus. *MBio* **7**, 1–10 (2016).
 11. Biagini, M. *et al.* The human pathogen *Streptococcus pyogenes* releases lipoproteins as lipoprotein-rich membrane vesicles. *Mol. Cell. Proteomics* **14**, 2138–2149 (2015).
 12. Sitkiewicz, I. & Musser, J. M. Expression microarray and mouse virulence analysis of four conserved two-component gene regulatory systems in group a streptococcus. *Infect. Immun.* **74**, 1339–1351 (2006).
 13. Avraham, R. *et al.* Pathogen Cell-to-Cell Variability Drives Heterogeneity in Host Immune Responses. *Cell* **162**, 1309–1321 (2015).
 14. Krause, K. *et al.* CASP4/caspase-11 promotes autophagosome formation in response to bacterial infection. *Autophagy* **14**, 1928–1942 (2018).
 15. Stennicke, H. R. *et al.* Caspase-9 Can Be Activated without Proteolytic Processing. *J. Biol. Chem.* **274**, 8359–8362 (1999).
 16. Guey, B., Bodnar, M., Manié, S. N., Tardivel, A. & Petrilli, V. Caspase-1 autoproteolysis is differentially required for NLRP1b and NLRP3 inflammasome function. *Proc. Natl. Acad. Sci.* **111**, 17254–17259 (2014).
 17. Green, D. R. Caspase Activation and Inhibition. *Cold Spring Harb. Perspect. Biol.* **14**, (2022).
 18. Conos, S. A., Lawlor, K. E., Vaux, D. L., Vince, J. E. & Lindqvist, L. M. Cell death is not essential for caspase-1-mediated interleukin-1 β activation and secretion. *Cell Death Differ.* **23**, 1827–1838 (2016).
 19. Boucher, D. *et al.* Caspase-1 self-cleavage is an intrinsic mechanism to terminate inflammasome activity. *J. Exp. Med.* **215**, 827–840 (2018).
 20. Liam-Or, R. *et al.* Cellular uptake and in vivo distribution of mesenchymal-stem-cell-derived

extracellular vesicles are protein corona dependent. *Nat. Nanotechnol.* **19**, 846–855 (2024).

21. Xu, L. *et al.* Design of experiment (DoE)-driven in vitro and in vivo uptake studies of exosomes for pancreatic cancer delivery enabled by copper-free click chemistry-based labelling. *J. Extracell. vesicles* **9**, 1779458 (2020).
22. Hagey, D. W. *et al.* The cellular response to extracellular vesicles is dependent on their cell source and dose. *Sci. Adv.* **9**, eadh1168 (2023).

Prof. Emmanuelle Charpentier
Max Planck Unit for the Sciences of Pathogens
Chariteplatz 1
Berlin, Berlin 10117
Germany

Dear Prof. Charpentier,

Thank you for the submission of your final revised manuscript to our editorial offices. I now went through it and your final p-b-p-response and see that the remaining points of the referees are adequately addressed, as indicated in your revision plan. As you know, referee #1 already looked through your revision plan and stated the his/her remaining concerns have been addressed (see below).

I am thus very pleased to accept your manuscript for publication in the next available issue of EMBO reports. Thank you for your contribution to our journal.

Yours sincerely,

Referee #1:

Thank you for sharing the authors' point-by-point response. I have carefully reviewed the document and would like to thank the authors for their clarifications and thoughtful revisions. Their responses adequately address the concerns I raised in the previous review round.

I therefore support publication of the further revised manuscript in EMBO Reports.